# Zero-Inflated Bandits

**Haoyu Wei** [*][1]  **Runzhe Wan** [*][2]  **Lei Shi** [2]  **Rui Song** [2]

## Abstract

Many real-world bandit applications are characterized by sparse rewards, which can significantly hinder learning efficiency. Leveraging problem-specific structures for careful distribution modeling is recognized as essential for improving estimation efficiency in statistics. However, this approach remains under-explored in the context of bandits. To address this gap, we initiate the study of zero-inflated bandits, where the reward is modeled using a classic semi-parametric distribution known as the zero-inflated distribution. We develop algorithms based on the Upper Confidence Bound and Thompson Sampling frameworks for this specific structure. The superior empirical performance of these methods is demonstrated through extensive numerical studies.

## 1. Introduction

Bandit algorithms have been widely applied in areas such as clinical trials (Durand et al., 2018), finance (Shen et al., 2015), recommendation systems (Zhou et al., 2017), among others. Accurate uncertainty quantification is key to addressing the exploration-exploitation trade-off and typically requires on certain reward distribution assumptions. Existing assumptions can be roughly classified into two groups:

- **Parametric**: the reward distribution is assumed to belong to a parameterized family, such as Gaussian or Bernoulli distributions (Audibert et al., 2010; Krause & Ong, 2011; Agrawal & Goyal, 2012). The strong assumption typically ensures clean algorithmic design and nice theoretical results. However, when misspecified, it may lead to over- or under-exploration.

---

[*]Equal contribution [1]Department of Economics, University of California San Diego, La Jolla, USA [2]Amazon, Seattle, USA. This work does not relate to the positions at Amazon. Correspondence to: Haoyu Wei <h8wei@ucsd.edu>, Rui Song <songray@gmail.com>.

*Proceedings of the $42^{nd}$ International Conference on Machine Learning*, Vancouver, Canada. PMLR 267, 2025. Copyright 2025 by the author(s).

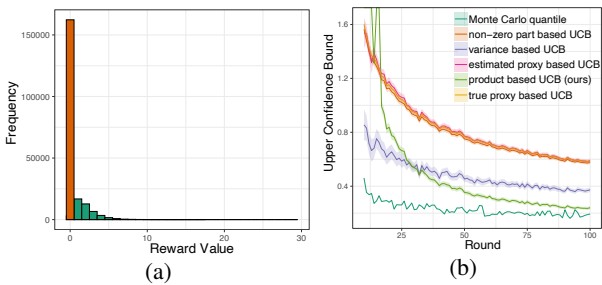

*Figure 1.* Results from a real personalized pricing dataset detailed in Section 5. (a) Histogram of rewards, with zero represented in orange. (b) Comparison of $1 - \delta$ upper confidence bounds across different methods. We use Monte Carlo to approximate the true quantile (the tightest valid upper confidence bound). All methods are validated as their curves lie above the Monte Carlo baseline. Our proposed method (green) achieves the tightest bound quickly, demonstrating effective utilization of the zero-inflated structure. The other bounds correspond to UCB baselines detailed in Appendix D.1. Notably, applying existing concentration inequalities directly on the reward (yellow), even when knowing the true size parameter but without utilizing the ZI structure, results in significantly looser bounds.

- **Non-parametric**: the reward distributions need to satisfy certain characteristics, such as having sub-Gaussian tails (Chowdhury & Gopalan, 2017; Jin et al., 2021; Zhu & Tan, 2020) or being bounded (Kveton et al., 2019; 2020; Kalvit & Zeevi, 2021). In some cases, such an approach can achieve regret rates comparable to those of parametric methods (Urteaga & Wiggins, 2018; Kalvit & Zeevi, 2021; Jin et al., 2021). However, in general, these weaker assumptions sacrifice statistical efficiency by ignoring structural information. Even when the rates are the same, there may still be a significant empirical performance gap between the two approaches when the parametric distribution is correctly specified (see Chapter 9 in Lattimore & Szepesvári, 2020).

As in many statistics and machine learning areas, using problem-specific structures to focus on a more detailed distribution family, if correctly specified, can improve the efficiency of estimation or uncertainty quantification. This leads to lower regrets in bandits. However, as discussed above, compared to the vast statistical literature on univari-

*Table 1.* Summary of the theoretical guarantees for our algorithms.

| BANDIT PROBLEM | ALGORITHM† | REWARD DISTRIBUTION | MINIMAX RATIO |
|---|---|---|---|
| MAB† | UCB | sub-Gaussian (Algorithm 1) | $\sqrt{\log T}$ |
| | | sub-Weibull (Algorithm 1) | $\sqrt{\log T}^{*}$ |
| | | heavy-tailed (Algorithm B.1) | $\sqrt{K/T} + (K/T)^{\frac{\epsilon-1}{2(1+\epsilon)}} \log^{\frac{\epsilon}{1+\epsilon}} K^{**}$ |
| | TS | sub-Gaussian (Algorithm C.1) | 1 |
| GLM | UCB | sub-Gaussian (Algorithm C.2) | $\log(d \vee q) \log(T/(d \wedge q))^{***}$ |
| | TS | sub-Gaussian (Algorithm C.3) | $(d \vee q) \log(d \vee q) \log(T/(d \wedge q))$ |

† Our MAB algorithms are designed for a fixed horizon $T$, but the *doubling trick* can adapt them into anytime algorithms with comparable guarantees (Section 6.2 in Lattimore & Szepesvári, 2020), preserving all the problem-dependent regret bounds (Theorems 7 and 9 in Besson & Kaufmann, 2018);

*, ** Our problem-dependent and problem-independent regrets achieve state-of-the-art performance for both sub-Weibull (Hao et al., 2019) and heavy-tailed (Bubeck et al., 2013) rewards;

*** Our problem-independent regret matches the minimax lower bound up to a logarithmic factor (Theorem 3 in Dani et al., 2008) and aligns with the state-of-the-art rate (Li et al., 2017).

ate distribution, this direction is underexplored in bandits. This paper initiates the study of this direction by focusing on the sparse reward problem. Specifically, this work is motivated by the observation that rewards tend to be sparse in many real-world applications, meaning they are zero (or a constant) in most instances, called zero-inflated (ZI). For instance, in online advertising, most customers will not click the advertisement and hence the reward is zero with high probability; while for those clicked, the reward will then follow a certain distribution. Similar structures exist in broad applications, including mobile health (Ling, 2019) and freemium games (Yu et al., 2021). While some standard bandit algorithms can still be applied, they fail to utilize the distribution property and hence can be less efficient. See Figure 1a for a real example.

**Contributions.** Our contributions are threefold. First, we propose a general bandit algorithm framework for zero-inflated bandit (ZIB). Both Upper Confidence Bound (UCB) and Thompson Sampling (TS)-type algorithms are proposed. Using the problem-specific structure, our algorithm is more efficient than the existing ones via more accurate uncertainty quantification. We illustrate this with Figure 1b, which shows that our method leads to tighter concentration bounds, and this will translate into lower regrets when used with UCB and TS. Our algorithms are designed for a wide range of reward distributions, including the sub-Weibull distribution and even more heavy-tailed distributions (with moments exceeding one). Second, we theoretically derive the regret bounds for our UCB and TS algorithms in multi-armed bandits (MAB) under weak reward distribution assumptions, as well as for contextual linear bandits. In many cases, our algorithms achieve regret rates that are either minimax optimal or state-of-the-art. A detailed summary is provided in Table 1. To our knowledge, this is the

first finite-sample concentration analysis of the general ZI models in the literature, even outside of bandits. Lastly, we show the value of our approach through both simulated and real experiments.

**Related work.** Besides the bandit literature with parametric or nonparametric reward distributions discussed above, our work connects to several related research areas. First, there is research on semiparametric bandits (Krishnamurthy et al., 2018; Kim & Paik, 2019; Ou et al., 2019; Peng et al., 2019; Choi et al., 2023). However, these works focus on addressing the misspecification of the regression function within the context of contextual bandits. This focus is orthogonal to our objectives. Second, the zero-inflated distribution can also be regarded as a special case of hierarchical distributions. In recent years, there is growing interest in leveraging hierarchical models in bandits (Hong et al., 2022; Wan et al., 2021; 2023a). However, all of them study the hierarchical structure among bandit instances (with meta-learning) instead of in the reward distribution as in our setup. Third, to be agnostic to the distribution assumption, besides relying on nonparametric distribution families, one may also consider bootstrap-based methods (Wan et al., 2023b; Kveton et al., 2019). Nonetheless, on one hand, these methods still rely on certain restrictive distribution assumptions to ensure a regret guarantee; on the other hand, they fail to fully utilize the problem-specific structure, which may lead to compromised efficiency. Fourth, our work is related to research on sparse rewards and heavy-tailed/asymmetric bandits. While traditional sparse bandits (Kwon et al., 2017; Perrault et al., 2020) focus on arms with zero mean rewards, our zero-inflated framework addresses structural sparsity where actions frequently yield zero rewards. This connects to heavy-tailed bandits (Bubeck et al., 2013; Zhang & Cutkosky, 2022) and asymmetric ban-

dits (Zhang & Ong, 2021; Li et al., 2023), as zero-inflated distributions can exhibit heavy-tailed or asymmetric properties. However, our approach is specifically tailored to the zero-inflated structure, allowing the non-zero component to be either heavy-tailed or asymmetric while maintaining computational efficiency.

Finally, while zero-inflated structures have been studied in offline settings (supervised/unsupervised learning, see Lambert, 1992; Hall, 2000; Cheung, 2002) and there is some literature focusing on model-specific zero-inflated bandits (Liu et al., 2023, which only considers two discrete count rewards: Poisson and negative binomial), to our knowledge, this work is the first formal study on this topic in bandits generally, which poses unique challenges such as finite-sample concentration bounds and regret rate analysis for a class of distributions.

## 2. Zero-Inflated Multi-Armed Bandits

In this section, we use the MAB setting to outline our motivation and strategy. We will extend to contextual bandits in Section 3.

**Setup.** For any positive integer $M$, we denote the set $\{1, \ldots, M\}$ by $[M]$. We start our discussion with the MAB problem: on each round $t \in [T]$, the agent can choose an action $A_t \in \mathcal{A}$ with the action space $\mathcal{A} = [K]$, and then receive a random reward $R_t = r_{A_t} + \varepsilon_t$, where $r_k = \mathbb{E}[R_t \mid A_t = k]$ is the mean reward of the $k$-th arm and $\varepsilon_t$ is the random error. The performance of a bandit algorithm is measured by the cumulative regret $\mathcal{R}(T) = \sum_{t=1}^{T} \mathbb{E}\big[\max_{a \in \mathcal{A}} r_a - r_{A_t}\big]$. We focus on applications where the reward is zero for a significant proportion of time, and propose to characterize the reward distribution by the following Zero-Inflated (ZI) model:

$$X_t = \mu_{A_t} + \varepsilon_t,$$
$$Y_t \sim \text{Bernoulli}(p_{A_t}),$$
$$\text{and} \quad R_t = 0 \times (1 - Y_t) + X_t \times Y_t.$$

Here, for each arm $k$, we introduce two unknown parameters, the non-zero probability $p_k \in [0, 1]$ and the mean of the non-zero part $\mu_k$ such that $r_k = \mu_k \times p_k$. Here $\varepsilon_t$ is a mean-zero random error term. For any arm $k$, we assume $\mathbb{P}(X_t = 0) = 0$. We note this assumption can always be satisfied: given a reward variable $R_t$, one can always define $Y_t = \mathbb{1}(R_t \neq 0)$ and $X_t = \mathbb{1}(R_t \neq 0) \times R_t$. Moreover, as such, it is natural to regard $Y_t$ as observable as well. In contrast, the value of $X_t$ is only observable when $Y_t \neq 0$ (equivalently, $R_t \neq 0$), and in this case it is equal to $R_t$. For simplicity of notations and without loss of generality, we assume $X_t \perp\!\!\!\perp Y_t$: even if $X_t \not\perp\!\!\!\perp Y_t$, we can re-define $\check{X}_t \mid Y_t = y$ to have the same distribution as $X_t \mid Y_t = 1$ for $y = 0, 1$. In this case, $\check{X}_t \perp\!\!\!\perp Y_t$, and the observable

reward $R_t = X_t \times Y_t = \check{X}_t \times Y_t$; thus, we can simply replace $X_t$ with $\check{X}_t$. Finally, the conditional distribution of $R_t$ is a mixture of two distributions, one of which is a delta distribution on zero and the other is only required to satisfy minimal assumptions; while the assignment $Y_t$ is Bernoulli. For simplicity, we will occasionally omit the subscript $t$ when there is no ambiguity.

To simplify the exposition, we first focus on pulling a single arm and may omit the subscript $k$. We begin with considering scenarios where $\varepsilon_t$ exhibits relatively light tails. Specifically, we consider the sub-Weibull tail property, i.e., there exists $\theta > 0$ for which the moment generating function (MGF) of $|\varepsilon_t|^\theta$ is defined within an interval around zero. This sub-Weibull distribution family is very general (Zhang & Chen, 2021; Zhang & Wei, 2022): for example, when $\theta = 1$ and $\theta = 2$, it reduces to the sub-exponential or sub-Gaussian family, respectively. Mathematically, we denote $\varepsilon_t \sim \text{subW}(\theta; C)$ if the noise $\varepsilon_t$ satisfies $\mathbb{E} \exp(|\varepsilon_t|^\theta / C^\theta) \leq 2$, with $\theta > 0$ and $C > 0$ representing the *tail* and *size* parameter (Rinne, 2008; Vladimirova et al., 2020). In the design and analysis of bandit algorithms, it is commonly assumed that the parameters $\theta$ and $C$ are known (Wu et al., 2016; Lattimore & Szepesvári, 2020; Wu et al., 2022; Zhou et al., 2025).

We first note that the ZI structure retains the sub-Weibull tail behavior of the non-zero component $X_t - \mu$, as established in Lemma 2.1.

**Lemma 2.1.** *Assuming independent $Y_t \sim \text{Bernoulli}(p)$ and $X_t - \mu \sim \text{subW}(\theta; C)$, let $R_t = X_t \times Y_t$. Then, there exists a constant $C_R > 0$ such that $R_t - \mu p \sim \text{subW}(\theta; C_R)$.*

**Naive approaches and their limitations.** With Lemma 2.1, once the size parameter for $R_t$ is known (or estimated), we can construct an upper confidence bound for $r = \mu p$ using existing concentration inequalities for sub-Weibull variables $\{R_t\}$ (Zhang & Chen, 2021; Zhang & Wei, 2022). The corresponding UCB algorithms then follows, which we refer to as *naive approaches*. While such approaches can theoretically attain minimax regret rates under appropriate parameter specifications, the zero-inflated structure introduces unique challenges that lead to two clear limitations.

First, even when the true size parameter is known, without leveraging the zero-inflated structure, such an approach leads to a loose concentration bound and hence under-exploration. This can be seen in Figure 1b, our numerical study in terms of regret, and also our regret bound (e.g. Lemma 6.1). We can appreciate the intuition from the following fact: $\text{var}(R_t) = \mathbb{E}_Y(\text{var}(R_t|Y_t)) + \text{var}_Y(\mathbb{E}(R_t|Y_t)) = p \, \text{var}(X_t) + \mu^2 p(1 - p)$. Therefore, if we directly apply a concentration bound with $R_t$, the width of the bound will roughly increase linearly with $\mu$. However, the noise level and the difficulty of estimating either $p$ or $\mu$ (and hence $r$) should not change only by shifting the

non-zero distribution.

Second, in practice, estimating a valid size parameter $C_R$ for $r$ (hence having a valid upper confidence bound) is challenging, as the size parameter has complex dependency on $\mu$, $p$, $\theta$ and $C$. In real applications, there are a few common methods to choose $C_R$: (1) Use the size parameter of $X_t - \mu$; (2) use the variance of $R_t$ (estimated on the fly); (3) Use the definition to calculate $C_R$, with $\mu$ and $p$ estimated on the fly. The first two approaches are not valid, while the third one is also not reliable due to the sensitiveness induced by the ZI structure. We illustrate the issues with these methods in Lemma E.1 (in Appendix E) using sub-Gaussian distributions, which we summarize here: 1) $C_R$ can significantly exceed $C$, hence approach 1 is not valid; 2) $C_R$ can significantly exceed $\text{var}(R_t)$, hence approach 2 is not valid; 3) $C_R$ is very sensitive to $(p, \mu, \sigma^2)$, and hence prone to be heavily influenced by their estimation errors - specifically, the partial derivatives of $C_R$ with respect to these parameters can be arbitrary large within some regions.

## 2.1. Proposed product method and upper confidence bound approach

To address the shortcomings of the naive approaches, we introduce the *product method*, which leverages the product structure of the true reward $R_t = X_t \times Y_t$. Utilizing the corresponding concentration inequalities, we establish valid upper confidence bounds for $\mu$ using $\{X_t\}_{t=1}^n$ and for $p$ using $\{Y_t\}_{t=1}^n$, formulated as $\mathbb{P}(\mu > \overline{X} + U_X) \leq \alpha/2$ and $\mathbb{P}(p > \overline{Y} + U_Y) \leq \alpha/2$. Here $U_X$ and $U_Y$ are known functions, and $\overline{X}$ and $\overline{Y}$ are the sample averages for $\{X_t\}_{t=1}^n$ and $\{Y_t\}_{t=1}^n$, respectively. Consequently,

$$
\begin{aligned}
&\mathbb{P}\big(\mu p > (\overline{X} + U_X)(\overline{Y} + U_Y)\big) \\
&\leq \mathbb{P}(\mu - \overline{X} > U_X) + \mathbb{P}(p - \overline{Y} > U_Y) = \alpha,
\end{aligned}
\tag{1}
$$

which suggests $(\overline{X} + U_X)(\overline{Y} + U_Y)$ is a valid upper confidence bound for $r = \mu \times p$.

Now, the key of establishing our method lies in determining sharp concentration bounds for both $p$ and $\mu$. For $p$, a standard concentration bound for Bernoulli variables can be applied, such as: $U_Y = \sqrt{1/(2n) \times \log(2/\alpha)}$. However, establishing sharp bounds for $\mu$ presents significant challenges: (i) $\overline{X}$ is not directly observable since we only observe $X_t$ when $Y_t = 1$, and (ii) the number of non-zero observations is random, complicating standard concentration analysis. We therefore define the average value of the observed $X_t$ as: $\overline{X}^* := \frac{1}{\#\{t:Y_t=1\}} \sum_{t:Y_t=1} X_t$. Let $B = \sum_{t=1}^n Y_t$ represent the count of $X_t$ observations. While $B$ is a random variable, given any fixed $B$, the sub-Weibull concentration inequality still holds. For example, if $X_t$ is sub-Gaussian (i.e., $\theta = 2$)

with a variance proxy $\sigma^2$,

$$
\begin{aligned}
&\mathbb{P}\big(\mu - \overline{X}^* > \sqrt{2\sigma^2/B \log(2/\alpha)}\big) \\
&= \mathbb{E}\big[\mathbb{P}\big(\mu - \overline{X}^* > \sqrt{(2\sigma^2/B) \log(2/\alpha)} \mid B\big)\big] \leq \alpha/2.
\end{aligned}
$$

With $U_X = \sqrt{(2\sigma^2/B) \log(2/\alpha)}$, based on (1), we can derive a valid upper confidence bound for $r_k$, and develop the corresponding UCB algorithm. The algorithm details are outlined in Algorithm 1.

---

**Algorithm 1** UCB for ZI MAB with light tails

**Data:** Horizon $T$, tail parameter $\theta$, and size parameter $C$.

1  Set $U_k^\mu = 1$ and $U_k^p = 1, \forall k \in [K]$.
2  Set the counters $c_k = 0$, and set $\widehat{\mu}_k = 0$ and $\widehat{p}_k = 0$.
3  **for** $t = 1, \ldots, K$ **do**
4       Take action $A_t = t$;
5       Observe $R_t$, set $X_t = \mathbb{1}(R_t \neq 0) \times R_t$ and $Y_t = \mathbb{1}(R_t \neq 0)$.
6  **end**
7  **for** $t = K + 1, \ldots, T$ **do**
8       Take action $A_t = \arg\max_{k \in [K]} U_k^\mu \times U_k^p$ (break tie randomly);
9       Observe $R_t$, set $Y_t = \mathbb{1}(R_t \neq 0)$;
10      Update $c_{A_t} = c_{A_t} + 1$, $\widehat{p}_{A_t} = \widehat{p}_{A_t} + \frac{Y_t - \widehat{p}_{A_t}}{c_{A_t}}$, and $U_{A_t}^p = \widehat{p}_{A_t} + \sqrt{\frac{\log(2/\delta)}{2c_{A_t}}}$;
11      **if** $R_t \neq 0$ **then**
12          Set $X_t = R_t$;
13          Update $\widehat{\mu}_{A_t} = \frac{1}{\#\{l \leq t: A_l = A_t \text{ and } R_{A_l} \neq 0\}} \sum_{l \leq t: A_l = A_t} R_{A_l}$ and $U_{A_t}^\mu = \widehat{\mu}_{A_t} + 2\mathrm{e}D(\theta)C\Big(\sqrt{\frac{\log(4/\delta)}{n\widehat{p}_{A_t}/2}} + E(\theta)\frac{\log^{(1/\theta)\vee 1}(4/\delta)}{n\widehat{p}_{A_t}/2}\Big)$, where $D(\theta)$ and $E(\theta)$ are defined in Lemma 2.2.
14 **end**

---

While the above construction provides a valid algorithm, the theoretical analysis presents additional complexities. The concentration bound $\sqrt{(2\sigma^2/B) \log(2/\alpha)}$ is *random* and depends on $\{Y_t\}_{t=1}^n$, which significantly complicates regret analysis. Fortunately, Lemma 2.2 demonstrates that we can have a similar concentration bound with rate $\sqrt{np_k}$, which is much easier to analyze. It also verifies that the observed average of i.i.d. sub-Weibull variables behave with a combination of a Gaussian and a Weibull tail.

**Lemma 2.2.** *Suppose $X_t - \mu \overset{i.i.d.}{\sim} \text{subW}(\theta; C)$ and $Y_t \overset{i.i.d.}{\sim}$ Bernoulli$(p)$. Let $\overline{X}^* = \frac{1}{\#\{t \in [n]: R_t \neq 0\}} \sum_{t=1}^n R_t$ be the observed sample mean for $X$ and*

$$
U_{X^*} = 2\mathrm{e}D(\theta)C \left( \sqrt{\frac{2\log(4/\delta)}{pn}} + \frac{2E(\theta)\log^{(1/\theta)\vee 1}(4/\delta)}{pn} \right),
$$

*then $\mathbb{P}\big\{ |\mu - \overline{X}^*| \geq U_{X^*} \big\} \leq \delta$ for any $\delta > 0$ and $n \geq$*

$4\log(2/\delta)/p^2$. *The constants $D(\theta)$ and $E(\theta)$ are defined in Lemma F.2.*

More importantly, $U_{X^*}$ is not only independent of $\mu$ but also enables a tighter product method-based concentration for $r = \mu \times p$ compared to standard sub-Weibull concentrations for $R_t = X_t \times Y_t$. Our numerical results in Figure 1b demonstrate this advantage.

In addition, some applications exhibit zero-inflated rewards where the non-zero part is heavy-tailed, possessing only finite moments of order $1 + \epsilon$ for some $\epsilon \in (0, 1]$. To handle such cases, we construct an upper bound for the trimmed mean (Bickel, 1965; Bubeck et al., 2013), enabling the corresponding UCB algorithm. Further details are provided in Appendix B.

### 2.2. Thompson sampling approach

We also extend our discussion to another widely adopted approach, Thompson Sampling (TS, Thompson, 1933). Similarly to our approach with the UCB algorithms above, we consider the non-zero part $X_t$ and the binary variable $Y_t$ separately. For ease of exposition, we consider the sub-Gaussian case for $X_t$. This can be easily extended to sub-Weibull cases by introducing an extra sampling step, known as *Chambers-Mallows-Stuck (CMS)* Generation, to rescale $X_t$ to a sub-Gaussian tail (Weron, 1996; Dubey & Pentland, 2019; Shi et al., 2023).

Following the standard TS framework, we sample $\mu_k$ and $p_k$ from their respective posteriors and multiply them. We prove this is equal to posterior sampling in Appendix C.1. To achieve a minimax optimal TS algorithm for general sub-Gaussian distributions, we follow Jin et al. (2021); Karbasi et al. (2021) to use a clipped Gaussian distribution, denoted as $\mathrm{cl}\mathcal{N}(\mu, \sigma^2; \vartheta) := \max\{\mathcal{N}(\mu, \sigma^2), \vartheta\}$, which curtails overestimation of suboptimal arms and ensures optimality. We adapt this idea to use the clipped Gaussian distribution for sampling the non-zero part $X_t$ and a clipped Beta distribution for the binary part $Y_t$. Our decision to use a clipped Beta distribution over a standard Beta distribution stems from our analysis, which shows it is critical for managing the risk of overestimating suboptimal arms in $R_t = X_t \times Y_t$. Further details are outlined in Appendix C.1.

## 3. Zero-Inflated Contextual Bandits

In this section, we extend our discussion to the Contextual Bandits problem. For concreteness, we consider the following setup of contextual bandits, although other setups can be similarly formulated and addressed: on each round $t$, the agent observes a context vector $\mathbf{x}_t$ and a set of feasible actions $\mathcal{A}_t$, choose an action $A_t \in \mathcal{A}_t$, and receive a random reward $R_t = r(\mathbf{x}_t, A_t) + \varepsilon_t$, where $r$ is the mean-reward function and $\varepsilon_t$ is the random er-

ror. The cumulative regret in this setup is defined as $\mathcal{R}(T) = \sum_{t=1}^{T} \mathbb{E}\big[\max_{a \in \mathcal{A}_t} r(\mathbf{x}_t, a) - r(\mathbf{x}_t, A_t)\big]$.

To utilize the ZI structure, we propose to consider the following model: $R_t = 0 \times (1 - Y_t) + X_t \times Y_t$ with

$$X_t = g(\mathbf{x}_t, A_t; \boldsymbol{\beta}) + \varepsilon_t,$$
$$Y_t \sim \mathrm{Bernoulli}\big(h(\mathbf{x}_t, A_t; \boldsymbol{\theta})\big),$$

where $\varepsilon_t$ is a mean-zero error term, $h$ is a function with codomain $[0, 1]$ and parameterized by $\boldsymbol{\theta} \in \mathbb{R}^q$, and $g$ is a function parameterized by $\boldsymbol{\beta} \in \mathbb{R}^d$. We remark the relationship that $r(\mathbf{x}, a) = h(\mathbf{x}, a; \boldsymbol{\theta}) \times g(\mathbf{x}, a; \boldsymbol{\beta})$.

Here, we present a general UCB template for our method in Algorithm 2, which extends the MAB version in Section 2.2. Specifically, based on the functional forms of $h(\cdot, \cdot; \boldsymbol{\theta})$ and $g(\cdot, \cdot; \boldsymbol{\beta})$, we construct confidence bounds for exploration, denoted as $U_{\mathrm{all},t}(\mathbf{x}, a)$ for $h$ and $U_{\mathrm{all},t}(\mathbf{x}, a)$ for $g$, respectively. At each step, we estimate $\boldsymbol{\theta}$ and $\boldsymbol{\beta}$, then structure the UCB algorithm by maximizing the UCB score for each action. Similarly, the TS algorithm follows by designing appropriate sampling rules with suitable priors for $\boldsymbol{\theta}$ and $\boldsymbol{\beta}$. As a concrete example, Appendix C.2 provides detailed update formulas for both the UCB and TS algorithms when $h$ and $g$ are modeled as generalized linear functions.

---

**Algorithm 2** General template of UCB for ZI contextual bandits

**Data:** Confidence bound exploration terms $U_{\mathrm{all},t}(\cdot, \cdot)$ and $U_{\mathrm{non-zero},t}(\cdot, \cdot)$, random selection period $\tau$, and other algorithm-specific parameters.

15   Set $\mathcal{H}_{\mathrm{all}} = \{\}$ and $\mathcal{H}_{\mathrm{non-zero}} = \{\}$.

16   Randomly choose action $a_t \in \mathcal{A}_t$ for $t \in [\tau]$;

17   **for** $t = \tau + 1, \ldots, T$ **do**

18     Estimate $\widehat{\boldsymbol{\theta}}_t$ from the binary outcomes, using $\mathcal{H}_{\mathrm{all}}$;

19     Estimate $\widehat{\boldsymbol{\beta}}_t$ from the non-zero outcomes, using $\mathcal{H}_{\mathrm{non-zero}}$;

20     Take action $A_t = \arg\max_{a \in \mathcal{A}_t} \big[h(\mathbf{x}_t, a; \widehat{\boldsymbol{\theta}}_t) + U_{\mathrm{all},t}(\mathbf{x}_t, a)\big] \times \big[g(\mathbf{x}_t, a; \widehat{\boldsymbol{\beta}}_t) + U_{\mathrm{non-zero},t}(\mathbf{x}_t, a)\big]$.

21     Observe reward $R_t$, set $X_t = \mathbb{1}(R_t \neq 0) \times R_t$ and $Y_t = \mathbb{1}(R_t \neq 0)$.

22     Update the dataset as $\mathcal{H}_{\mathrm{all}} \leftarrow \mathcal{H}_{\mathrm{all}} \cup \{(\mathbf{x}_t, A_t, Y_t)\}$.

23     **if** $R_t \neq 0$ **then**

24       Update the dataset as $\mathcal{H}_{\mathrm{non-zero}} \leftarrow \mathcal{H}_{\mathrm{non-zero}} \cup \{(\mathbf{x}_t, A_t, R_t)\}$;

25   **end**

---

## 4. Theory

### 4.1. Regret bounds for ZI MAB

Although concentration bounds for both components (e.g. van de Geer & Lederer, 2013; Kuchibhotla & Chakrabortty,

2022; Zhang & Wei, 2022, for sub-Weibull) and $Y$ (e.g. Bentkus, 2004; Mattner & Roos, 2007; Zhang & Chen, 2021, for Bernoulli) have been extensively studied, there present non-trivial challenges to analyze our algorithm with the product of them. These challenges, as discussed above, arise because the reward, and consequently the action selection, is jointly determined by both $X$ and $Y$. Unlike standard bandits where the observability of $X$ depends solely on arm selection, here it also depends on $Y$'s value. Consequently, the number of times $X$ is observed becomes a random variable. The lack of a precise distribution for sub-Weibull $X$ also complicates analytical analysis. Fortunately, our Lemmas 2.2 and B.1 help address these challenges to support the regret analysis of our algorithms.

Without loss of generality, we assume $r_k \in (0, 1)$, and $r_1 = \max_{k \in [K]} r_k$, i.e., the first arm is the optimal arm. To demonstrate the prior properties of considering the ZI structure from a theoretical perspective, we present the regret bound for our light-tailed ZI UCB algorithm for MAB.

**Theorem 4.1.** *Consider a $K$-armed zero-inflated bandit with sub-Weibull noise* $\mathrm{subW}(\theta; C)$. *We have an upper bound for the problem-dependent regret of Algorithm 1 with* $\delta = 4/T^2$ *as* $\mathcal{R}(T) \lesssim$

$$\sum_{k=2}^{K} p_k^{-2} \log T / \Delta_k + \sum_{k=2}^{K} p_k^{-1} \log^{(1/\theta) \vee 1} T + p_1^{-2} \log T \sum_{k=2}^{K} \Delta_k,$$

*where* $\Delta_k = r_1 - r_k$ *for* $k = 2, \ldots, K$.

In Theorem 4.1, the notation "$\lesssim$" indicates inequality up to constants independent of bandit-specific parameters. Since $p_k$ is bounded in $(0, 1]$ and $\Delta_k \leq 1$, the problem-independent regret simplifies to $\mathcal{R}(T) \lesssim \sqrt{KT \log T} + K$. This matches the minimax lower bound up to a factor of $\mathcal{O}(\sqrt{\log T})$ as given in Theorem 15.2 of Lattimore & Szepesvári (2020).

Similarly, we establish both problem-dependent and problem-independent regret bounds for our heavy-tailed UCB algorithm (Algorithm B.1 in Appendix C.1), which matches the sharpness upper bounds rates in the current literature (Bubeck et al., 2013; Dubey et al., 2020; Chatterjee & Sen, 2021) and hence achieves state-of-the-art performance. The detailed regret analysis is provided in Appendix B.

We also provide the regret analysis for our TS algorithm, Algorithm C.1 in Appendix C.1, when the non-zero part is sub-Gaussian. In contrast to the proofs of UCB algorithms, we require the anti-concentration properties of the distributions to control the probability of underestimating the optimal arm (Agrawal & Goyal, 2013; Jin et al., 2021; 2022). Fortunately, we prove that the clipped Gaussian and clipped Beta distributions, as well as their products, exhibit anti-concentration with optimal decay rates (Lemma F.3 and Lemma F.4). This finding allows us to establish the problem-dependent regret bound for our TS algorithm.

**Theorem 4.2.** *Consider a $K$-armed ZIB with sub-Gaussian rewards. Let the tuning parameters satisfy $\gamma \geq 4$, $\rho \in (1/2, 1)$, and $\alpha_k, \beta_k, v_k \in [0, 1]$. Then Algorithm C.1 has*

$$\mathcal{R}(T) \lesssim \sqrt{KT} + \sum_{k=2}^{K} \left( p_k^{-1} \Delta_k + p_1^{-1} \sqrt{T/(p_k K)} \right).$$

Given that $p_k$ are bounded within $(0, 1]$, the problem-independent regret is $\mathcal{R}(T) \lesssim \sqrt{KT} + K$. Compared to UCB (Theorem 4.1), Algorithm C.1 improves by a factor of $\sqrt{\log T}$. The regret is both minimax optimal (Auer et al., 2002) for sub-Gaussian MAB. One can extend Theorem 4.2 for sub-Weibull rewards with $\theta < 2$ with the CMS generation (Dubey & Pentland, 2019). This introduces an additional regret term scaling as $(KT)^{1/\theta}$ (Dubey & Pentland, 2019; Shi et al., 2023), which remains state-of-the-art.

### 4.2. Regret bounds for ZI contextual bandits

Analyzing ZI contextual bandit algorithms requires constructing confidence regions for both $\boldsymbol{\beta}$ and $\boldsymbol{\theta}$. Unlike in standard contextual linear bandit literature, the ZI structure means that updates to $\boldsymbol{\beta}$ only occur when non-zero outcomes are observed. Fortunately, the concentration analysis we established for MAB can be similarly applied here. To illustrate the theoretical advantage of our ZI contextual bandit algorithms, we analyze an instantiation where both the non-zero part $X_t$ and the binary part $Y_t$ follow generalized linear models (GLMs):

$$X_t = g\big(\boldsymbol{\beta}^\top \psi_X(\mathbf{x}_t, A_t)\big) + \varepsilon_t$$
$$\text{and} \quad Y_t \sim \text{Bernoulli}\big\{ h\big(\boldsymbol{\theta}^\top \psi_Y(\mathbf{x}_t, A_t)\big)\big\}.$$

Here, $\varepsilon_t$ is sub-Gaussian, $g(\cdot)$ and $h(\cdot)$ are strictly increasing link functions, $\psi_X(\cdot)$ and $\psi_Y$ are known transformations. The unknown parameter vectors $\boldsymbol{\beta} \in \mathbb{R}^d$ and $\boldsymbol{\theta} \in \mathbb{R}^q$ govern the models, and the action space $\mathcal{A} = [K] \subseteq \mathbb{N}$ can be large or even infinite. This GLM setting is widely studied in bandit literature (Filippi et al., 2010; Li et al., 2017; Wu et al., 2022; Li et al., 2010; 2012; Lattimore & Szepesvári, 2020).

Denote $\|\mathbf{z}\|_{\mathbf{A}}^2 = \mathbf{z}^\top \mathbf{A} \mathbf{z}$ for any $\mathbf{z} \in \mathbb{R}^d$ and positive definite matrix $\mathbf{A} \in \mathbb{R}^{d \times d}$. For the GLM setting, we design explicit forms for the confidence bound exploration terms $U_{\text{all}, t}$ and $U_{\text{non-zero}, t}$, along with the estimation steps in Algorithm 2. Specifically, at round $t$, the estimates $\widehat{\boldsymbol{\beta}}_t$ and $\widehat{\boldsymbol{\theta}}_t$ are obtained via ridge regression

$$\widehat{\boldsymbol{\beta}}_t := \arg \min_{\boldsymbol{\beta} \in \Gamma} \left\| \sum_{s \in [t]: Y_s = 1} \left[ R_s - g\big( \psi_X(\mathbf{x}_s, A_s)^\top \boldsymbol{\beta}\big) \right] \right\|_{\mathbf{V}_t^{-1}}$$

$$\text{and } \widehat{\boldsymbol{\theta}}_t := \arg \min_{\boldsymbol{\theta} \in \Theta} \left\| \sum_{s=1}^{t} \left[ Y_s - h\big( \psi_Y(\mathbf{x}_s, A_s)^\top \boldsymbol{\theta}\big) \right] \right\|_{\mathbf{U}_t^{-1}}.$$

The corresponding sample covariance matrices are $\mathbf{V}_t = \lambda_V \mathbf{I}_d + \sum_{s \in [t]: Y_s = 1} \psi_X(\mathbf{x}_s, A_s) \psi_X(\mathbf{x}_s, A_s)^\top$ and $\mathbf{U}_t = \lambda_U \mathbf{I}_q + \sum_{s \in [t]} \psi_Y(\mathbf{x}_s, A_s) \psi_Y(\mathbf{x}_s, A_s)^\top$, where $\lambda_V, \lambda_U$ are regularization parameters, and $\Gamma, \Theta$ are compact parameter spaces. To balance exploitation and exploration, we select actions that maximize the sum of the estimated mean and variance, which can be interpreted as an upper confidence bound. The confidence bound exploration terms are designed as $U_{\text{non-zero},t}(\mathbf{x}_t, a) = \rho_{X,t} \|\psi_X(\mathbf{x}_t, a)\|_{\mathbf{V}_t^{-1}}$ and $U_{\text{all},t}(\mathbf{x}_t, a) = \rho_{Y,t} \|\psi_Y(\mathbf{x}_t, a)\|_{\mathbf{U}_t^{-1}}$, where $\{\rho_{X,t}, \rho_{Y,t}\}_{t \geq 0}$ are ellipsoidal scaling factor sequences controlling exploration. This leads to our UCB algorithm for ZI GLM (Algorithm C.2 in Appendix C).

To derive the regret bounds for our UCB algorithm, we impose regularity conditions on the link functions, parameter spaces, and underlying distributions. Informally, we assume that the link functions have bounded derivatives, the parameter spaces are compact, the variance matrices for both the non-zero and binary components are positive definite with bounded minimal eigenvalues, and the noise is sub-Gaussian. These assumptions are standard and mild in the literature on generalized linear contextual bandits (e.g., Li et al., 2010; Deshpande et al., 2018; Wu et al., 2022). The formal assumptions are detailed in Assumptions C.1 and C.2 in Appendix C.2. Utilizing results on self-normalized martingales (Abbasi-Yadkori et al., 2011), we derive the following regret bounds for Algorithm C.2.

**Theorem 4.3.** *Fix any $\delta > 0$. Suppose Algorithm C.2 is run with a suitable random selection period $\tau$ and tuning parameters $\{\rho_{X,t}, \rho_{Y,t}\}_{t \geq 0}$, as specified in Assumption C.3 in Appendix C.2. Under the regularity conditions in Assumptions C.1 and C.2, the regret is bounded with probability at least $1 - 5\delta$ as*

$$\mathcal{R}(T) \lesssim \tau + \sqrt{dT \log(1 + d^{-1}T)\big[\log(1/\delta) + d\log(1 + d^{-1}T)\big]}$$
$$+ \sqrt{qT \log(1 + q^{-1}T)\big[\log(1/\delta) + q\log(1 + q^{-1}T)\big]}$$

*for any $\lambda_U, \lambda_V > 0$.*

A notable observation is that the regularity conditions, random selection period, tuning parameter choices, and regret bound in Theorem 4.3 are independent of the number of arms $K$. This regret rate $\widetilde{\mathcal{O}}((d \vee q)\sqrt{T})$ matches the minimax lower bound up to a logarithmic factor for contextual bandit problems with both finite and countably infinite action spaces (Theorem 3 in Dani et al., 2008).

While Theorem 4.3 focuses on the UCB approach, the TS variant can achieve the same regret rate. Specifically, the TS algorithm samples parameters as

$$\widetilde{\boldsymbol{\beta}}_t \sim \mathcal{N}(\widehat{\boldsymbol{\beta}}_t, \varrho_{X,t}^2 \mathbf{V}_t^{-1}) \quad \text{and} \quad \widetilde{\boldsymbol{\theta}}_t \sim \mathcal{N}(\widehat{\boldsymbol{\theta}}_t, \varrho_{Y,t}^2 \mathbf{U}_t^{-1}),$$

using the same estimators $\widehat{\boldsymbol{\beta}}_t, \widehat{\boldsymbol{\theta}}_t, \mathbf{V}_t, \mathbf{U}_t$ as in the UCB algorithm. With appropriately chosen confidence radii $\{\varrho_{X,t}, \varrho_{Y,t}\}_{t \geq 0}$, the TS algorithm selects actions as $A_t^{\text{TS}} = \arg\max_{a \in [K]} [\psi_X(\mathbf{x}_t, a)^\top \widetilde{\boldsymbol{\beta}}_t] \times [\psi_Y(\mathbf{x}_t, a)^\top \widetilde{\boldsymbol{\theta}}_t]$. The corresponding regret bounds are provided in the following corollary.

**Corollary 4.4.** *Suppose the conditions in Theorem 4.3 hold. If Algorithm C.3 is run with the same random selection period $\tau$ as Algorithm C.2 and the tuning parameters $\varrho_{X,t}, \varrho_{Y,t}$ specified in (C.3), then the regret is bounded with probability at least $1 - 5\delta$ by $\widetilde{\mathcal{O}}((d \vee q)^2 \sqrt{T})$.*

## 5. Experiment

**Simulation with MAB.** For MAB problems, we evaluate our algorithms across three reward distributions: Gaussian, Gaussian mixture, and exponential. We set up several UCB baselines using the sub-Weibull concentration (Bogucki, 2015; Hao et al., 2019), with difference in the size parameter specification: size parameter of the non-zero component, estimated variances, on-the-fly estimated size parameters, and the true size parameters (strong baseline). Additionally, we include the MOTS algorithm from Jin et al. (2021) as a TS baseline. The details of these baselines and our setting are presented in Appendix D.1.

For the reward distributions, we consider: (i) Gaussian rewards with unit standard deviation and means $\mu_k$; (ii) Mixed Gaussian rewards drawn from $(1 - p_k) \times \mathcal{N}\left(\frac{\mu_k}{2(1-p_k)}, \sigma^2\right) + p_k \times \mathcal{N}\left(\frac{\mu_k}{2p_k}, \sigma^2\right)$ ensuring overall mean $\mu_k$; and (iii) Exponential rewards with mean $\mu_k$. All mean parameters $\mu_k$ are independently drawn from $U(0, 100)$. Throughout our experimental evaluation, all plotted lines represent mean cumulative regret over multiple independent replications. Error bars around the curves indicate $\pm 1/10$ standard deviation, chosen to ensure visual clarity while reflecting variability. In some cases, error bars may appear negligible or invisible due to very low variability across replications.

We run simulations with different distributions of $p = \max_{k \in [K]} p_k$ with $\delta = 4/T^2$. Here, we only present the result with $p \sim U[0.30, 0.35]$ in Figure 2. Besides, the $1 - \delta$ upper confidence bounds for different UCB algorithms are shown in Figure 1 (b). Findings from other settings (including high and very low values of $p$, and bounded rewards) are similar and detailed in Appendix D.1. Our algorithms demonstrated sub-linear regrets across various distributions. In contrast, except for the strong baseline, all other methods exhibited either linear or significantly higher regrets in some cases, due to the challenge to correctly quantifying uncertainty in ZIB. Consistent with our observations in Figure 1b, our UCB algorithm even exhibits a much lower regret than the strong baseline in both Gaussian and Gaussian-mixture cases. This proves our motivation that ignoring the ZI structure leads to looser concentration bounds and hence

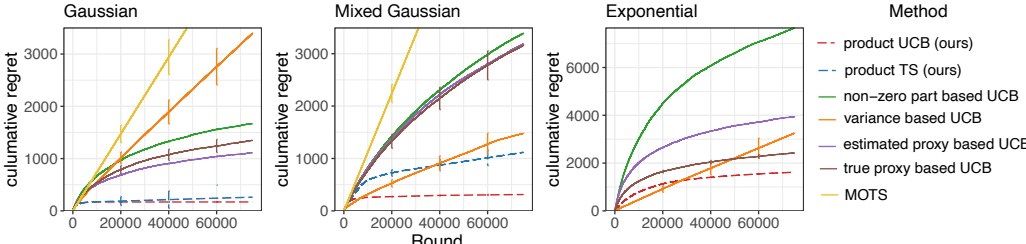

*Figure 2.* Zero-inflated MAB with $K = 10$ and $T = 75000$ with $N = 50$ replications for $p \sim U[0.30, 0.35]$.

higher regret. Similarly, unlike the baseline TS algorithm (Jin et al., 2021), our TS algorithm outperforms other baseline algorithms. We present results for TS algorithms only in Gaussian and Gaussian-mixture cases. For exponential distributions, the required *CMS transformation* to rescale the distribution to sub-Gaussian tails—discussed in Section 2.2—complicates fair comparisons.

**Simulation with contextual bandits.** We adapt the settings in Bastani et al. (2021); Wu et al. (2022) to ZIB. We design $K = 100$ arms and generate a vector $\boldsymbol{\nu}_k \in \mathbb{R}^d$ with $d = 10$ for each arm $k \in [K]$. At each round $t$ the context for arm $k$ is sampled as $\mathbf{x}_{k,t} \sim \mathcal{N}_d\left(\boldsymbol{\nu}_k, \frac{1}{2K} I_d\right)$. The non-zero part of the reward, $X_t$, is generated using a linear model: $\boldsymbol{\beta}^\top \mathbf{x}_{A_t,t} + \varepsilon_t$, with $\varepsilon_t$ representing white noise. For the binary part, we model the probability using a generalized linear function with the link function $h(\boldsymbol{\theta}^\top \sin(\mathbf{x}_{A_t,t}))$. Both $\boldsymbol{\beta}$ and $\boldsymbol{\theta}$ are $d$-dimensional, with $\boldsymbol{\theta}$ having $s$ non-zero elements, where $s \in [d]$. The sparsity level $s$ signifies that the non-zero elements in $\boldsymbol{\theta}$ influence the occurrence of non-zero rewards. Unlike the MAB problem, the contextual bandits problem introduces an additional layer of complexity for fair comparison due to model specification. We evaluate Algorithm C.2 against two baseline UCB methods: The first one is a naive method, called misspecified UCB, that overlooks the ZI structure, treating it as a linear bandit and thus misspecifying the model. The second, termed integrated UCB, aligns with the UCB baselines in MAB: it correctly specifies the model, but directly quantifies the uncertainty of $(\boldsymbol{\beta}^\top, \boldsymbol{\theta}^\top)$ from the model $\boldsymbol{\beta}^\top \mathbf{x}_t h(\boldsymbol{\theta}^\top \sin(\mathbf{x}_t)) + \varepsilon_t$, by utilizing generalized contextual linear bandit algorithms (Li et al., 2017). Similarly, we assess our TS algorithm, Algorithm C.3, against the misspecified TS, which disregards the ZI structure, and integrated TS, which mirrors the integrated UCB. Various sparsity levels and link functions $h(\cdot)$ are considered. Results for $s = 7$ using the Probit function are shown in Figure 3, with additional simulation details provided in Appendix D.2. Again, TS algorithms are excluded for exponential noise to ensure fair comparison. As shown in Figure 3, our UCB and TS algorithms consistently achieve lower sub-linear regrets across all tested distributions. In contrast, other methods, except for integrated TS, exhibit linear regrets. While integrated TS also achieve sub-linear regret, our product TS demonstrates superior performance.

**Real Data.** We apply our ZI contextual bandit algorithms to a real dataset of loan records from a U.S. online auto loan company, a widely studied public dataset (Columbia Business School, Columbia University, 2024; Phillips et al., 2015; Ban & Keskin, 2021; Bastani et al., 2022). We mainly follow the setup in Chen et al. (2023). At each round $t$, an applicant arrives with certain covariates (the context) $\mathbf{x}_t$. The company then proposes a loan interest rate, which can be transformed into the nominal profit $A_t$ for the company (the raw action). This raw action, along with $\mathbf{x}_t$, determines the actual profit $X_t$ and affects the applicant's decision to accept $Y_t \in \{0, 1\}$. The resulting reward for the company is hence $R_t = X_t \times Y_t$. We categorize the raw action $A_t \in \mathbb{R}$ into discrete levels: $\widetilde{A}_t \in \mathcal{A}_{\text{discrete}} = \{\text{"low", "medium", "high", "very high", "luxury"}\}$, each representing a distinct price level. To simulate counterfactual outcomes, we begin by fitting the entire dataset using

$$X_t = (A_t, 1, b(\mathbf{x}_t)^\top)^\top \boldsymbol{\beta} + \varepsilon_t$$
$$\text{and} \quad Y_t \sim \text{Bernoulli}\left(h\left((A_t, 1, b^\top(\mathbf{x}_t))^\top \boldsymbol{\theta}\right)\right),$$

where $h(\cdot)$ is the logistic function and the transformation function $b(\cdot)$ applied to the context is detailed in Appendix C.2.

After deriving initial estimators $\widehat{\boldsymbol{\beta}}$ and $\widehat{\boldsymbol{\theta}}$, we then compare our method with two baseline approaches discussed in simulations for contextual bandits at each round $t$, an action $\widetilde{A}_t$ is selected from $\mathcal{A}_{\text{discrete}}$ using our algorithm. The profit $A_t$ is then sampled from its truncated distribution based on $\widetilde{A}_t$. Subsequently, the real profit $X_t$ and the binary decision $Y_t$ based on the offered $A_t$, and the corresponding reward $R_t = X_t \times Y_t$ are calculated. With the estimated parameters, we determine the optimal actions for each round to compute the regret. Detailed procedures and tuning parameters are in Appendix C.2. The average cumulative regrets over 5000 rounds are shown in Figure 4. Our UCB and TS algorithms demonstrate significantly lower regret compared to the integrated UCB and TS algorithms, highlighting the importance of leveraging the ZI structure in real-world contextual bandit problems. Furthermore, it is worth noting that while the misspecified UCB and TS algorithms may appear to perform well on this dataset, we should approach their performance with caution: As shown in Figure 3, these algorithms sometimes result in linear regrets. In contrast,

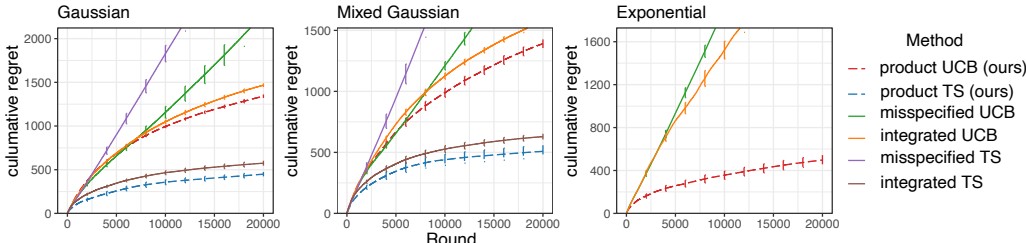

Figure 3. Zero-inflated contextual bandits with $T = 20000$ and $s = 7$ under $N = 25$ replications.

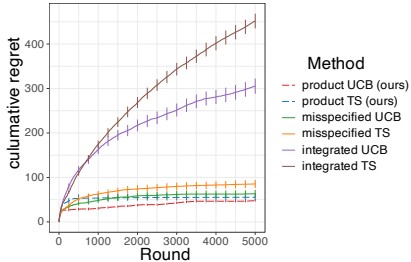

Figure 4. Results with the real dataset.

our UCB and TS algorithms consistently outperform the competition, achieving the lowest overall regret.

## 6. Discussion

In this paper, we introduce a new bandit model and the corresponding algorithmic framework tailored to the common ZI distribution structure. The primary advantage lies in lower regrets from more accurate uncertainty quantification by utilizing the ZI structure. We establish theoretical regret bounds for UCB and TS algorithms in MABs under various reward distributions, as well as for contextual linear bandits.

**Time and Space Complexity.** Computational efficiency is crucial for practical applications. An important advantage of our approach is that our ZI-based methods retain the same big-O time and space complexity as standard baselines for both MAB and GLM settings. The only difference is a small constant overhead from maintaining two estimators: one for the zero indicator $Y_t$ and one for the nonzero magnitude $X_t$, instead of a single reward estimator. Thus, our methods preserve the same computational order as existing approaches without adding extra computational cost.

**Asymptotic Order-Optimality.** A natural theoretical question is whether our algorithms achieve asymptotic order-optimality. The following lemma establishes a problem-dependent lower bound for ZIB with a Gaussian non-zero component.

**Lemma 6.1.** *Consider a $K$-armed zero-inflated bandit with the non-zero components belong to Gaussian distributions with variance $\sigma^2$. For any consistent algorithm (see Definition 16.1 in [Lattimore & Szepesvári, 2020](#)), the following*

*lower bound holds:*

$$\liminf_{T \to +\infty} \frac{\mathcal{R}(T)}{\log T} \geq \sum_{k=2}^{K} \left[ \frac{[0 \vee (\mu_k - p_k^{-1} r_1)]^2}{2\sigma^2} + \right.$$

$$\left. p_k \log \left( \frac{p_k}{p_k \wedge \mu_k^{-1} r_1} \right) + (1 - p_k) \log \left( \frac{1 - p_k}{1 - p_k \wedge \mu_k^{-1} r_1} \right) \right].$$

*Specifically, a necessary condition for an algorithm to achieve asymptotic order-optimality for sub-Gaussian ZI MABs is that its regret satisfies* $\liminf_{T \to +\infty} \mathcal{R}(T)/\log T \lesssim \sum_{k=2}^{K} p_k^{-1} \Delta_k^{-1}$.

We call an algorithm as *asymptotic order-optimal*, also known as *finite-time instance near-optimality* ([Lai & Robbins, 1985](#); [Lattimore, 2015](#); [Ménard & Garivier, 2017](#); [Lattimore & Szepesvári, 2020](#)), if its regret satisfies $\liminf_{T \to +\infty} \mathcal{R}(T)/\log T$ and achieves the lower bound in Lemma [6.1](#), up to universal constants independent of both the horizon $T$ and problem-specific parameters $r_k$ (and thus $\Delta_k, p_k$). While we do not explicitly prove that our UCB and TS algorithms for MABs achieve asymptotic order-optimality, they satisfy the necessary condition by accounting for the ZI structure parameter $p_k$. However, designing an algorithm that attains this optimality may require additional refinements. For example, in [(1)](#), we allocate equal probability $\alpha/2$ for the two concentration bounds, $\mathbb{P}(\mu > \overline{X} + U_X) \leq \alpha/2$ and $\mathbb{P}(p > \overline{Y} + U_Y) \leq \alpha/2$. To attain asymptotic order-optimality, these probability allocations should potentially be adapted based on the zero-inflation parameters $\{p_k\}_{k=1}^{K}$. We leave this as an open problem for future research.

**Broader Implications.** The ZI structure studied in this paper can be viewed as a special case of a broader class of problems where the reward distribution follows a hierarchical structure. This framework is particularly useful when the reward distribution is multimodal or when the reward generation mechanism follows a hierarchical process. For instance, in product recommendations, a customer's initial reaction (e.g., "highly interested", "somewhat interested", "not interested") could determine the subsequent reward distribution. Exploring these broader applications remains an interesting direction for future research.

## Impact Statement

This paper presents an improved algorithm framework for multi-armed bandits and contextual bandits, both are mature area with many applications. We are not aware of any particular negative social impacts of this work.

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

## A. Appendix Overview and Roadmap

We provide a roadmap to help readers navigate the supplementary material.

*Table 2.* Appendix overview. Click on section names to navigate directly to the content.

## B. Heavy-tailed MAB

In many applications with zero-inflated rewards, the non-zero part can be heavy-tailed (with only finite moments of order $1 + \epsilon$ for some $\epsilon \in (0, 1]$). To accommodate these scenarios, we adopt the trimmed mean (Bubeck et al., 2013) as a solution. The truncation follows a Hoeffding-type upper bound under the condition $|X_t|^{1+\epsilon} \leq M$. Specifically, for fully observable data $\{X_t\}_{t=1}^{n}$, we can construct a $1 - \delta$ upper confidence bound for $\mu$ with the trimmed mean

$$\overline{X}^{\text{trimmed}} := \frac{1}{n} \sum_{t=1}^{n} X_t \mathbb{1}\left( |X_t| \leq \left(\log^{-1}(\delta^{-1}) M t\right)^{1/(1+\epsilon)} \right).$$

The resulting upper confidence bound (Bickel, 1965; Bubeck et al., 2013; Dubey et al., 2020) is given by

$$\overline{X}^{\text{trimmed}} + 4M^{\frac{1}{1+\epsilon}} \left( n^{-1} \log \left(\delta^{-1}\right) \right)^{1/(1+\epsilon)}.$$

In ZIB, we similarly define

$$\overline{X}^{**} := \frac{1}{\#\{t \in [n] : Y_t = 1\}} \sum_{t:Y_t=1} X_t \mathbb{1}\left( |X_t| \leq \left(\log^{-1}(2/\delta) j(t) M\right)^{1/(1+\epsilon)} \right) \qquad \text{with} \qquad j(t) = \sum_{\ell \leq t} \mathbb{1}(Y_\ell = 1).$$

Then we can construct a valid upper bound for $\mu$ as $U_{X^{**}} = \overline{X}^{**} + M^{1/(1+\epsilon)} \left(32 \log S(n)/n\right)^{\epsilon/(1+\epsilon)}$, where $S(n)$ is the round index when the arm that we focus on has been pulled for $n$ times. The corresponding UCB algorithm is then constructed using $U_{X^{**}} \times U_Y$, with $U_Y$ as previously defined. This approach is further detailed in Algorithm B.1.

Similar to Lemma 2.2, Lemma B.1 below confirms that this upper bound maintains a deterministic rate, simplifying the analysis.

---

**Algorithm B.1** UCB for zero-inflated bandits with heavy tails

**Data:** Horizon $T$, parameters $\epsilon$ and $M$.

26 Set $U_k^\mu = 1$ and $U_k^p = 1, \forall k \in [K]$.

27 Set the counters $c_k = 0$, set the mean estimator $\widehat{p}_k = 0$ and $\widehat{\mu}_k = 0, \forall k \in [K]$.

28 **for** $t = 1, \ldots, K$ **do**

29      Take action $A_t = t$, set $X_t = \mathbb{1}(R_t \neq 0) \times R_t$ and $Y_t = \mathbb{1}(R_t \neq 0)$.

30 **end**

31 **for** $t = K+1, \ldots, T$ **do**

32      Take action $A_t = \arg\max_{k \in [K]} U_k^\mu \times U_k^p$ (break tie randomly);

33      Observe $R_t$, and set $Y_t = \mathbb{1}(R_t \neq 0)$.

34      Update $c_{A_t} = c_{A_t} + 1$, $\widehat{p}_{A_t} = \widehat{p}_{A_t} + \frac{Y_t - \widehat{p}_{A_t}}{c_{A_t}}$, and $U_k^p = \widehat{p}_{A_t} + \sqrt{\frac{2\log t^2}{c_{A_t}}}$;

35      **if** $R_t \neq 0$ **then**

36          Set $X_t = R_t$;

37          Update

$$\widehat{\mu}_{A_t} = \frac{1}{\#\{l \leq t : A_l = A_t \text{ and } R_{A_l} \neq 0\}} \sum_{l \leq t : A_l = A_t} R_{A_l} \mathbb{1}\left\{|R_{A_l}| \leq g(\widehat{p}_{A_l}, \epsilon) M^{\frac{1}{1+\epsilon}} \left(\frac{\log l^2}{c_{A_l}}\right)^{\frac{\epsilon}{1+\epsilon}}\right\}$$

         where the function $g(\cdot, \epsilon)$ is defined in Lemma B.1, and $U_k^\mu = \widehat{\mu}_{A_t} + M^{1/(1+\epsilon)}\left(32\log t/c_k(t)\right)^{\epsilon/(1+\epsilon)}$;

38 **end**

---

**Lemma B.1.** *Suppose* $Y_t \overset{i.i.d.}{\sim} \text{Bernoulli}(p)$ *and* $X_t$ *satisfy* $\mathbb{E}|X_t - \mu|^{1+\epsilon} \leq M$ *for some* $\epsilon \in (0, 1]$ *with positive* $M > 0$. *Then*

$$\mathbb{P}\left(\mu - \overline{X}^{**} \geq g(p, \epsilon) M^{\frac{1}{1+\epsilon}} \left(n^{-1}\log(2/\delta)\right)^{\frac{\epsilon}{1+\epsilon}}\right) \leq \delta$$

*for any* $\delta > 0$ *and* $n \geq 4\log(2/\delta)/p^2$, *where* $g(p, \epsilon) := p^{-\frac{\epsilon}{1+\epsilon}}(1+\epsilon)2^{\frac{\epsilon}{1+\epsilon}} + p^{-1}4/3 + 2/\sqrt{p}$.

Similarly, using Lemma B.1, we can establish the problem-dependent regret bound for our heavy-tailed UCB algorithm in Algorithm B.1.

**Theorem B.2.** *For a $K$-armed ZIB with noise satisfying* $\max_{k \in [K]} \mathbb{E}|\varepsilon_k|^{1+\epsilon} < \infty$ *for some* $\epsilon \in (0, 1]$, *Algorithm B.1 has a problem-dependent regret bound as*

$$\mathcal{R}(T) \leq 2\sum_{k=2}^{K}\left(3\Delta_k + 4p_1^{-2}\Delta_k + 9p_k^{-2}\Delta_k^{-1}\right) + \sum_{k=2}^{K}\left(2p_k g(p_k, \epsilon)\right)^{\frac{1+\epsilon}{\epsilon}} M^{1/\epsilon}\Delta_k^{-1/\epsilon}\log T.$$

When treating the number of arms as finite and assuming that $p_k$ are fixed, comparing our results with those in the heavy-tailed bandit literature (Bubeck et al., 2013; Dubey et al., 2020; Chatterjee & Sen, 2021) and we know that our algorithm achieves state-of-the-art regret rate. Moreover, from Theorem B.2, the problem-independent regret for Algorithm B.1 is given by

$$\mathcal{R}(T) \lesssim p_1^{-2}K + \sum_{k=2}^{K} p_k^{-1}\sqrt{T} + (MT)^{\frac{1}{1+\epsilon}}(K\log T)^{\frac{\epsilon}{1+\epsilon}} \tag{B.1}$$

$$\lesssim K\sqrt{T} + T^{\frac{1}{1+\epsilon}}(K\log T)^{\frac{\epsilon}{1+\epsilon}}$$

where the last "$\lesssim$" is due to $p_1, p_k \in (0, 1]$. This finalizes the minimax ratio in Table 1 for MAB.

## C. Additional Algorithms Details

### C.1. TS Algorithms for MAB

As outlined in Section 2, we have developed a light-tailed TS algorithm for MAB. The light-tailed TS-type algorithm achieves the minimax optimal rate when applied under the clipped distributions. The comprehensive procedure of the is presented in Algorithm C.1.

---

**Algorithm C.1** TS for zero-inflated MAB with light tails

---

**Data:** Prior parameters $\{\alpha_k, \beta_k, v_k\}_{k=1}^K$ and $\rho, \gamma$.

39 Set the counter $c_k = 0, \forall k \in [K]$;

40 **for** $t = 1, \ldots, K$ **do**

41 $\quad$ Take action $A_t = t$, set $X_t = \mathbb{1}(R_t \neq 0) \times R_t$ and $Y_t = \mathbb{1}(R_t \neq 0)$.

42 **end**

43 **for** $t = K+1, \ldots, T$ **do**

44 $\quad$ Sample $\widetilde{p}_k \sim \text{cl Beta}(\alpha_k, \beta_k; \vartheta_k(p))$ and $\widetilde{\mu}_k \sim \text{cl}\mathcal{N}\left(v_k, \frac{2\sigma^2}{\rho c_k \widehat{p}_k}; \vartheta_k(\mu)\right)$

45 $\quad$ Take action $A_t = \arg\max_{k \in \mathcal{A}} \widetilde{p}_k \times \widetilde{\mu}_k$;

46 $\quad$ Observe reward $R_t$, and set $Y_t = \mathbb{1}(R_t \neq 0)$.

47 $\quad$ Update $\alpha_{A_t} = \alpha_{A_t} + Y_t$ and $\beta_{A_t} = \beta_{A_t} + 1 - Y_t$;

48 $\quad$ Update $\widehat{p}_{A_t} = \widehat{p}_{A_t} + \frac{Y_t - \widehat{p}_{A_t}}{c_{A_t}}$ and $c_{A_t} = c_{A_t} + 1$;

49 $\quad$ Update

$$\vartheta_{A_t}(p) = \widehat{p}_{A_t} + \sqrt{\frac{\gamma}{4c_{A_t}} \log^+\left(\frac{T}{4c_{A_t}K}\right)}.$$

50 $\quad$ **if** $R_t \neq 0$ **then**

51 $\quad\quad$ Calculate:

$$\widehat{\mu}_{A_t} = \frac{1}{\#\{l \leq t : A_l = A_t \text{ and } R_{A_l} \neq 0\}} \sum_{l \leq t: A_l = A_t} R_{A_t};$$

52 $\quad\quad$ Update

$$v_{A_t} = \widehat{\mu}_{A_t}$$

$\quad\quad$ and

$$\vartheta_{A_t}(\mu) = \widehat{\mu}_{A_t} + \sqrt{\frac{4\gamma\left[1 + \log^{-1}(1 + 1/\sqrt{c_{A_t}T})\right]\sigma^2}{\widehat{p}_{A_t}^2 c_{A_t}}} \sqrt{\log^+\left(\frac{4\left[1 + \log^{-1}(1 + 1/\sqrt{c_{A_t}T})\right]\sigma^2 T}{\widehat{p}_{A_t}^2 c_{A_t} K}\right)}.$$

53 **end**

---

The equivalence of directly sampling $r_k$ and the procedure of sampling $\mu_k$ and $p_k$ from their respective posteriors and then multiplying them in Algorithm C.1, which can be mathematically expressed as follows:

$$
\begin{aligned}
&\mathbb{P}(r_k | \{R_t\}_{t:A_t=k}) \\
&\propto \mathbb{P}(r_k) \times \mathbb{P}(\{R_t\}_{t:A_t=k} \mid r_k) \\
&= \int_{\mu_k p_k = r_k} \mathbb{P}(\mu_k p_k) \times \mathbb{P}(\{R_t\}_{t:A_t=k, R_t=0} | u_k, p_k) \mathbb{P}(\{R_t\}_{t:A_t=k, R_t \neq 0} | u_k, p_k) \mathrm{d}\mu_k \mathrm{d}p_k \\
&= \int_{\mu_k p_k = r_k} \left[\mathbb{P}(p_k)\mathbb{P}(\{R_t\}_{t:A_t=k, R_t=0} | p_k)\right] \left[\mathbb{P}(\mu_k)\mathbb{P}(\{R_t\}_{t:A_t=k, R_t \neq 0} | u_k)\right] \mathrm{d}\mu_k \mathrm{d}p_k,
\end{aligned}
\tag{C.1}
$$

where the last equality results from two assumptions: 1) independent priors for $\mu_k$ and $p_k$, and 2) non-zero rewards depend solely on $\mu_k$, while zero rewards depend only on $p_k$.

An important aspect of the TS-algorithm is our use of clipped distributions for both the sub-Gaussian non-zero part $X$ and the precisely Bernoulli distributed part $Y$. As explained in Section 2.2, the reason for using a clipped Gaussian distribution for $X$ is due to its deviation from an exact Gaussian distribution. The rationale behind employing a clipped Beta distribution for the exactly Bernoulli distributed $Y$ is as follows: the objective is to establish concentration for the product random variable $R = X \times Y$. The product of the original Beta distribution and the clipped Gaussian does not adequately control the overestimation probability of sub-optimal $R_k$ in our proofs. This is primarily due to the proof techniques employed. For more details, refer to the proofs of Theorem 4.2 in Appendix H.

## C.2. Algorithms for Generalized Linear Contextual Bandits

### C.2.1. UCB ALGORITHMS AND REGULARITY CONDITIONS

As a concrete example, we consider the widely-used generalized linear contextual bandits with finite action set $\mathcal{A} = [K]$, where both functions $h$ and $g$ are structured as generalized linear functions here.

This setup is characterized by the *known* transformation functions $\psi_X(\cdot)$, $\psi_Y(\cdot)$, and the *known* link functions $g(\cdot)$ and $h(\cdot)$, such that $g(\mathbf{x}_t, A_t; \boldsymbol{\beta}) = g(\psi_X(\mathbf{x}_t, A_t)^\top \boldsymbol{\beta})$ and $h(\mathbf{x}_t, A_t; \boldsymbol{\theta}) = h(\psi_Y(\mathbf{x}_t, A_t)^\top \boldsymbol{\theta})$. When the $\varepsilon_t$ is sub-Gaussian, confidence radii for the ridge estimations $\widehat{\boldsymbol{\theta}}_t$ (or $\widehat{\boldsymbol{\beta}}_t$) in a compact parameter space can be fully determined by $\psi_Y(\mathbf{x}_t, A_t)$ (or $\psi_X(\mathbf{x}_t, A_t)$) and the sub-Gaussian variance proxy of $\varepsilon_t$. Specifically, $\rho_{Y,t}$ is chosen such that (Lemma 17.8 in Zhang, 2023)

$$\mathbb{P}\left(\forall 0 \leq t \leq T : \rho_{Y,t} \geq \sqrt{\lambda_U} + \sup_{a \in [K], \mathbf{x}_{1:t} \in \mathcal{X}} \left\| \sum_{s=1}^{t} \varepsilon_s \psi_Y(\mathbf{x}_s, a) \right\|_{\mathbf{U}_t^{-1}} \right) \geq 1 - \delta.$$

Thus, the ellipsoidal ratio $\rho_{Y,t}$ in Algorithm C.2 is constructed accordingly. A similar approach applies to $\rho_{X,t}$ for the non-zero component, though additional randomness from $X_s = g(\psi_X(\mathbf{x}_s, A_s)^\top \boldsymbol{\beta}^*)$ must be carefully handled, as discussed in Section 2.1. The proof of Theorem 4.3 provides further details, which we omit here for brevity.

---

**Algorithm C.2** General template of UCB for zero-inflated generalized linear bandits

---

**Data:** Link functions $\psi_X(\cdot)$, $\psi_Y(\cdot)$, $g(\cdot)$, and $h(\cdot)$. Ellipsoidal ratio sequences $\{\rho_{X,t}, \rho_{Y,t}\}$. Rridge parameters $\lambda_U$ and $\lambda_V$.

54   Set $\mathcal{H}_{\text{all}} = \{\}$ and $\mathcal{H}_{\text{non-zero}} = \{\}$. Set $\mathbf{U}_t = \lambda_U \mathbf{I}_q$ and $\mathbf{V}_t = \lambda_V \mathbf{I}_d$.

55   Randomly choose action $a_t \in [K]$ for $t \in [\tau]$;

56   **for** $t = \tau + 1, \ldots, T$ **do**

57     **for** $a \in [K]$ **do**

58       Set
$$\text{UCB}_t(a) = \left[ \psi_X(\mathbf{x}_t, a)^\top \widehat{\boldsymbol{\beta}}_t + \rho_{X,t} \| \psi_X(\mathbf{x}_t, a) \|_{\mathbf{V}_t^{-1}} \right] \left[ \psi_Y(\mathbf{x}_t, a)^\top \widehat{\boldsymbol{\theta}}_t + \rho_{Y,t} \| \psi_Y(\mathbf{x}_t, a) \|_{\mathbf{U}_t^{-1}} \right].$$

59     **end**

60     Take action $A_t = \arg\max_{a \in [K]} \text{UCB}_t(a)$.

61     Observe reward $R_t$, and set $Y_t = \mathbb{1}(R_t \neq 0)$.

62     Update the dataset as $\mathcal{H}_{\text{all}} \leftarrow \mathcal{H}_{\text{all}} \cup \{(\mathbf{x}_t, A_t, Y_t)\}$ and $\mathbf{U}_t = \mathbf{U}_t + \psi_Y(\mathbf{x}_t, A_t)\psi_Y(\mathbf{x}_t, A_t)^\top$.

63     Solve the equation
$$\widehat{\boldsymbol{\theta}}_t \in \left\{ \boldsymbol{\theta} \in \Theta : \sum_{s=1}^{t} \left[ Y_s - h(\psi_Y(\mathbf{x}_s, A_s)^\top \boldsymbol{\theta}) \right] \psi_Y(\mathbf{x}_s, A_s) = \mathbf{0} \right\}$$

64     **if** $R_t \neq 0$ **then**

65       Update the dataset as $\mathcal{H}_{\text{non-zero}} \leftarrow \mathcal{H}_{\text{non-zero}} \cup \{(\mathbf{x}_t, A_t, R_t)\}$;

66       Update $\mathbf{V}_t = \mathbf{V}_t + \psi_X(\mathbf{x}_t, A_t)\psi_X(\mathbf{x}_t, A_t)^\top$;

67       Solve
$$\widehat{\boldsymbol{\beta}}_t \in \left\{ \boldsymbol{\beta} \in \Gamma : \sum_{s \in [t]: Y_s = 1} \left[ R_s - g(\psi_X(\mathbf{x}_s, A_s)^\top \boldsymbol{\beta}) \right] \psi_X(\mathbf{x}_s, A_s) = \mathbf{0} \right\};$$

68   **end**

---

Next, we present the regularity conditions and selections of tuning parameters for Theorem 4.3. Let the true parameters be $\boldsymbol{\beta}^*$ and $\boldsymbol{\theta}^*$, and $\mathbf{Z}_{t,a} = \psi_X(\mathbf{x}_t, a) \in \mathbb{R}^d$ and $\mathbf{W}_{t,a} = \psi_Y(\mathbf{x}_t, a) \in \mathbb{R}^q$ for $a \in [K]$. These assumptions are quite standard in generalized linear contextual bandits (see, for example, Li et al., 2010; Deshpande et al., 2018; Wu et al., 2022).

**Assumption C.1** (Link Functions and Parameters). (i) The parameter space $\Gamma$ and $\Theta$ for $\boldsymbol{\beta}$ and $\boldsymbol{\theta}$ satisfies $\sup_{\boldsymbol{\beta} \in \Gamma} \|\boldsymbol{\beta}\|_2 \leq 1$ and $\sup_{\boldsymbol{\beta} \in \Theta} \|\boldsymbol{\theta}\|_2 \leq 1$;

(ii) The link function $g(\cdot)$ and $h(\cdot)$ is twice differentiable. Its first and second order derivatives are upper-bounded by $L_g$ and $M_g$, and $L_h$ and $M_h$, respectively;

(iii) $\kappa_g := \inf_{\|\mathbf{z}\|_2 \leq 1, \|\boldsymbol{\beta} - \boldsymbol{\beta}^*\|_2 \leq 1} \dot{g}(\mathbf{z}^\top \boldsymbol{\beta}) > 0$ and $\kappa_h := \inf_{\|\mathbf{w}\|_2 \leq 1, \|\boldsymbol{\theta} - \boldsymbol{\theta}^*\|_2 \leq 1} \dot{h}(\mathbf{w}^\top \boldsymbol{\theta}) > 0$;

(iv) $p_* := \inf_{\|\mathbf{w}\|_2 \leq 1, \boldsymbol{\theta} \in \Theta} h(\mathbf{w}^\top \boldsymbol{\theta}) > 0$.

**Assumption C.2** (Distributions). (i) $\|\mathbf{Z}_{t,a}\|_2 \leq 1$ and $\|\mathbf{W}_{t,a}\|_2 \leq 1$ for all $a \in [K]$;

(ii) The minimal eigenvalues for $\mathbb{E}[\mathbf{Z}_{t,a} \mathbf{Z}_{t,a}^\top]$ and $\mathbb{E}[\mathbf{W}_{t,a} \mathbf{W}_{t,a}^\top]$ satisfy $\lambda_{\min}\big(\mathbb{E}[\mathbf{Z}_{t,a} \mathbf{Z}_{t,a}^\top]\big) \geq \sigma_{\mathbf{z}}^2$ and $\lambda_{\min}\big(\mathbb{E}[\mathbf{W}_{t,a} \mathbf{W}_{t,a}^\top]\big) \geq \sigma_{\mathbf{w}}^2$ for all $a \in [K]$ with some positive $\sigma_{\mathbf{z}}$ and $\sigma_{\mathbf{w}}$.

(iii) The noise $\varepsilon_t$ is sub-Gaussian distributed, satisfying $\mathbb{E}[\varepsilon_t \mid \mathcal{F}_{t-1}] = 0$ and $\mathbb{E}[e^{s\varepsilon_t} \mid \mathcal{F}_{t-1}] \leq e^{s^2\sigma^2/2}$ for any $s \in \mathbb{R}$, where $\mathcal{F}_t := \sigma\langle\{(\mathbf{x}_1, R_1), \ldots, (\mathbf{x}_t, R_t)\}\rangle$ is an increasing sequence of sigma field.

**Assumption C.3** (Tuning Parameters). For any $\delta \in (0, 4/T)$,

(i) the random selection period $\tau$ is chosen as

$$\tau = \max\left\{\left[\left(\frac{C_1\sqrt{d/p_*} + C_2\sqrt{\log(1/\delta)/p_*}}{\sigma_{\mathbf{z}}^2}\right)^2 + \frac{2}{p_*\sigma_{\mathbf{z}}^2}\right], \frac{4\log(1/\delta)}{p_*^2}, \left(\frac{C_3\sqrt{q} + C_4\sqrt{\log(1/\delta)}}{\sigma_{\mathbf{w}}^2}\right)^2 + \frac{2}{\sigma_{\mathbf{w}}^2}\right\};$$

(ii) the ellipsoidal ratio sequences are chosen as

$$\rho_{X,t} = \kappa_g^{-1}\sigma\sqrt{4\log(1/\delta) + d\log(1 + \lambda_V^{-1}t/d)} \quad \text{and} \quad \rho_{Y,t} = \kappa_h^{-1}\sqrt{4\log(1/\delta) + q\log(1 + \lambda_U^{-1}t/q)},$$

where $C_\ell, \ell = 1, 2, 3, 4$ are some universal positive constants detailed in Appendix I.

A quick remark here is that all regularity conditions in Assumption C.1, C.2, and tuning parameter selections inAssumptionn C.3 are independent of the number of arms $K$.

### C.2.2. THOMPSON SAMPLING FOR GLM

Just like in the standard TS algorithm for linear contextual bandits (Agrawal & Goyal, 2013), we employ Gaussian priors for the sub-Gaussian noise. At each round $t \in [T]$, we sample parameters $\{\widetilde{\boldsymbol{\beta}}_t, \widetilde{\boldsymbol{\theta}}_t\}$ from Gaussian posterior distributions centered at the estimates $\{\widehat{\boldsymbol{\beta}}_t, \widehat{\boldsymbol{\theta}}_t\}$ with covariance matrices $\{\varrho_{X,t}^2 \mathbf{V}_t^{-1}, \varrho_{Y,t}^2 \mathbf{U}_t^{-1}\}$, where these estimates and covariance matrices are obtained from the UCB algorithm. The arm $A_t$ is then selected by maximizing the Thompson Sampling objective function $\psi_X(\mathbf{x}_t, a)^\top \widetilde{\boldsymbol{\beta}}_t \times \psi_Y(\mathbf{x}_t, a)^\top \widetilde{\boldsymbol{\theta}}_t$ leveraging the fact that both link functions are strictly increasing.

Let $A_t^*$ denote the optimal action at round $t$, and define $\mathbf{W}_t^* := \psi_X(\mathbf{W}_t, A_t^*)$. To ensure an appropriate choice of confidence radius sequences $\{\varrho_{Y,t}\}_{t \geq 0}$ (or $\{\varrho_{X,t}\}_{t \geq 0}$, both of which can be variance-dependent), it suffices to find $\varrho_{Y,t} > 0$ such that (Lemma 2 in Agrawal & Goyal, 2013)

$$\mathbb{P}\left(\mathbf{W}_t^* \widetilde{\boldsymbol{\theta}}_t > \mathbf{W}_t^* \boldsymbol{\theta}^* + \sqrt{\varrho_{Y,t}^2 \log^3 T}\|\mathbf{W}_t\|_{\mathbf{U}_t^{-1}} \mid \mathcal{F}_{t-1} \cap \mathcal{G}_t\right) \gtrsim T^{-\epsilon/2} \tag{C.2}$$

for some $\epsilon \in (0, 1)$, where $\mathcal{G}_t$ is a high-probability *good* event defined as

$$\mathcal{G}_t := \left\{\forall a \in [K] : \mathbf{W}_{t,a}^\top\big(\widehat{\boldsymbol{\theta}}_t - \boldsymbol{\theta}^*\big) \leq \varrho_{Y,t}\|\mathbf{W}_{t,a}\|_{\mathbf{U}_t^{-1}}\right\}.$$

From the proof of Theorem 4.3, we can set $\varrho_{Y,t} \gtrsim \rho_{Y,t}$ to ensure that $\mathcal{G}_t$ holds with probability at least $1 - \delta$ for $t > \tau$. To validate the anti-concentration property in (C.2), we use the inequality

$$\frac{1}{2\sqrt{\pi}x}\exp\left(-x^2/2\right) \leq \mathbb{P}(|\mathcal{N}(\mu, \sigma^2) - \mu| > x\sigma) \leq \frac{1}{\sqrt{\pi}x}\exp\left(-x^2/2\right).$$

Thus, choosing $\varrho_{Y,t} = \sqrt{\epsilon^{-1}q\log(1/\delta)}$ with $\epsilon \asymp \log T$ ensures that

$$\mathbb{P}\left(\mathbf{W}_t^* \widetilde{\boldsymbol{\theta}}_t > \mathbf{W}_t^* \boldsymbol{\theta}^* + \sqrt{\varrho_{Y,t}^2 \log^3 T}\|\mathbf{W}_t\|_{\mathbf{U}_t^{-1}} \mid \mathcal{F}_{t-1} \cap \mathcal{G}_t\right) \geq \frac{1}{4e\sqrt{\pi}T^\epsilon}.$$

---

**Algorithm C.3** General template of TS for ZI GLM bandits

---

**Data:** Link functions $\psi_X(\cdot)$, $\psi_Y(\cdot)$, $g(\cdot)$, and $h(\cdot)$. Confidence radio sequences $\{\varrho_{X,t}, \varrho_{Y,t}\}$. Rridge parameters $\lambda_U$ and $\lambda_V$.

69 Set $\mathcal{H}_{\text{all}} = \{\}$ and $\mathcal{H}_{\text{non-zero}} = \{\}$. Set $\mathbf{U}_t = \lambda_U \mathbf{I}_q$ and $\mathbf{V}_t = \lambda_V \mathbf{I}_d$.

70 Randomly choose action $a_t \in [K]$ for $t \in [\tau]$;

71 **for** $t = \tau + 1, \ldots, T$ **do**

72     **for** $a \in [K]$ **do**

73         Set

$$\text{TS}_t(a) = \left[\psi_X(\mathbf{x}_t, a)^\top \widetilde{\boldsymbol{\beta}}_t\right]\left[\psi_Y(\mathbf{x}_t, a)^\top \widetilde{\boldsymbol{\theta}}_t\right].$$

74     **end**

75     Take action $A_t = \arg\max_{a \in [K]} \text{TS}_t(a)$.

76     Observe reward $R_t$, and set $Y_t = \mathbb{1}(R_t \neq 0)$.

77     Update the dataset as $\mathcal{H}_{\text{all}} \leftarrow \mathcal{H}_{\text{all}} \cup \{(\mathbf{x}_t, A_t, Y_t)\}$ and $\mathbf{U}_t = \mathbf{U}_t + \psi_Y(\mathbf{x}_t, A_t)\psi_Y(\mathbf{x}_t, A_t)^\top$.

78     Solve the equation

$$\widehat{\boldsymbol{\theta}}_t \in \left\{\boldsymbol{\theta} \in \Theta : \sum_{s=1}^{t}\left[Y_s - h\big(\psi_Y(\mathbf{x}_s, A_s)^\top \boldsymbol{\theta}\big)\right]\psi_Y(\mathbf{x}_s, A_s) = \mathbf{0}\right\}.$$

79     Sample $\widetilde{\boldsymbol{\theta}}_t$ from distribution $\mathcal{N}(\widehat{\boldsymbol{\theta}}_t, \varrho_{Y,t}^2 \mathbf{U}_t^{-1})$.

80     **if** $R_t \neq 0$ **then**

81         Update the dataset as $\mathcal{H}_{\text{non-zero}} \leftarrow \mathcal{H}_{\text{non-zero}} \cup \{(\mathbf{x}_t, A_t, R_t)\}$;

82         Update $\mathbf{V}_t = \mathbf{V}_t + \psi_X(\mathbf{x}_t, A_t)\psi_X(\mathbf{x}_t, A_t)^\top$;

83         Solve

$$\widehat{\boldsymbol{\beta}}_t \in \left\{\boldsymbol{\beta} \in \Gamma : \sum_{s \in [t]: Y_s = 1}\left[R_s - g\big(\psi_X(\mathbf{x}_s, A_s)^\top \boldsymbol{\beta}\big)\right]\psi_X(\mathbf{x}_s, A_s) = \mathbf{0}\right\};$$

84         Sample $\widetilde{\boldsymbol{\beta}}_t$ from distribution $\mathcal{N}(\widehat{\boldsymbol{\beta}}_t, \varrho_{X,t}^2 \mathbf{V}_t^{-1})$.

85 **end**

---

Moreover, for any $t \in [T]$,

$$\varrho_{Y,t} \asymp \sqrt{q \log(1/\delta)\log(T)} \gtrsim \sqrt{\log(1/\delta) + q\log(1 + t/q)} \asymp \rho_{Y,t}.$$

Similarly, we select $\varrho_{X,t}$ correspondingly, leading to the following choices:

$$\varrho_{X,t} \asymp \sqrt{d \log(1/\delta)\log(T)} \qquad \text{and} \qquad \varrho_{Y,t} \asymp \sqrt{q \log(1/\delta)\log(T)}. \tag{C.3}$$

### C.2.3. EXTENSION FOR HEAVY-TAILED NOISE

Similarly, one can also devise the UCB-type algorithm for sub-Gaussian $\varepsilon_t$, as detailed in Algorithm C.2 in Appendix C. In cases where $\varepsilon_t$ follows a general sub-Weibull distribution or only has finite moments of order $1 + \epsilon$ with $\epsilon \in (0, 1]$, specific adjustments are necessary, such as applying the median of means (MoM) estimator for linear bandits (Medina & Yang, 2016; Shao et al., 2018) or Huber regression (Sun et al., 2020; Kang & Kim, 2023).

## D. Supplement to Simulation

### D.1. Detailed Simulation Setting for Multi-Armed Bandits

Our simulations were performed on a Mac mini (2020) with an M1 chip and 8GB of RAM. We first detail the UCB baselines and TS baselines as follows:

**UCB baselines:**

We consider following UCB-type algorithms for comparison. At round $t$, the agent takes action $A_t = \max_{k \in [K]} U_k(t)$ with

the $k$-th arm's upper bounds $U_k(t) = \overline{R}_k(t) + \sqrt{\frac{2\tau_k^2 \log(2/\delta)}{c_k(t)}}$ for sub-Gaussian rewards $U_k(t) = \overline{R}_k(t) + \alpha_k^2 \sqrt{\frac{2 \log(2/\delta)}{c_k(t)}} + \alpha_k \frac{\log(2/\delta)}{c_k(t)}$ for sub-Exponential rewards. Here, the size parameters $\tau^2$ and $\alpha$ for the true rewards are determined using the following methods:

- Using the original size parameters for the non-zero part $X_k$, assuming they are known, as the size parameter for constructing $U_k(t)$;

- Using the estimated variance of the rewards from $k$-th arm as the size parameter $\tau_k^2$ and $\alpha_k$ for constructing $U_k(t)$;

- Using the estimated size parameter for $k$-th arm as follows:

  - For sub-Gaussian $X_k$ with sub-Gaussian variance proxy $\sigma_k^2$, the sub-Gaussian variance proxy for $R_k$ is solved by

  $$\tau_k^2 = \max_{s \in \mathbb{R}} \frac{2}{s^2} \left[ -s\widehat{\mu}_k\widehat{p}_k + \log(1 - \widehat{p}_k + \widehat{p}_k e^{s\widehat{\mu}_k + s^2\sigma_k^2/2}) \right].$$

  where $\widehat{p}_k$ is taken as the average of observations $Y_k$;
  - For sub-Exponential $X_k$ with the single sub-Exponential parameter $\lambda_k$, the sub-Exponential parameter $\alpha_k$ for the rewards from the $k$-th arm is solved by

  $$\alpha_k^2 = \lambda_k^2 \vee \max_{s \in \mathbb{R}} \frac{2}{s^2} \left[ -s\widehat{\mu}_k\widehat{p}_k + \log(1 - \widehat{p}_k + p_k e^{s\widehat{\mu}_k + s^2\lambda_k^2/2}) \right].$$

  Here $\widehat{\mu}_k$ and $\widehat{p}_k$ are taken as the averages of observations $X_k$ and $Y_k$, respectively.

- (**Strong baseline**) Using the true size parameter for the reward of each arm $\{R_k\}_{k=1}^K$.

**TS baselines:**

Here we exclusively consider the TS-type algorithm suitable for general sub-Gaussian distributions, namely the MOTS algorithm (Jin et al., 2021). For Gaussian and mixed-Gaussian rewards, we can directly apply both Algorithm C.1 and the MOTS algorithm. But, when applying with Exponential rewards, we adopt Algorithm 1 from Shi et al. (2023). In doing so, we integrate their step 5 with our algorithm and the MOTS algorithm. This ensures that both our method and the one proposed in Jin et al. (2021) are correctly adapted for use with sub-Gaussian distributions after the GMS generation.

**Simulation settings and extended results:**

For UCB-type algorithms, we set the confidence level $\delta = 4/T^2$ consistently across all experiments. Prior parameters and tuning parameters for TS-type algorithms follow the recommendations in (Jin et al., 2021; Shi et al., 2023) for the MOTS algorithm and GMS generation. Beyond Figure 2, we provide additional simulations across different parameter ranges: Figure 5 shows results for alternative $p$ settings, while Figures 6 and 7 examine extremely small and relatively large zero-inflation probabilities, respectively. We also extend our analysis to bounded rewards in Figure 8.

Our methods consistently outperform existing approaches across most settings. However, occasional underperformance against certain proxy-based UCB methods can occur, particularly with exponential rewards under specific parameter ranges (e.g., $p_k \sim U[0.1, 0.3]$). As demonstrated in Lemma E.1, such proxy methods can become unreliable under high variance or strong zero-inflation, whereas our approach demonstrates greater robustness by directly modeling the zero-inflated structure.

For bounded rewards, our UCB method shows clear advantages in most cases. The only exception occurs when the zero-inflation level is low (i.e., $p_k$ is large), in which case the true Hoeffding-based UCB (with access to the exact bound) performs slightly better. Nonetheless, this case represents a rare (as the exact bound baseline is not available in practice). In all other cases, especially when skewness or sparsity increases, our method shows clear advantages.

All simulations for each parameter setting and distribution were completed within 24 hours.

## D.2. Detailed Simulation Setting for Contextual Bandits

In our contextual bandits scenario, we employ two UCB baseline methods for comparison of our method:

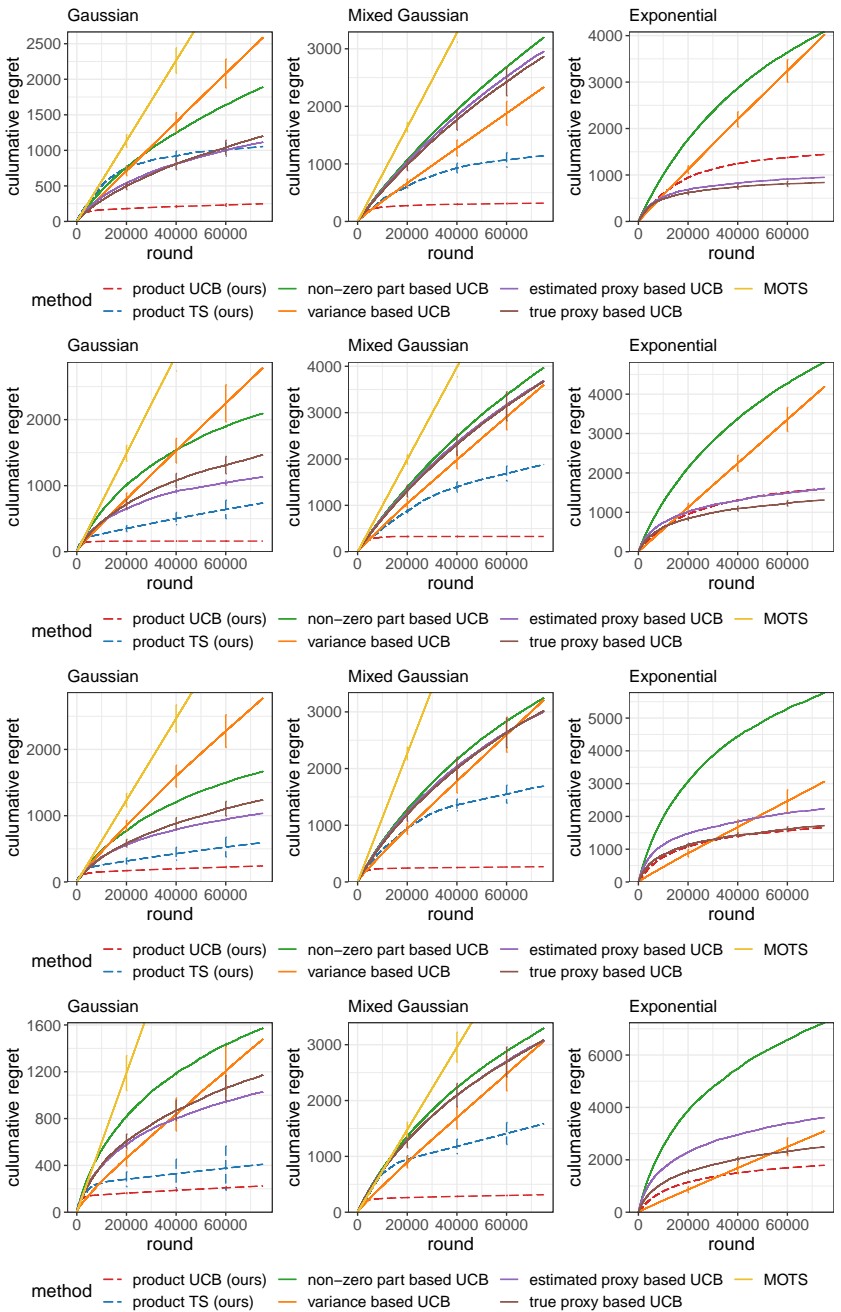

*Figure 5.* Simulation for zero-inflated MAB with $K = 10$ and $T = 75000$ with $N = 50$ replications. The four rows represent $p \sim U[0.10, 0.15]$, $p \sim U[0.15, 0.20]$, $p \sim U[0.20, 0.25]$, and $p \sim U[0.25, 0.30]$, respectively.

- (Naive Method) We ignore the zero-inflated structure. The upper bound for any action $a \in \mathcal{A}_t$ is defined as:

$$\mathrm{UCB}_t(a) := \psi_X(\mathbf{x}_t, a)^\top \widehat{\boldsymbol{\beta}}_{\mathrm{naive}} + \sqrt{\rho_{X,t}} \|\psi_X(\mathbf{x}_t, a)\|_{U_{\mathrm{naive}}}$$

  with $U_{\mathrm{naive}}$ and $\widehat{\boldsymbol{\beta}}_{\mathrm{naive}}$ are computed by $U_{\mathrm{naive}} = \lambda_U I_d + \sum_{s=1}^{t} \psi_X(\mathbf{x}_s, A_s) \psi_X(\mathbf{x}_s, A_s)^\top$ and $\widehat{\boldsymbol{\beta}}_{\mathrm{naive}} = U_{\mathrm{naive}}^{-1} \sum_{s=1}^{t} R_s \psi_X(\mathbf{x}_s, A_s)$.

- (Integrated Component Method) We correctly account for the zero-inflated structure. The upper bound for any action

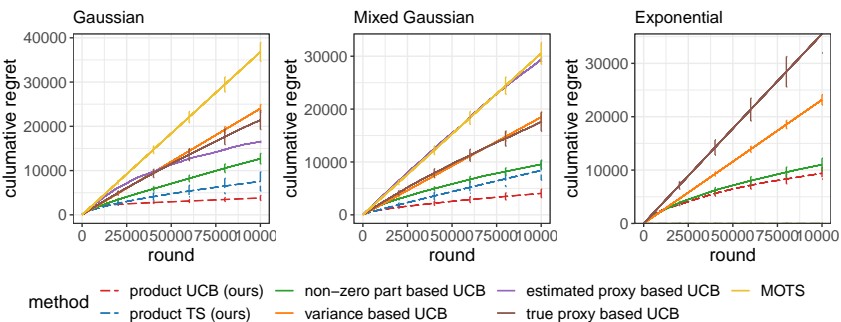

**Figure 6.** Simulation for zero-inflated MAB under $p \in U[0.0005, 0.01]$ with $K = 10$ and $T = 10000$ with $N = 50$ replications.

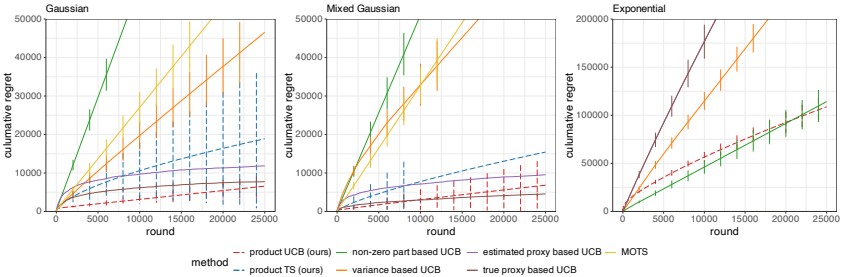

**Figure 7.** Simulation for zero-inflated MAB under $p \in U[0.75, 0.95]$ with $K = 10$ and $T = 25000$ with $N = 50$ replications.

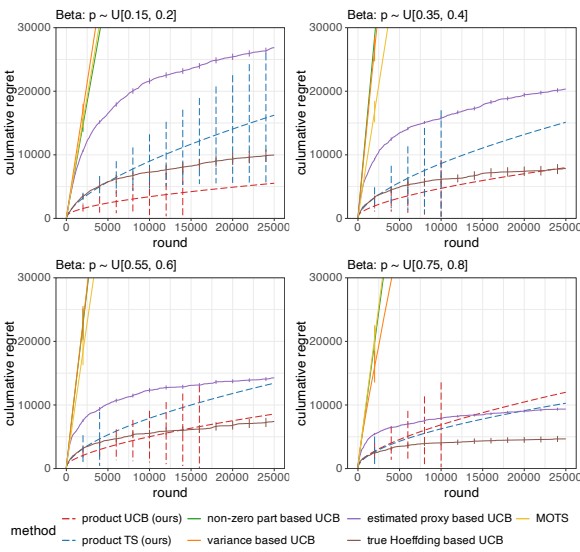

**Figure 8.** Simulation for zero-inflated MAB under the scaled Beta distribution of the form $X_k \sim 2\mu_k \cdot \text{Beta}(p_k, p_k)$ such that $\mathbb{E}[X_k] = \mu_k$ with $K = 10$ and $T = 10000$ with $N = 50$ replications.

$a \in \mathcal{A}_t$ is:

$$\text{UCB}_t(a) := \psi_X(\mathbf{x}_t, a)^\top \widehat{\boldsymbol{\beta}}_{\text{integrated}} \times h\big(\psi_Y(\mathbf{x}_t, a)^\top \widehat{\boldsymbol{\theta}}_{\text{integrated}}\big) + \sqrt{\rho_{X,t} \vee \rho_{Y,t}} \left\| \left[ \begin{array}{c} \psi_X(\mathbf{x}_t, a) \\ \psi_Y(\mathbf{x}_t, a) \end{array} \right] \right\|_{W_{\text{integrated}}}$$

with

$$\left[ \begin{array}{c} \widehat{\boldsymbol{\beta}}_{\text{integrated}} \\ \widehat{\boldsymbol{\theta}}_{\text{integrated}} \end{array} \right] = \arg \min_{\boldsymbol{\theta}, \boldsymbol{\beta} \in \Theta \times B} \left\| \sum_{s=1}^{t} \left[ R_s - \psi_X(\mathbf{x}_s, A_s)^\top \boldsymbol{\beta} \times h\big(\psi_Y(\mathbf{x}_s, A_s)^\top \boldsymbol{\theta}\big) \right] \right\|$$

$$\text{and } W_{\text{integrated}} = \lambda_V I_{2d} + \sum_{s=1}^{t} \begin{bmatrix} \psi_X(\mathbf{x}_s, A_s) \\ \psi_Y(\mathbf{x}_s, A_s) \end{bmatrix} \begin{bmatrix} \psi_X(\mathbf{x}_s, A_s)^\top & \psi_Y(\mathbf{x}_s, A_s)^\top \end{bmatrix}.$$

The first baseline *Naive Method* results in model misspecification. We design this baseline because this issue will arise when researchers fail to correctly specify the zero-inflation structure. In contrast, the second baseline *Integrated Component Method*, appropriately specifies the model. The second method estimates two unknown vector $\boldsymbol{\beta}$ and $\boldsymbol{\theta}$ as a single vector $(\boldsymbol{\beta}^\top, \boldsymbol{\theta}^\top)^\top$ from the model $\psi_X(\mathbf{x}_t, a)^\top \boldsymbol{\beta} \times h(\psi_Y(\mathbf{x}_t, a)^\top \boldsymbol{\theta}) + \varepsilon_t$, which originates from a range of generalized linear contextual bandit literature (Li et al., 2017; Chen et al., 2020; Zhao et al., 2020). Similarly, for the naive TS and integrated TS algorithm, we sample $\widetilde{\boldsymbol{\beta}}_{\text{naive}}$ from $\mathcal{N}(\widehat{\boldsymbol{\beta}}_{\text{native}}, \phi_{\boldsymbol{\beta}}^2 U_{\text{naive}}^{-1})$ and $(\widetilde{\boldsymbol{\beta}}_{\text{integrated}}^\top, \widetilde{\boldsymbol{\theta}}_{\text{integrated}}^\top)^\top$ from $\mathcal{N}((\widehat{\boldsymbol{\beta}}_{\text{integrated}}^\top, \widehat{\boldsymbol{\theta}}_{\text{integrated}}^\top)^\top, (\phi_{\boldsymbol{\beta}}^2 \vee \phi_{\boldsymbol{\theta}}^2) \times W_{\text{naive}}^{-1})$ respectively.

Our experimental setup aligns with that described in Section D.2 of (Wu et al., 2022): the dimension $d$ is set to 10, and there are $K = 100$ arms in this context. The true parameter for the non-zero part, $\boldsymbol{\beta}$, has a norm of 1 and is uniformly distributed, with each entry $\boldsymbol{\beta}_i \sim U[0, 1]$ for $i \in [d]$. The zero-inflation structure $\boldsymbol{\theta}$ exhibits sparsity $s$, meaning only $s$ elements in $\boldsymbol{\theta}$ are derived from a uniform distribution, while the rest are zeros. Similarly, for each arm $k \in [K]$, we generate $\boldsymbol{\nu}_k$ exactly the same as $\boldsymbol{\theta}$. At each round $t$, the context of each arm $k$ is sampled as $\mathbf{x}_{k,t} \sim \mathcal{N}_d(\boldsymbol{\nu}_k, \frac{1}{2K} I_d)$. Thus, in this setting, we have $\psi_X(\mathbf{x}_t, A_t) = \mathcal{N}(\boldsymbol{\nu}_{A_t}, \frac{1}{2K} I_d)$ and $\psi_Y(\mathbf{x}_t, A_t) = \varphi_Y(\mathcal{N}(\boldsymbol{\nu}_{A_t}, \frac{1}{2K} I_d))$ with element-wised mapping $\varphi_Y : \mathbf{x} \mapsto \sin(\mathbf{x})$. Lastly, the noise term $\epsilon_t$ follows the distribution $F(0, \sigma^2)$ with a mean-zero distribution $F$ and a standard deviation of $\sigma = 1$.

In detailing the tuning parameters for UCB algorithms, we initially adopt the configurations in Appendix C.2. For TS algorithms, we follow the original setting in Agrawal & Goyal (2013) while adding hyper-tuning parameters $\epsilon_X$ and $\epsilon_Y$ such that $\epsilon_X \asymp \epsilon_Y \asymp \log^{-1} T$ suggested by Zhong et al. (2021) as $\phi_{\boldsymbol{\beta}}^2 = \frac{24\sigma^2 d}{K^2 \epsilon_X} \log(1/\delta)$ and $\phi_{\boldsymbol{\theta}}^2 = \frac{24\sigma^2 d}{K^2 \epsilon_Y} \log(1/\delta)$. For simplicity, in our implementation, we fix $\epsilon_X$ and $\epsilon_Y$ to $\log^{-1} T$. Additionally, for both UCB and TS algorithms, we set the parameter $\delta$ to $1/T$.

Further details are provided in Section 5 of Kang & Kim (2023). In addition to Figure 3, we present simulation results with different sparsity levels $s$ in Figure 9 and comparisons with another semiparametric contextual bandit algorithm in Figure 10. We choose SPUCB from Peng et al. (2019) as the representative baseline, as other semiparametric algorithms exhibit similar performance. The results for various link functions $h(\cdot)$ are quite similar, and therefore, have been omitted for brevity. Like in MAB, simulations were conducted on a Mac mini (2020) with an M1 chip and 8GB of RAM, with all runs completed within 48 hours.

### D.3. Detailed Simulation Setting for Real Data Application

The loan records data, collected by Columbia Business School, encompasses all auto loan records (totaling 208,085) from a major online auto loan lending company, spanning from July 2002 to November 2004. This dataset can be accessed upon request at "https://business.columbia.edu/cprm". In our analysis, we treat the monthly payment and loan term as the *raw* action $A_t \in \mathbb{R}_+$. We then compute the price as:

$$\text{price} = \text{Monthly payment} \times \sum_{t=1}^{\text{Term}} (1 + \text{Prime rate})^{-t} - \text{ amount of loan requested},$$

which accounts for the net present value of future payments, following the approach used in (Phillips et al., 2015; Bastani et al., 2022; Chen et al., 2023). Similarly, we exclude outlier records, as done in (Chen et al., 2023). In this dataset, the binary decision of the applicant is denoted as $Y_t \in \{0, 1\}$. Following the approach in (Chen et al., 2023), we define the synthetic **observed** reward as $R_t = A_t \times Y_t$. The context vector $\mathbf{x}_t \in \mathbb{R}^5$ represents a set of features identified as significant in (Ban & Keskin, 2021; Chen et al., 2023). This includes the FICO credit score, the approved loan amount, the prime rate, the competitor's rate, and the loan term.

The range of the raw action $A_t$ spans from 0.0004 to 28.5725. To simplify the online decision process, we introduce a discretization of $A_t$ into categorized actions, denoted as $\widetilde{A}_t$. This discretization follows a specific transformation, $\ell(A_t)$,

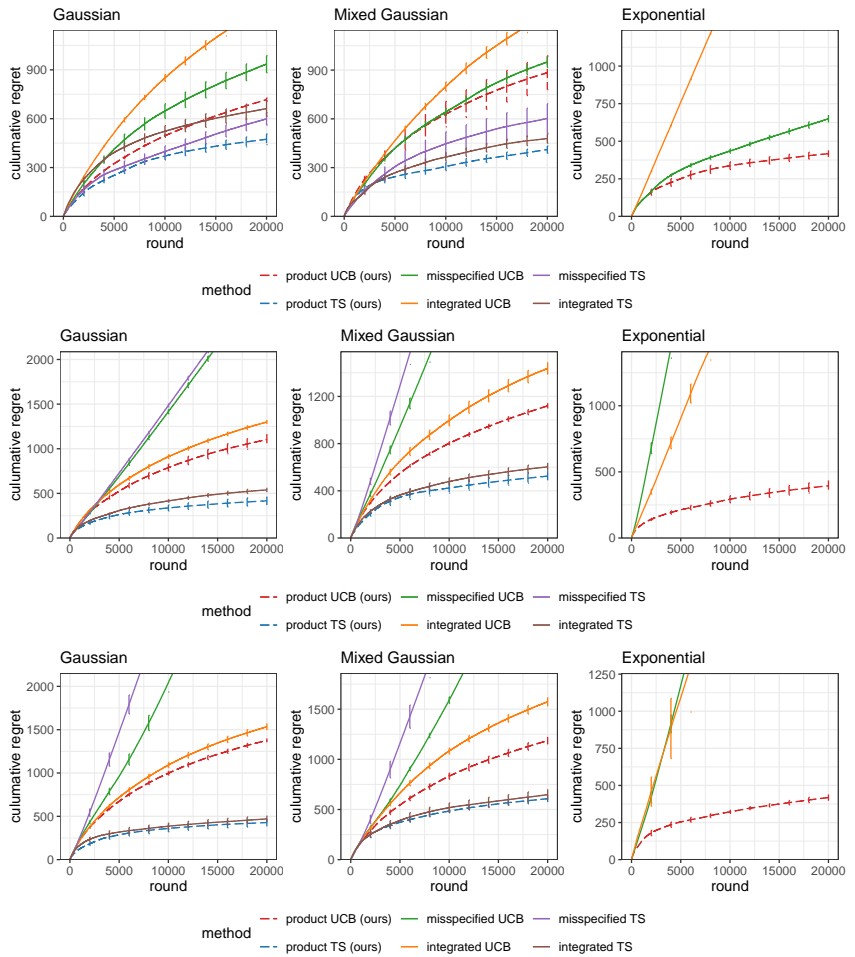

*Figure 9.* Simulation for zero-inflated contextual bandits and $T = 20000$ with $N = 25$ replications with the different sparsity levels. The three rows represent $s = 1, 3, 5$, respectively.

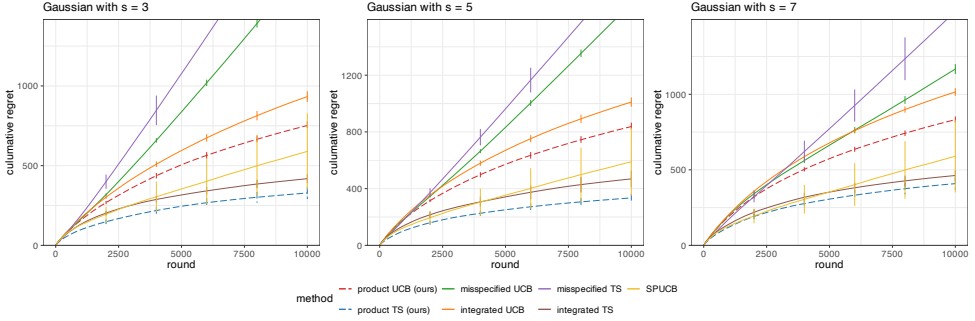

*Figure 10.* Comparison with SPUCB algorithm for zero-inflated contextual bandits with $T = 10,000$ rounds over $N = 25$ independent replications across different sparsity levels.

defined as follows:

$$
\widetilde{A}_t = \ell(A_t) := \begin{cases} \text{``low''}, & \text{if } A_t \leq 1.5400, \\ \text{``medium''}, & \text{if } 1.5400 < A_t \leq 3.6390, \\ \text{``high''}, & \text{if } 3.6390 < A_t \leq 5.4164, \\ \text{``very high''}, & \text{if } 5.4164 < A_t \leq 8.5466, \\ \text{``luxury''}, & \text{if } A_t > 8.5466 \end{cases} .
$$

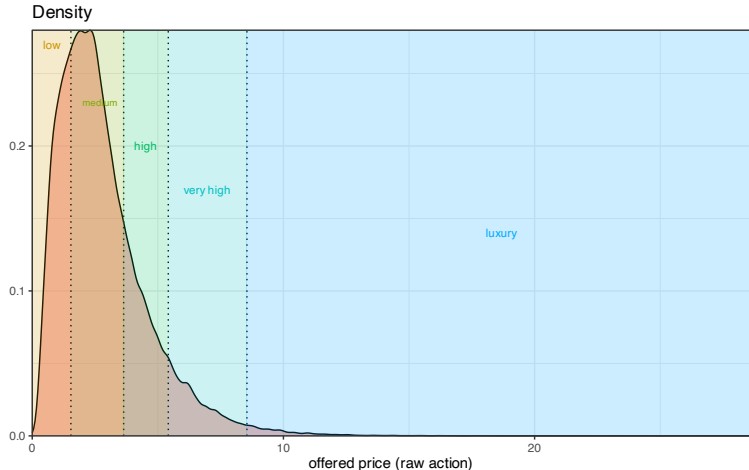

*Figure 11.* The density of the raw actions and the corresponding discretization.

This discretization is derived from specific quantiles of the offered price: the $25\%$ and $75\%$ quantiles, the $66\%$ quantile of the remaining offer beyond the $75\%$ quantile, and the $95\%$ quantile of the remaining offer. We have accordingly plotted the density $f(a)$ of the raw action set $\{A_t\}_{t=1}^T$, as shown in Figure 11.

Again, the decision of applicants mainly depends on the nominal profit $A_t$ offered, but it will also depend on the environment. Thus, we first model the binary selection $Y_t \in \{0, 1\}$, whether accept the offer, of the applicant as a binary regression as

$$\mathbb{P}(Y_t = 1) = h\big((A_t, 1, b^\top(\mathbf{x}_t))\boldsymbol{\theta}\big) = h\big(A_t\theta_1 + \theta_2 + b^\top(\mathbf{x}_t)\boldsymbol{\theta}_{-1,-2}\big)$$

with some different functions $h(\cdot)$ and suitable context transformation $b(\cdot)$. Intuitively, the raw action $A_t$, the nominal profit that company offer, is the most relevant factor. The offline estimation and the performance of $\boldsymbol{\theta}$ via estimated FDR (Dai et al., 2023) using the whole dataset is shown in Table 3.

*Table 3.* Estimated FDR under data splitting. The function $b(\cdot)$ is chosen as the best degree in the bracket.

| $b(\cdot)$ \ $h(\cdot)$ | logit | probit | cauchit | log | cloglog |
|---|---|---|---|---|---|
| linear | 0.1831 | 0.1912 | 0.1699 | 0.0983 | 0.1835 |
| polynomial | 0.1682 **(3)** | 0.1705 **(3)** | 0.1610 **(3)** | 0.3483 **(1)** | 0.1707 **(3)** |
| spline | 0.1743 **(5)** | 0.1755 **(5)** | 0.1692 **(3)** | 0.3483 **(1)** | 0.1790 **(5)** |
| kernel (Epanechnikov) | 0.2052 | | | | |
| kernel (Triangale) | 0.2050 | | | | |
| kernel (Gaussian) | 0.2049 | | | | |

As shown in Table 3, when we choose $b(\cdot) : \mathbf{x} \mapsto \mathbf{x}$ , the estimated FDR for the whole dataset is no more than $0.1$, a quite small value, which means this model characterize the dataset quite good (Dai et al., 2023; Ren & Barber, 2024; Wei et al., 2024).

Next, for to simulate the counterfactual outcomes in the online setting, recognizing that rewards are deterministic when based on non-zero rewards, we introduce noise to better simulate real-world scenarios, such as deviations in payment plans or additional service purchases. Two important points here are: (1) Conditional on the raw action the noise is mean-zero; (2) that the noise should be related to the context. This indicates the true reward in the online procedure at each round $X_t = g(A_t, \mathbf{x}_t; \boldsymbol{\beta}) + \varepsilon_t$ the company will obtain satisfies

$$\mathbb{E}[X_t \mid A_t] = \mathbb{E}[g(A_t, \mathbf{x}_t; \boldsymbol{\beta}) + \varepsilon_t \mid A_t] = A_t.$$

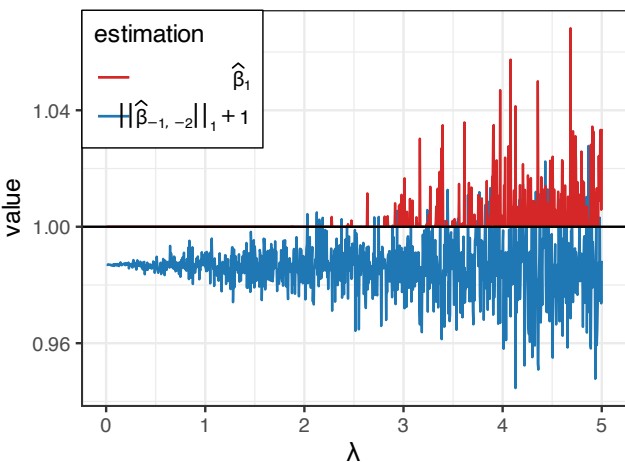

*Figure 12.* The estimation of $\boldsymbol{\beta}$ under different $\lambda$, the red line is the estimated coefficient for the raw action $A_t$ and the blue line is the $\ell_1$-norm plus one for the estimated coefficients of the environment $b(\mathbf{x}_t)$.

For simplifying the procedure, here we just let

$$g(A_t, \mathbf{x}_t; \boldsymbol{\beta}) = \left(A_t, 1, b^\top(\mathbf{x}_t)\right)\boldsymbol{\beta} = \beta_1 A_t + \beta_2 + \boldsymbol{\beta}^\top_{-1,-2} b(\mathbf{x}_t)$$

with the same context transformation function $b(\cdot)$. Note that we do not have $X_t$ for the real data (the observed reward is $R_t = A_t \times Y_t$ instead of $R_t = X_t \times Y_t$), so we manually create $X_t = A_t + \text{Exp}(\lambda) - \lambda$, which satisfy $\mathbb{E}[X_t \mid A_t] = A_t$ and use the whole dataset to estimate $\widehat{\boldsymbol{\beta}}$. The result of offline estimation $\widehat{\boldsymbol{\beta}}$ with different $\lambda$ is shown in Figure 12. From the Figure 12, we know that any $\lambda \in (0, 2]$ is reasonable as we have $\widehat{\beta}_1 \approx 1$ and $\|\widehat{\boldsymbol{\beta}}_{-1,-2}\|_1 \approx 0$, which means the nominal profit the company proposed, $A_t$, by the company plays a major role, but the context $\mathbf{x}_t$ does have some influence.

Then we implemented our method and the different baselines described in Section 5 in an online setting to compare their performance. Specially, at each round $t$, the company selects an discrete action $\widetilde{A}_t$ from the discrete set $\mathcal{A}_{\text{discrete}} = \{\text{"low", "medium", "high", "very high", "luxury"}\}$. Then a company to have a potential nominal profit $A_t$ with randomly sampled from the truncated original density for the profit

$$f(a)\mathbb{1}\{a \in \ell^{-1}(\widetilde{A}_t)\},$$

Next, the potential real profit the company would receive with the current context $\mathbf{x}_t$ is $\widetilde{X}_t = \left(A_t, 1, b^\top(\mathbf{x}_t)\right)\widehat{\boldsymbol{\beta}}$. Besides, the applicant accordingly makes a binary decision $\widetilde{Y}_t \in \{0, 1\}$ with probability $\mathbb{P}(\widetilde{Y}_t = 1) = h\left((A_t, 1, b^\top(\mathbf{x}_t))\boldsymbol{\theta}\right)$. Next, the company receives the real reward calculated as $\widetilde{R}_t = \widetilde{X}_t \times \widetilde{Y}_t$.

The regret for each round is calculated as $\mathbb{E}[\widetilde{Y}_t^* \widetilde{X}_t^*] - \mathbb{E}[\widetilde{Y}_t \widetilde{X}_t]$, where $\widetilde{Y}_t^*$ and $\widetilde{X}_t^*$ represent the decision of the company and the random true profit according to the optimal action $\widetilde{A}_t^*$ defined as

$$\widetilde{A}_t^* \in \arg\max_{\widetilde{a} \in \mathcal{A}_{\text{discrete}}} \int (a, 1, b^\top(\mathbf{x}_t))\widehat{\boldsymbol{\beta}} h\left((a, 1, b^\top(\mathbf{x}_t))\widehat{\boldsymbol{\theta}}\right) f(a)\mathbb{1}\{\ell^{-1}(\widetilde{a})\} \, \mathrm{d}a.$$

These online process mirrors that in Appendix D.2, with the same tuning parameters. We treat this dataset as a heavy-tailed contextual bandit, and thus use the Huber regression technique as described in (Kang & Kim, 2023). Once again, the simulations were conducted on a Mac mini (2020) with an M1 chip and 8GB of RAM, with results obtained within 6 hours.

## E. Supporting Lemma and Figures for Motivations

**Lemma E.1.** *Assume $X_t - \mu \sim \text{subG}(\sigma^2)$ and $Y_t \sim \text{Bernoulli}(p)$, independent of $Y_t$. Let $R_t = X_t \times Y_t$. Then $R_t - \mu p$ is sub-Gaussian, with its sub-Gaussian variance proxy denoted by $\tau^2$, satisfying:*

(i) *Given any $p \in (0,1)$, $\sigma^2 > 0$, and arbitrarily large $M > 0$, there exists $\mu_* > 0$ such that $\tau^2 > M\sigma^2$ for any $\mu > \mu_*$;*

(ii) *For any arbitrarily large $M > 0$, there exist $\sigma_*^2, p_*, \mu_*$ such that $\tau^2 > M\,\mathrm{var}(R_t)$ for any $\mu > \mu_*$, $\sigma^2 < \sigma_*^2$, and $p \in [p_*, 1)$;*

(iii) (a) *For any $\sigma^2 > 0$ and arbitrarily large $M > 0$, there exists $\mu_* > 0$ and $p_* \in (0,1)$ such that*

$$\left. \frac{\partial \tau^2}{\partial p} \right|_{\mu=\mu',p=p'} > M$$

*for any $\mu' \in (0, \mu_*]$ and $p' \in [p_*, 1)$;*

(b) *For any $\mu > 0$ and arbitrarily large $M > 0$, there exists $\sigma_*^2 > 0$ and $p_* \in (0,1)$ such that*

$$\left. \frac{\partial \tau^2}{\partial p} \right|_{\sigma^2=\sigma'^2,p=p'} > M$$

*for any $\sigma'^2 \in (0, \sigma_*^2]$ and $p' \in [p_*, 1)$;*

(c) *For any $\mu \in \mathbb{R}$ and arbitrarily large $M > 0$, there exists $\sigma_*^2 > 0$ and $p_* \in (0,1)$ such that*

$$\left. \frac{\partial \tau^2}{\partial \mu} \right|_{\sigma^2=\sigma'^2,p=p'} > M$$

*for any $\sigma'^2 \in (0, \sigma_*^2]$ and $p' \in [p_*, 1)$.*

*Analogous results apply for $X_t - \mu \sim \mathrm{subE}(\lambda)$, with $\sigma^2$ replaced by $\lambda^2$ and $\tau^2$ by $\alpha^2$.*

We provide visual clarification for Lemma E.1 in Figure 13, which illustrates the lemma in the context where $X_t \sim \mathcal{N}(\mu, \sigma^2)$. Figure 13 (a) and (b) show the values of $\tau^2$ under different $\mu$ and $p$ values when $\sigma^2 = 1$, and the ratio of $\tau^2/\mathrm{var}(R_t)$ under varying $p$ values when $\mu = 100$, respectively. Figure 13 (c) presents the numerical derivative $\frac{\partial \tau^2}{\partial p}$ under different $\mu$ and $p$ values when $\sigma^2 = 1$. Figures 13(d) and (e) display the numerical derivatives $\frac{\partial \tau^2}{\partial p}$ and $\frac{\partial \tau^2}{\partial \mu}$ across a range of $\mu$ and $\sigma^2$ values when $\mu = 100$ and $\mu = 0.5$, respectively.

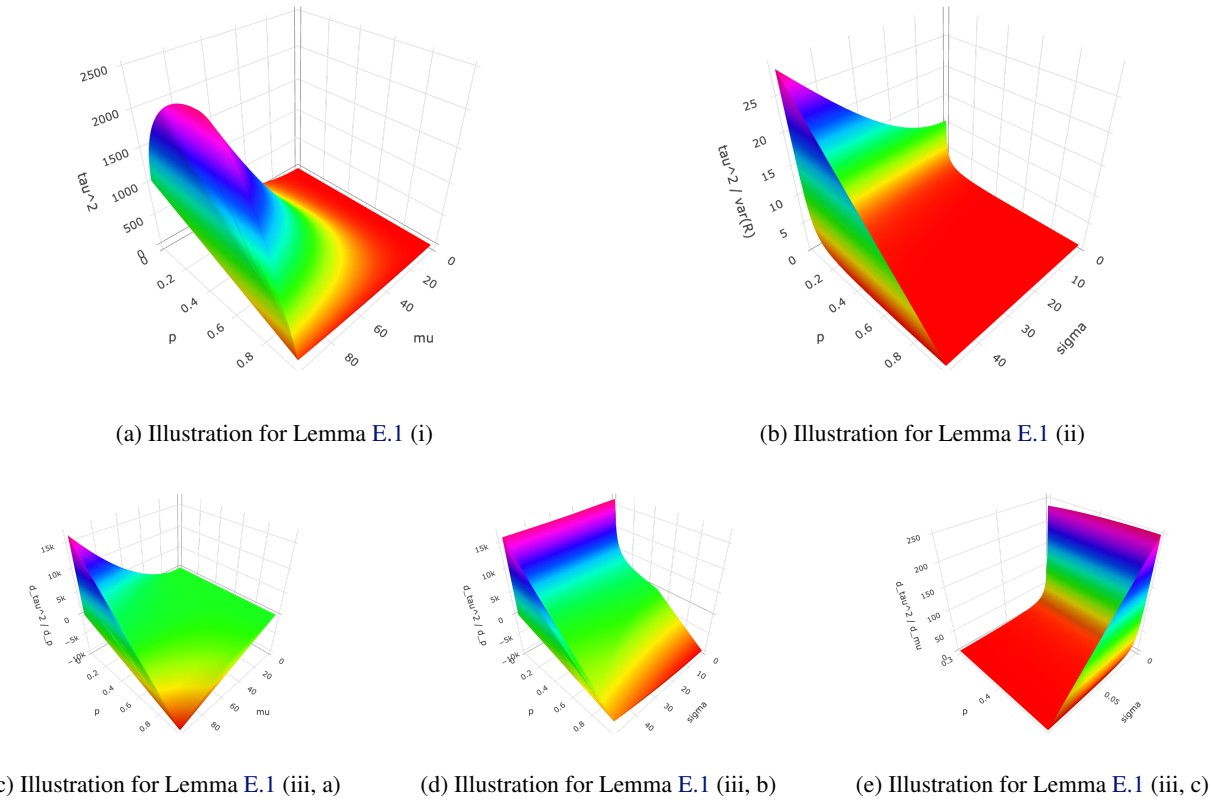

(a) Illustration for Lemma E.1 (i)

(b) Illustration for Lemma E.1 (ii)

(c) Illustration for Lemma E.1 (iii, a)

(d) Illustration for Lemma E.1 (iii, b)

(e) Illustration for Lemma E.1 (iii, c)

*Figure 13.* The illustration Lemma E.1 when $X_t$ is exactly Gaussian distributed.

# F. Proof of Lemmas

In this section, we will provide the proofs for the concentration results presented in Section 2. We will also include relevant theoretical background and discussions.

**Notations:** Let $\mathbb{P}_\xi(A) = \int_A \mathrm{d}F_\xi(x)$ denote the probability of the event $A$, where $F_\xi(x)$ is the distribution function of the random variable $\xi$. Similarly, let $\mathbb{E}_\xi f(\xi) = \int f(x)\mathrm{d}F_\xi(x)$ represent the expectation. We define $a \vee b = \max\{a, b\}$ and $a \wedge b = \min\{a, b\}$ for any real numbers $a$ and $b$.

We first define the *Revised-Generalized Bernstein-Orlicz (RGBO)* transformation function $\Psi_{\theta,L}(\cdot)$ based on the inverse function

$$\Psi_{\theta,L}^{-1}(s) := \sqrt{\log(1+s)} + L(\log(1+s))^{(1/\theta)\vee 1}$$

for any $t \geq 0$. It is worthy to note that we replace $(\log(1+s))^{1/\theta}$ with $(\log(1+s))^{(1/\theta)\vee 1}$ in the Generalized Bernstein-Orlicz function defined in van de Geer & Lederer (2013); Kuchibhotla & Chakrabortty (2022). It is easy to verify that is monotone increasing and $\Psi_{\theta,L}(0) = 0$, and we can define the RGBO norm of a variable random $X$ such that $\|X\|_{\Psi_{\theta,L}} = \inf\{\eta > 0 : \mathbb{E}\Psi_{\theta,L}(|X|/\eta) \leq 1\}$. In contrast to the existent literature only care about heavy tail case $\theta < 1$, Lemma F.2 provides the uniform optimal concentration in sense of rate for any $\theta > 0$ with explicit constants. Before stating this lemma, we first give the equivalence of RGBO norm and concentration inequality stated as follows.

**Lemma F.1.** *For any zero-mean variable $X$ and $L > 0$, we have*

$$\|X\|_{\Psi_{\theta,3^{1/\theta-1/2}L}} \leq \sqrt{3}\tau \iff \mathbb{P}\left\{|X| > \tau\left(\sqrt{s} + Ls^{(1/\theta)\vee 1}\right)\right\} \leq 2\mathrm{e}^{-s}, \text{ for any } s \geq 0.$$

*Proof.* The proof is exactly the same as the proof of Lemma 1 and Lemma 2 in (van de Geer & Lederer, 2013). It is worthy

to note that the probability here can be rewritten as

$$\mathbb{P}\left\{|X| > \|X\|_{\Psi_{\theta,3^{1/\theta-1/2}L}} \Psi_{\theta,L}^{-1}(e^s - 1)/\sqrt{3}\right\} \le 2e^{-s}$$

for any $s \ge 0$ and hence

$$\mathbb{P}\left\{|X| \ge \ell \|X\|_{\theta,K}\right\} \le \frac{2}{1 + \Psi_{\theta,3^{1/2-1/\theta}K}(\sqrt{3}\ell)} \tag{F.1}$$

for any $\ell \ge 0$. $\qquad\qquad\qquad\qquad\qquad\qquad\qquad\qquad\qquad\qquad\qquad\qquad\qquad\square$

Under this lemma, another way to state Lemma 2.2 is using Bernstein-Orlicz norm (van de Geer & Lederer, 2013). As delineated in Lemma F.1, Lemma 2.2 can be reformulated as:

$$\left\|\mu - \overline{X}^*\right\|_{\Psi_{\theta,\frac{pE(\theta)}{2\sqrt{n}}}} \le \frac{4eD(\theta)C}{p\sqrt{n}}$$

with probability $1 - \delta/2$ for any $n \ge 4\log(2/\delta)/p^2$.

**Lemma F.2** (Sharper Sub-Weibull Concentrations). *Suppose $X_i - \mu \overset{i.i.d.}{\sim} \text{subW}(\theta; C)$, then for any $s \ge 0$,*

$$\left\|\mu - \overline{X}\right\|_{\Psi_{\theta,n^{-1/2}E(\theta)}} \le 2n^{-1/2}e^{-1}D(\theta)C,$$

*and*

$$\mathbb{P}\left\{|\mu - \overline{X}| > 2eD(\theta)C\left(\sqrt{\frac{s}{n}} + E(\theta)\frac{s^{(1/\theta)\vee 1}}{n}\right)\right\} \le 2e^{-s},$$

*where $D(\theta)$ and $E(\theta)$ are defined as*

$$D(\theta) = \begin{cases} (\sqrt{2} \vee 2^{1/\theta})\sqrt{8}e^3(2\pi)^{1/4}e^{1/24}\left(e^{2/e}/\theta\right)^{1/\theta}, & \text{if } \theta < 1, \\ \sqrt{3/(2e^2)}(C^{-1} \vee C^{\theta-1}), & \text{if } 1 \le \theta < 2, \\ \sqrt{17/(6e^2)}(C^{-1} \vee C^{\theta/2-1}), & \text{if } \theta \ge 2, \end{cases}$$

*and*

$$E(\theta) = \begin{cases} 2^{2/\theta-1/2}, & \text{if } \theta < 1, \\ 1/\sqrt{6}, & \text{if } 1 \le \theta < 2, \\ 0, & \text{if } \theta \ge 2. \end{cases}$$

*Proof.* We consider the case $\theta < 1$, $1 \le \theta < 2$, and $\theta \ge 2$ separately.

• If $\theta \ge 2$. From $\mathbb{E}\exp\left\{|X - \mu|^\theta/C^\theta\right\} \le 2$, we know that

$$\mathbb{E}\sum_{j=1}^{\infty} \frac{\left(|X - \mu|^\theta/C^\theta\right)^j}{j!} = \sum_{j=1}^{\infty} \frac{\mathbb{E}|X - \mu|^{j\theta}}{j!C^{j\theta}} \le 1. \tag{F.2}$$

This implies for any $k \in \mathbb{N}$, by $\theta \ge 2$,

$$\mathbb{E}|X - \mu|^{2k} \le 1 + \mathbb{E}|X - \mu|^{k\theta}$$

$$\overset{\text{by (F.2)}}{\le} 1 + k!C^{k\theta}$$

$$= 1 + k!(1 \vee C^{\theta/2})^{2k}$$

$$\overset{\text{by } k^k \times k! \le (2k)!}{\le} 1 + (1 \vee C^{\theta/2})^{2k} \times (2k-1)!!\frac{(2k)!!}{k^k}$$

$$\overset{\text{by Bohr–Mollerup theorem}}{\le} 1 + (1 \vee C^{\theta/2})^{2k} \times (2k-1)!! \times \frac{2^k k!}{k^k}$$

$$\le 1 + (2k-1)!! \times \left(\sqrt{2} \vee \sqrt{2}C^{\theta/2}\right)^{2k}$$

$$\le (2k-1)!! \times \left(2 \vee 2C^{\theta/2}\right)^{2k}.$$

Therefore, the sub-Gaussian intrinsic moment norm for $X - \mu$ satisfies $\|X - \mu\|_G \leq 2 \vee 2C^{\theta/2}$, and thus

$$\mathbb{P}\left(|\mu - \overline{X}| > \left(2 \vee 2C^{\theta/2}\right)\sqrt{\frac{17s}{6n}}\right) \leq \mathbb{P}\left(|\mu - \overline{X}| > \|X - \mu\|_G\sqrt{\frac{17s}{6n}}\right) \leq 2\mathrm{e}^{-s}$$

for any $s \geq 0$ by Theorem 2(b) in Zhang et al. (2023).

• If $1 \leq \theta < 2$, (F.2) still holds for $\theta \in [1, 2)$. We claim there exist positive $\nu$ and $\kappa$ such that

$$\mathbb{E}|X - \mu|^k \leq \frac{1}{2}\nu^2\kappa^{k-2}k!, \qquad k = 2, 3, \ldots. \tag{F.3}$$

Indeed, (F.2) together with $\theta \geq 1$ implies

$$\mathbb{E}|X - \mu|^k \leq 1 + \mathbb{E}|X - \mu|^{k\theta} \leq 1 + k!C^{k\theta}$$

Hence, a sufficient condition for (F.3) is $1 + k!C^{k\theta} \leq \frac{1}{2}\nu^2\kappa^{k-2}k!$ for any $k = 2, 3, \ldots$. We rewrite this as

$$\frac{\nu^2}{\kappa^2}\kappa^k \geq \frac{2}{k!} + 2C^{k\theta}, \qquad k = 2, 3, \ldots.$$

Therefore, we can take $\kappa = 1 \vee C^{\theta}$ and $\nu = \sqrt{3}\kappa$, and then $X_i - \mu \overset{\text{i.i.d.}}{\sim} \mathrm{sub}\Gamma(\nu, \kappa)$ by Lemma 2.2.11 in Vaart & Wellner (2023). We can then apply concentration for sub-Gamma distributions in Corollary 5.2 of Zhang & Chen (2021); Boucheron et al. (2013) and obtain that

$$\mathbb{P}\left(|\mu - \overline{X}| > \sqrt{6}(1 \vee C^{\theta})\sqrt{\frac{s}{n}} + (1 \vee C^{\theta})\frac{s}{n}\right) = \mathbb{P}\left(|\mu - \overline{X}| > \sqrt{2}\nu\sqrt{\frac{s}{n}} + \kappa\frac{s}{n}\right) \leq 2\mathrm{e}^{-s}.$$

• If $\theta < 1$. Denote $\beta$ is the conjugate of $\theta$, i.e., $\beta = \infty$. Then from Theorem 1 in Zhang & Wei (2022), we know that

$$\mathbb{P}\left(\left|\sum_{i=1}^{n} a_i(\mu - X_i)\right| \geq 2\mathrm{e}D(\theta)\|b\|_2\sqrt{s} + 2\mathrm{e}L_n^*(\theta)s^{1/\theta}\|b\|_\beta\right) \leq 2\mathrm{e}^{-s}$$

where $b = n^{-1}C1_n \in \mathbb{R}^n$ with $\|b\|_2 = Cn^{-1/2}$, $\|b\|_\beta = Cn^{-1}$, and $D(\theta) = (\sqrt{2} \vee 2^{1/\theta})\sqrt{8}\mathrm{e}^3(2\pi)^{1/4}\mathrm{e}^{1/24}\left(\mathrm{e}^{2/\mathrm{e}}/\theta\right)^{1/\theta}$, and $L_n^*(\theta) = L_n(\theta)D(\theta)\|b\|_2/\|b\|_\beta$ with $L_n(\theta) := \frac{4^{1/\theta}\|b\|_\beta}{\sqrt{2}\|b\|_2}$. Then we have

$$2\mathrm{e}D(\theta)\|b\|_2\sqrt{s} + 2\mathrm{e}L_n^*(\theta)s^{1/\theta}\|b\|_\beta = 2\mathrm{e}D(\theta)C\left(\sqrt{\frac{s}{n}} + E(\theta)\frac{s^{1/\theta}}{n}\right)$$

where $E(\theta) = \frac{4^{1/\theta}}{\sqrt{2}} = 2^{\frac{4-\theta}{2\theta}}$. Combining these results, we obtain the concentration inequality in the lemma. For the result of the RGBO norm, we just use the lemma F.1. $\qquad\square$

Lemma F.2 establishes a uniform result for the sample mean of i.i.d. sub-Weibull random variables. It is important to note that our result here is sharper than those in Kuchibhotla & Chakrabortty (2022) and Zhang & Wei (2022), especially for the case when $\theta \geq 1$, as they focus on general weighted summations. Another notable difference is in comparison to the sub-Weibull concentration results for sample means in Adamczak et al. (2011); Bogucki (2015); Hao et al. (2019), which require symmetry, while our approach does not. Consequently, we present a novel concentration result for the sample mean of i.i.d. sub-Weibull random variables.

Next, we will show some anti-concentrations for the posterior distributions in Algorithm C.1, which are essential for the proof of TS-type algorithms.

**Lemma F.3.** *For any $0 \leq x \leq \frac{\beta}{\alpha+\beta}$, we have*

$$\mathbb{P}\left(\mathrm{Beta}(\alpha, \beta) > \frac{\alpha}{\alpha + \beta} + x\right) \geq \frac{\Gamma(\beta + \alpha)}{\beta\Gamma(\beta)\Gamma(\alpha)}\left(\frac{\beta}{\alpha + \beta} - x\right)^\beta\left(\frac{\alpha}{\alpha + \beta} + x\right)^\alpha\left(\frac{\beta + 2}{\beta + 1} - \frac{\alpha + \beta}{\beta + 1}x\right).$$

*Proof.* First, we note that $\text{Beta}(\alpha, \beta) = 1 - \text{Beta}(\beta, \alpha)$. Then we can rewrite the probability as

$$\mathbb{P}\left(\text{Beta}(\alpha, \beta) > \frac{\alpha}{\alpha + \beta} + x\right)$$

$$=\mathbb{P}\left(1 - \text{Beta}(\beta, \alpha) > \frac{\alpha}{\alpha + \beta} + x\right)$$

$$=\mathbb{P}\left(\text{Beta}(\beta, \alpha) \leq 1 - \frac{\alpha}{\alpha + \beta} - x\right)$$

$$\overset{\text{by Theorem 1 in (Henzi \& Dümbgen, 2023)}}{\geq} \frac{\Gamma(\beta + \alpha)}{\beta\Gamma(\beta)\Gamma(\alpha)}\left(1 - \frac{\alpha}{\alpha + \beta} - x\right)^{\beta}\left(\frac{\alpha}{\alpha + \beta} + x\right)^{\alpha}\left[1 + \frac{\beta + \alpha}{\beta + 1}\left(1 - \frac{\alpha}{\alpha + \beta} - x\right)\right]$$

$$=\frac{\Gamma(\beta + \alpha)}{\beta\Gamma(\beta)\Gamma(\alpha)}\left(\frac{\beta}{\alpha + \beta} - x\right)^{\beta}\left(\frac{\alpha}{\alpha + \beta} + x\right)^{\alpha}\left(\frac{\beta + 2}{\beta + 1} - \frac{\alpha + \beta}{\beta + 1}x\right).$$

$\square$

**Lemma F.4.** *Suppose $\xi \sim \mathcal{N}(\mu, \sigma^2)$ and $\zeta \sim \text{Beta}(\alpha, \beta)$, then*

$$\mathbb{P}\left(\xi \times \zeta \geq \frac{\alpha\mu}{\alpha + \beta} + x\right)$$

$$\geq \begin{cases} \frac{1}{2}c(\alpha, \beta), & \text{if } x \leq \frac{\mu\beta}{2(\alpha + \beta)}, \\ \frac{(2\alpha + \beta)\sigma}{\sqrt{2\pi}} \frac{2(\alpha + \beta)x - \mu\beta}{[2(\alpha + \beta)x - \mu\beta]^2 + (2\alpha + \beta)^2\sigma^2} \exp\left\{-\frac{1}{2}\left(\frac{2(\alpha + \beta)x - \mu\beta}{(2\alpha + \beta)\sigma}\right)^2\right\}, & \text{if } x > \frac{\mu\beta}{2(\alpha + \beta)}, \end{cases}$$

*where*

$$c(\alpha, \beta) = \frac{\Gamma(\beta + \alpha)}{\beta\Gamma(\beta)\Gamma(\alpha)}\left[\frac{\beta}{2(\alpha + \beta)}\right]^{\beta}\left[\frac{2\alpha + \beta}{2(\alpha + \beta)}\right]^{\alpha}\left[\frac{\beta + 4}{2(\beta + 1)}\right].$$

*Proof.* First, we note that

$$\mathbb{P}\left(\xi \times \zeta \geq \frac{\alpha\mu}{\alpha + \beta} + x\right) = \mathbb{P}\left(\xi \times \zeta \geq (\mu + y)\left(\frac{\alpha}{\alpha + \beta} + z\right)\right)$$

$$\geq \mathbb{P}\left(\xi \geq \mu + y\right) \times \mathbb{P}\left(\zeta \geq \frac{\alpha}{\alpha + \beta} + z\right)$$

with $y, z$ defined as

$$y = \frac{2(\alpha + \beta)x - \mu\beta}{2\alpha + \beta}, \qquad z = \frac{\beta}{2(\alpha + \beta)}.$$

From Lemma F.3, we obtain

$$\mathbb{P}\left(\zeta \geq \frac{\alpha}{\alpha + \beta} + z\right) \geq \frac{\Gamma(\beta + \alpha)}{\beta\Gamma(\beta)\Gamma(\alpha)}\left(\frac{\beta}{\alpha + \beta} - z\right)^{\beta}\left(\frac{\alpha}{\alpha + \beta} + z\right)^{\alpha}\left(\frac{\beta + 2}{\beta + 1} - \frac{\alpha + \beta}{\beta + 1}z\right)$$

$$= \frac{\Gamma(\beta + \alpha)}{\beta\Gamma(\beta)\Gamma(\alpha)}\left[\frac{\beta}{2(\alpha + \beta)}\right]^{\beta}\left[\frac{2\alpha + \beta}{2(\alpha + \beta)}\right]^{\alpha}\left[\frac{\beta + 4}{2(\beta + 1)}\right] = c(\alpha, \beta)$$

with $c(\alpha, \beta)$ is a constant only depending on $\alpha$ and $\beta$. If $y \leq 0$, i.e., $x \leq \frac{\mu\beta}{2(\alpha + \beta)}$, then $\mathbb{P}\left(\xi \geq \mu + y\right) \geq \frac{1}{2}$, and thus

$$\mathbb{P}\left(\xi \times \zeta \geq \frac{\alpha\mu}{\alpha + \beta} + x\right) \geq \frac{c(\alpha, \beta)}{2}.$$

If $y > 0$, i.e., $x > \frac{\mu\beta}{2(\alpha+\beta)}$, then by Abramowitz et al. (1988),

$$
\begin{aligned}
\mathbb{P}\left(\xi \geq \mu + y\right) &= \mathbb{P}\left(\xi \geq \mu + \sigma \frac{2(\alpha+\beta)x - \mu\beta}{(2\alpha+\beta)\sigma}\right) \\
&\geq \frac{1}{\sqrt{2\pi}} \frac{\frac{2(\alpha+\beta)x-\mu\beta}{(2\alpha+\beta)\sigma}}{\left(\frac{2(\alpha+\beta)x-\mu\beta}{(2\alpha+\beta)\sigma}\right)^2 + 1} \exp\left\{-\frac{1}{2}\left(\frac{2(\alpha+\beta)x-\mu\beta}{(2\alpha+\beta)\sigma}\right)^2\right\} \\
&= \frac{(2\alpha+\beta)\sigma}{\sqrt{2\pi}} \frac{2(\alpha+\beta)x - \mu\beta}{[2(\alpha+\beta)x - \mu\beta]^2 + (2\alpha+\beta)^2\sigma^2} \exp\left\{-\frac{1}{2}\left(\frac{2(\alpha+\beta)x-\mu\beta}{(2\alpha+\beta)\sigma}\right)^2\right\},
\end{aligned}
$$

which leads to the final result. $\qquad\square$

The remaining part of this section will consist of the proofs of the lemmas presented in the main content.

**Proof of Lemma 2.1:**

*Proof.* From the definition of sub-Weibull distribution, we know that $\mathbb{E}\exp\{|X-\mu|^\theta/C_X^\theta\} \leq 2$. Note that for any $a, b \geq 0$: if $0 \leq \theta \leq 1$, $(a+b)^\theta \leq a^\theta + b^\theta$; if $\theta > 1$, $(a+b)^\theta \leq 2^{\theta-1}(a^\theta + b^\theta)$. Hence,

$$
(a+b)^\theta \leq \left(2^{\theta-1} \vee 1\right)(a^\theta + b^\theta).
$$

Thus, for any $C > 0$, we have

$$
\begin{aligned}
&\mathbb{E}\exp\{|R-\mu p|^\theta/C^\theta\} \\
&= \mathbb{E}\exp\left\{\left|(X-\mu)Y + \mu(Y-p)\right|^\theta/C^\theta\right\} \\
&\leq \mathbb{E}\exp\left\{\left(|X-\mu|Y + \mu|Y-p|\right)^\theta/C^\theta\right\} \\
&\leq \mathbb{E}\exp\left\{\left(2^{\theta-1} \vee 1\right)\left(|X-\mu|^\theta Y^\theta + \mu^\theta|Y-p|^\theta\right)/C^\theta\right\} \\
&\leq \mathbb{E}\exp\left\{\left(2^{\theta-1} \vee 1\right)\left(|X-\mu|^\theta + \mu^\theta(p^\theta + (1-p)^\theta)\right)/C^\theta\right\} \\
&= \exp\left\{\left(2^{\theta-1} \vee 1\right)\mu^\theta(p^\theta + (1-p)^\theta)/C^\theta\right\}\mathbb{E}\exp\left\{\left(2^{\theta-1} \vee 1\right)|X-\mu|^\theta/C^\theta\right\}.
\end{aligned}
$$

Since $\mathbb{E}\exp\{|X-\mu|^\theta/C_X^\theta\} \leq 2$, we have

$$
\begin{aligned}
0 &\leq \lim_{C\uparrow+\infty} \mathbb{E}\exp\{|R-\mu p|^\theta/C^\theta\} \\
&\leq \lim_{C\uparrow+\infty} \exp\left\{\left(2^{\theta-1} \vee 1\right)\mu^\theta(p^\theta + (1-p)^\theta)/C^\theta\right\} \lim_{C_X \leq C\uparrow+\infty} \mathbb{E}\exp\left\{\left(2^{\theta-1} \vee 1\right)|X-\mu|^\theta/C^\theta\right\} \\
&= 0.
\end{aligned}
$$

This implies there exists $C_R \geq C_X$ such that $\mathbb{E}\exp\{|R-\mu p|^\theta/C_R^\theta\} \leq 2$. $\qquad\square$

**Proof of Lemma E.1:**

*Proof.* The result that $R - \mu p$ is sub-Gaussian or sub-Exponential in the lemma directly comes from Lemma 2.1 by setting $\theta = 2$ and $\theta = 1$. For the second result, we first prove the results for sub-Gaussian case. Denote that $\tau^2$ as the minimal value which satisfies

$$
\mathbb{E}\exp\{s(R-\mu p)\} \leq \exp\{s^2\tau^2/2\}
$$

for any $s \in \mathbb{R}$. By the definition, we have

$$
\begin{aligned}
\mathbb{E}\exp\{s(R-\mu p)\} &= \mathbb{E}\exp\{s(XY-\mu p)\} \\
&= \mathbb{E}\exp\{s(0-\mu p)\}\mathbb{P}(Y=0) + \mathbb{E}\exp\{s(X-\mu p)\}\mathbb{P}(Y=1) \\
&= e^{-s\mu p}(1 - p + p\mathbb{E}e^{sX}).
\end{aligned}
$$

Since $\sigma^2$ is the minimal value such that $\mathbb{E}\exp\{s(X-\mu)\} \leq \exp\{s^2\sigma^2/2\}$ for all $s \in \mathbb{R}$, it also is the minimal value such that $\mathbb{E}e^{sX} \leq e^{s\mu+s^2\sigma^2/2}$. This indicates $\tau^2$ satisfies

$$e^{-s\mu p}(1 - p + pe^{s\mu+s^2\sigma^2/2}) \leq \exp\{s^2\tau^2/2\},$$

i.e.,

$$\tau^2 = \max_{s \in \mathbb{R}} \frac{2}{s^2}\left[-s\mu p + \log(1 - p + pe^{s\mu+s^2\sigma^2/2})\right].$$

Denote

$$f(s, \mu, p, \sigma^2) := \frac{2}{s^2}\left[-s\mu p + \log(1 - p + pe^{s\mu+s^2\sigma^2/2})\right]$$

with $\tau^2 = \max_{s \in \mathbb{R}} f(s, \mu, p, \sigma^2) = f(s_*, \mu, p, \sigma^2)$. We will first show that

$$\forall \mu > 0, \quad \lim_{p \downarrow 0} s_* = +\infty \quad \text{and} \quad \lim_{|\mu| \vee \sigma^2 \downarrow 0} \lim_{p \uparrow 1} s_* = 0+.$$

Indeed, $s_*$ satisfies $\frac{\partial}{\partial s} f(s, \mu, p, \sigma^2) = 0$, i.e.,

$$\frac{2\mu p}{s_*^2} - \frac{4}{s_*^3}\log(1 - p + pe^{s_*\mu+s_*^2\sigma^2/2}) + \frac{2}{s_*^2}\frac{p(\mu + \sigma^2 s_*)}{p + (1-p)e^{-s_*\mu-s_*^2\sigma^2/2}} = 0,$$

or say,

$$\frac{2}{p}\log(1 - p + pe^{s_*\mu+s_*^2\sigma^2/2}) = s_*\left[\frac{(\mu + \sigma^2 s_*)}{p + (1-p)e^{-s_*\mu-s_*^2\sigma^2/2}} + \mu\right]. \tag{F.4}$$

As we can see whenever $s_*$ is finite, we have $\lim_{p\downarrow 0}\frac{2}{p}\log(1 - p + pe^{s_*\mu+s_*^2\sigma^2/2}) = +\infty$, then there must be $\lim_{p\downarrow 0} s_* = +\infty$ since $\mu > 0$. On the other hand, by letting $p = 1$, the above equation becomes

$$\log(1 + e^{s_*\mu+s_*^2\sigma^2/2}) = s_*\mu + s_*^2\sigma^2/2.$$

Since $\lim_{x\to-\infty}\log(1 + e^x) - x = +\infty$, $\lim_{x\to 0}\log(1 + e^x) - x = \log 2$, we must have $\lim_{|\mu|\vee\sigma^2 \downarrow 0}\lim_{p\uparrow 1} s_* = 0+$. Now, consider

$$\frac{\tau^2}{\sigma^2} = \frac{2}{s_*^2\sigma^2}\left[-s_*\mu p + \log(1 - p + pe^{s_*\mu+s_*^2\sigma^2/2})\right]$$

$$= \frac{p}{p + (1-p)e^{-s_*\mu-s_*^2\sigma^2/2}} + \frac{\mu(1-p)p}{s\sigma^2}\frac{e^{s_*\mu+s_*^2\sigma^2/2} + 1}{pe^{s_*\mu+s_*^2\sigma^2/2} + 1 - p}$$

Then consider $\mu > 0$, by taking a fixed $\bar{p} \in (0, 1)$, and denote $\bar{s} = s_*(p = \bar{p}, \mu, \sigma^2)$ we get that

$$\left.\frac{\tau^2}{\sigma^2}\right|_{p=\bar{p}} = \frac{\bar{p}}{\bar{p} + (1-\bar{p})e^{-\bar{s}\mu-\bar{s}^2\sigma^2/2}} + \frac{\mu(1-\bar{p})\bar{p}}{\bar{s}\sigma^2}\frac{e^{\bar{s}\mu+\bar{s}^2\sigma^2/2} + 1}{\bar{p}e^{\bar{s}\mu+\bar{s}^2\sigma^2/2} + 1 - \bar{p}}$$

$$\geq \frac{\mu(1-\bar{p})\bar{p}}{\bar{s}\sigma^2}\frac{e^{\bar{s}\mu+\bar{s}^2\sigma^2/2} + 1}{\bar{p}e^{\bar{s}\mu+\bar{s}^2\sigma^2/2} + 1 - \bar{p}}$$

Since for any $x \geq 0$ and $p \in (0, 1)$

$$x = \log e^x \geq \log(1 - p + pe^x) = \log(1 - p) + \log\left(1 + \frac{p}{1-p}e^x\right) \geq \log(1 - p) + \log\left(\frac{p}{1-p}e^x\right) = x - \log p,$$

and note that $\bar{s}$ satisfies (F.4), we have

$$\bar{s}\left[\frac{(\mu + \sigma^2\bar{s})}{\bar{p} + (1-\bar{p})e^{-\bar{s}\mu-\bar{s}^2\sigma^2/2}} + \mu\right] = \frac{2}{\bar{p}}\log(1 - \bar{p} + \bar{p}e^{\bar{s}\mu+\bar{s}^2\sigma^2/2})$$

$$\overset{\text{by the inequality above}}{\in} \frac{2}{\bar{p}}\left[\bar{s}\mu + \bar{s}^2\sigma^2/2 + \log\bar{p}, \ \bar{s}\mu + \bar{s}^2\sigma^2/2\right].$$

Note that the left hand above is less than

$$\overline{s}\left[\frac{(\mu + \sigma^2\overline{s})}{\overline{p} + (1-\overline{p})e^{-\overline{s}\mu - \overline{s}^2\sigma^2/2}} + \mu\right] \le \overline{s}\left[\frac{(\mu + \sigma^2\overline{s})}{\overline{p}} + \mu\right] = (1 + \overline{p}^{-1})\overline{s}\mu + \overline{p}^{-1}\overline{s}^2\sigma^2,$$

while the right hand above is larger than

$$\frac{2}{\overline{p}}\log(1 - \overline{p} + \overline{p}e^{\overline{s}\mu + \overline{s}^2\sigma^2/2}) \ge \frac{2(\overline{s}\mu + \overline{s}^2\sigma^2/2 + \log\overline{p})}{\overline{p}} = 2\overline{p}^{-1}\overline{s}\mu + 2\overline{p}^{-1}\log\overline{p} + \overline{p}^{-1}\overline{s}^2\sigma^2.$$

Since for any fixed $\overline{p} \in (0,1)$, we have $\overline{p}^{-1} > 1$. Thus, we must have $\lim_{\mu \to +\infty} \overline{s} = 0$, or

$$\lim_{\mu \to +\infty}\left\{\frac{2}{\overline{p}}\log(1 - \overline{p} + \overline{p}e^{\overline{s}\mu + \overline{s}^2\sigma^2/2}) - \overline{s}\left[\frac{(\mu + \sigma^2\overline{s})}{\overline{p} + (1-\overline{p})e^{-\overline{s}\mu - \overline{s}^2\sigma^2/2}} + \mu\right]\right\}$$
$$\ge \lim_{\mu \to +\infty}\left(\overline{p}^{-1} - 1\right)\overline{s}\mu + 2\overline{p}^{-1}\log\overline{p} > 0,$$

which leads to a contradiction on the equation (F.4). Therefore, we must have

$$\lim_{\mu \to +\infty}\left.\frac{\tau^2}{\sigma^2}\right|_{p=\overline{p}} \ge \lim_{\mu \to +\infty}\frac{\mu(1-\overline{p})\overline{p}}{\overline{s}\sigma^2}\frac{e^{\overline{s}\mu + \overline{s}^2\sigma^2/2} + 1}{\overline{p}e^{\overline{s}\mu + \overline{s}^2\sigma^2/2} + 1 - \overline{p}} = \lim_{\mu \to +\infty}\frac{2\mu(1-\overline{p})\overline{p}}{\overline{s}\sigma^2} = +\infty,$$

which concludes the results in (i). Then we will prove the results in (iii). By envelope theorem,

$$\frac{\partial\tau^2}{\partial p} = \left.\frac{\partial}{\partial p}f(s, \mu, p, \sigma^2)\right|_{s=s^*} = \frac{2}{s_*^2}\left[-s_*\mu + \frac{e^{s_*\mu + s_*^2\sigma^2/2} - 1}{1 + p\left(e^{s_*\mu + s_*^2\sigma^2/2} - 1\right)}\right].$$

The fact that $\frac{e^x}{1+e^x}$ is bounded on $x \in [0, \infty)$ and the fact $\lim_{|\mu|\vee\sigma^2 \downarrow 0}\lim_{p\uparrow 1} s_* = 0+$ ensure that

$$\lim_{|\mu|\vee\sigma^2 \downarrow 0}\lim_{p\uparrow 1}\frac{\partial\tau^2}{\partial p} = \lim_{|\mu|\vee\sigma^2 \downarrow 0}\lim_{p\uparrow 1}\frac{2}{s_*^2}[-s_*\mu + 1] = +\infty.$$

By the continuity of $f(s^*, \cdot, \cdot, \cdot)$ on $\mu, p$, and $\sigma^2$, we get the first two results in (iii). Similarly, we can show that

$$\frac{\partial\tau^2}{\partial\mu} = \frac{2}{s_*}\left[-p + \frac{pe^{s_*\mu + s_*^2\sigma^2/2}}{1 + p\left(e^{s_*\mu + s_*^2\sigma^2/2} - 1\right)}\right],$$

which ensures $\lim_{\sigma^2 \downarrow 0}\lim_{p\uparrow 1}\frac{\partial\tau^2}{\partial\mu} = +\infty$ the last result in (iii). Finally, for the result in (ii), we note that

$$\text{var}(R) = \mathbb{E}\left[(XY - \mu p)^2\right] = p\,\text{var}(X) + p(1-p)\mu^2.$$

Then by $\sigma^2 \ge \text{var}(X)$, we have

$$\frac{\tau^2}{\text{var}(R)} = \frac{\tau^2}{p\,\text{var}(X) + p(1-p)\mu^2} \ge \frac{\tau^2}{p\sigma^2 + p(1-p)\mu^2}.$$

Take $\mu = c\mu_*$ with arbitrary $c > 1$, by letting $\sigma^2 \downarrow 0$ and $p \uparrow 1$, we have

$$\left.\frac{\tau^2}{\text{var}(R)}\right|_{\mu=c\mu_*} \ge \left.\frac{\tau^2}{p\sigma^2 + p(1-p)\mu^2}\right|_{\mu=c\mu_*}$$

$$\overset{\tau^2|_{\mu=\mu^*}>0}{\ge} \frac{\min_{\mu'\in[\mu_*,c\mu_*]}\left.\frac{\partial\tau^2}{\partial\mu}\right|_{\mu=\mu'}(c\mu_* - \mu_*)}{p\sigma^2 + p(1-p)c^2\mu_*^2}$$

$$\overset{\text{by the result in (iii) (c)}}{\ge} \frac{(c-1)M\mu_*}{p\sigma^2 + p(1-p)c^2\mu_*^2} \uparrow +\infty$$

which gives the result in (ii). For the case that $X - \mu \sim \text{subE}(\lambda)$, one only need to note that the sub-Exponential parameter $\alpha$ for $R - \mu p$ satisfies

$$\mathrm{e}^{-s\mu p}(1 - p + p\mathrm{e}^{s\mu + s^2\lambda^2/2}) \leq \exp\{s^2\alpha^2/2\}$$

for any $s \leq \frac{1}{\lambda}$, which implies

$$\alpha^2 = \lambda^2 \vee \max_{s \in \mathbb{R}} \frac{2}{s^2}\left[-s\mu p + \log(1 - p + p\mathrm{e}^{s\mu + s^2\lambda^2/2})\right].$$

Since $\lambda^2 \vee g(\lambda, \mu, p)$ with differential $g(\cdot, \cdot, \cdot)$ is also differential on its domain except the points that $\lambda^2 = g(\lambda, \mu, p)$, the above results regarding large values will still hold. Thus, we finish the proof. $\qquad\square$

**Proof of Lemma 2.2:**

*Proof.* Denote $B \sim \text{binomial}(n; p)$ independent with $\{X_t\}_{t=1}^n$, we consider the positive $p/2 > 0$, by concentration for Bernoulli, we have

$$\mathbb{P}(B \geq pn/2) = 1 - \mathbb{P}(p - \overline{B} \geq p/2)$$
$$\geq 1 - \exp\left[-np^2/4\right].$$

Given any $\delta > 0$, the above inequality ensures

$$B \geq pn/2$$

with probability at least $1 - \delta/2$ for any $n \geq \frac{4}{p^2}\log(2/\delta)$. Now, denote $X_k$ as the $k$-th observed $X_t$. Consider $s > 0$ which will be determined later,

$$\mathbb{P}\left\{|\mu - \overline{X}^*| > 2\mathrm{e}D(\theta)C_X\left(\sqrt{\frac{s}{np/2}} + E(\theta)\frac{s^{(1/\theta)\vee 1}}{np/2}\right)\right\}$$

$$=\mathbb{E}_B\left[\mathbb{P}_X\left\{\left|\mu - \frac{1}{B}\sum_{k=1}^B X_k\right| > 2\mathrm{e}D(\theta)C_X\left(\sqrt{\frac{s}{np/2}} + E(\theta)\frac{s^{(1/\theta)\vee 1}}{np/2}\right)\right\}\right]$$

$$\leq\mathbb{E}_B\left[\mathbb{P}_X\left\{\left|\mu - \frac{1}{B}\sum_{k=1}^B X_k\right| > 2\mathrm{e}D(\theta)C_X\left(\sqrt{\frac{s}{np/2}} + E(\theta)\frac{s^{(1/\theta)\vee 1}}{np/2}\right), B \geq np/2\right\}\right] + \frac{\delta}{2}$$

$$\leq\mathbb{E}_B\left[\mathbb{P}_X\left\{\left|\mu - \frac{1}{B}\sum_{k=1}^B X_k\right| > 2\mathrm{e}D(\theta)C_X\left(\sqrt{\frac{s}{B}} + E(\theta)\frac{s^{(1/\theta)\vee 1}}{B}\right)\right\}\right] + \frac{\delta}{2}$$

$$\leq 2\mathrm{e}^{-s} + \frac{\delta}{2},$$

where the last step is by Lemma F.2. Finally, by letting $2\mathrm{e}^{-s} = \delta/2$, i.e., $s = \log(4/\delta)$, we conclude the inequality in the lemma. $\qquad\square$

**Proof of Lemma B.1:**

*Proof.* Denote

$$M_k = \left(\frac{kM}{\log z}\right)^{\frac{1}{1+\epsilon}}$$

with $z$ will be determined later. The proof idea comes from Lemma 1 in Bubeck et al. (2013). Denote $X_k$ as the $k$-th

observed $X_t$. Consider some positive $s$,

$$\mathbb{P}\big(\mu - \overline{X}^{**} > s\big)$$

$$=\mathbb{E}_B\left[\mathbb{P}_X\left(\mu - \frac{1}{B}\sum_{k=1}^{B} X_k \mathbb{1}(|X_k| \le M_k) > t\right)\right]$$

$$\le \mathbb{E}_B\left[\mathbb{P}_X\left(\frac{1}{B}\sum_{k=1}^{B}\mathbb{E}X\mathbb{1}(|X| > M_k) + \frac{1}{B}\sum_{k=1}^{B}\Big[\mathbb{E}X\mathbb{1}(|X| \le M_k) - X_k\mathbb{1}(|X_k| \le M_k)\Big] > s\right)\right]$$

$$\le \mathbb{E}_B\left[\mathbb{P}_X\left(\frac{1}{B}\sum_{k=1}^{B}\frac{M}{M_k^\epsilon} + \frac{1}{B}\sum_{k=1}^{B}\Big[\mathbb{E}X\mathbb{1}(|X| \le M_k) - X_k\mathbb{1}(|X_k| \le M_k)\Big] > s\right)\right]$$

$$\le \mathbb{E}_B\left[\mathbb{P}_X\left(\frac{(1+\epsilon)M^{\frac{1}{1+\epsilon}}\log^{\frac{\epsilon}{1+\epsilon}} z}{B^{\frac{\epsilon}{1+\epsilon}}} + \frac{1}{B}\sum_{k=1}^{B}\Big[\mathbb{E}X\mathbb{1}(|X| \le M_k) - X_k\mathbb{1}(|X_k| \le M_k)\Big] > s\right)\right]$$

where $B = \sum_{i=1}^{n} Y_i \sim \text{binomial}(n; p)$ is independent with $\{X_i\}_{i=1}^{n}$. The last inequality is using

$$\frac{1}{B}\sum_{k=1}^{B}\frac{M}{M_k^\epsilon} = M^{\frac{1}{1+\epsilon}}\log^{\frac{\epsilon}{1+\epsilon}} z \frac{1}{B}\sum_{k=1}^{B} s^{-\frac{\epsilon}{1+\epsilon}}$$

$$\le M^{\frac{1}{1+\epsilon}}\log^{\frac{\epsilon}{1+\epsilon}} z \frac{1}{B}\int_0^B s^{-\frac{\epsilon}{1+\epsilon}}\,\mathrm{d}s$$

$$= (1+\epsilon)M^{\frac{1}{1+\epsilon}}\log^{\frac{\epsilon}{1+\epsilon}} z \times B^{-\frac{\epsilon}{1+\epsilon}}.$$

Next, we consider the positive $p/2 > 0$, by concentration for Bernoulli, we have

$$\mathbb{P}(B \ge pn/2) = 1 - \mathbb{P}(p - \overline{B} \ge p/2)$$

$$\ge 1 - \exp\big[-np^2/4\big].$$

Given any $\delta$, the above inequality ensures

$$B \ge pn/2$$

with probability at least $1 - \delta/2$ for any $n \ge \frac{4}{p^2}\log(2/\delta)$. Then by Bernstein's inequality

$$\mathbb{P}\big(\mu - \overline{X}^{**} > s\big)$$

$$\le \mathbb{E}_B\left[\mathbb{P}_X\left(\frac{1}{B}\sum_{k=1}^{B}\Big[\mathbb{E}X\mathbb{1}(|X| \le M_k) - X_k\mathbb{1}(|X_k| \le M_k)\Big] > s - \frac{(1+\epsilon)M^{\frac{1}{1+\epsilon}}\log^{\frac{\epsilon}{1+\epsilon}} z}{B^{\frac{\epsilon}{1+\epsilon}}}\right)\right]$$

$$\le \mathbb{E}_B\left[\mathbb{P}_X\left(\frac{1}{B}\sum_{k=1}^{B}\Big[\mathbb{E}X\mathbb{1}(|X| \le M_k) - X_k\mathbb{1}(|X_k| \le M_k)\Big] > s - \frac{(1+\epsilon)M^{\frac{1}{1+\epsilon}}\log^{\frac{\epsilon}{1+\epsilon}} z}{B^{\frac{\epsilon}{1+\epsilon}}}, B \ge pn/2\right)\right] + \frac{\delta}{2}$$

$$\le \mathbb{E}_B\left[\mathbb{P}_X\left(\frac{1}{B}\sum_{k=1}^{B}\Big[\mathbb{E}X\mathbb{1}(|X| \le M_k) - X_k\mathbb{1}(|X_k| \le M_k)\Big] > s - \frac{(1+\epsilon)M^{\frac{1}{1+\epsilon}}\log^{\frac{\epsilon}{1+\epsilon}} z}{(pn/2)^{\frac{\epsilon}{1+\epsilon}}}, B \ge pn/2\right)\right] + \frac{\delta}{2}$$

$$\le \mathbb{E}_B\left[\exp\left(-\frac{B\left(s - \frac{(1+\epsilon)M^{\frac{1}{1+\epsilon}}\log^{\frac{\epsilon}{1+\epsilon}} z}{(pn/2)^{\frac{\epsilon}{1+\epsilon}}}\right)^2/2}{MM_k^{1-\epsilon} + M_k\left(s - \frac{(1+\epsilon)M^{\frac{1}{1+\epsilon}}\log^{\frac{\epsilon}{1+\epsilon}} z}{(pn/2)^{\frac{\epsilon}{1+\epsilon}}}\right)/3}\right)\mathbb{1}(B \ge pn/2)\right] + \frac{\delta}{2}$$

$$\le \exp\left(-\frac{pn\left(s - \frac{(1+\epsilon)M^{\frac{1}{1+\epsilon}}\log^{\frac{\epsilon}{1+\epsilon}} z}{(pn/2)^{\frac{\epsilon}{1+\epsilon}}}\right)^2/4}{MM_n^{1-\epsilon} + M_n\left(s - \frac{(1+\epsilon)M^{\frac{1}{1+\epsilon}}\log^{\frac{\epsilon}{1+\epsilon}} z}{(pn/2)^{\frac{\epsilon}{1+\epsilon}}}\right)/3}\right) + \frac{\delta}{2}$$

By letting

$$s - \frac{(1+\epsilon)M^{\frac{1}{1+\epsilon}}\log^{\frac{\epsilon}{1+\epsilon}}z}{(pn/2)^{\frac{\epsilon}{1+\epsilon}}} = \frac{4\log(2/\delta)}{3pn}M_n + \sqrt{\frac{4MM_n^{1-\epsilon}\log(2/\delta)}{pn}},$$

by $\frac{s^2/2}{\sigma^2 + Ms/3} = \frac{A}{n} \iff s = \frac{AM}{3n} \pm \sqrt{\frac{A^2M^2}{9n^2} + \frac{2A\sigma^2}{n}} \le \frac{2AM}{3n} + \sqrt{\frac{2A\sigma^2}{n}}$. Let $z = \log(2/\delta)$, we have

$$\begin{aligned}
s &= \frac{(1+\epsilon)M^{\frac{1}{1+\epsilon}}\log^{\frac{\epsilon}{1+\epsilon}}(2/\delta)}{(pn/2)^{\frac{\epsilon}{1+\epsilon}}} + \frac{4\log(2/\delta)}{3pn}M_n + \sqrt{\frac{4MM_n^{1-\epsilon}\log(2/\delta)}{pn}} \\
&= \frac{(1+\epsilon)2^{\frac{\epsilon}{1+\epsilon}}M^{\frac{1}{1+\epsilon}}}{p^{\frac{\epsilon}{1+\epsilon}}}\left(\frac{\log(2/\delta)}{n}\right)^{\frac{\epsilon}{1+\epsilon}} + \frac{4M^{\frac{1}{1+\epsilon}}}{3p}\left(\frac{\log(2/\delta)}{n}\right)^{\frac{\epsilon}{1+\epsilon}} + \frac{2M^{\frac{1}{1+\epsilon}}}{\sqrt{p}}\left(\frac{\log(2/\delta)}{n}\right)^{\frac{\epsilon}{1+\epsilon}} \\
&= \left[\frac{(1+\epsilon)2^{\frac{\epsilon}{1+\epsilon}}}{p^{\frac{\epsilon}{1+\epsilon}}} + \frac{4}{3p} + \frac{2}{\sqrt{p}}\right]M^{\frac{1}{1+\epsilon}}\left(\frac{\log(2/\delta)}{n}\right)^{\frac{\epsilon}{1+\epsilon}},
\end{aligned}$$

which leads to the result. $\qquad\square$

**Proof of Lemma 6.1:**

*Proof.* From Theorem 16.2 in Lattimore & Szepesvári (2020), an algorithm is deemed asymptotically optimal for problem-dependent regret if it satisfies:

$$\liminf_{T\to+\infty}\frac{\mathcal{R}(T)}{\log T} = \sum_{k=2}^{K}\frac{\Delta_k}{d_{\inf}(P_k, r_1, \mathcal{M}_k)}.$$

where $d_{\inf}(P, r, \mathcal{M}) = \inf_{P'\in\mathcal{M}}\{\text{KL}(P, P') : \mathbb{E}_{R\sim P'}R > r\}$. Here the model class $\mathcal{M}_k = \mathcal{X}_k \times \mathcal{Y}_k$ is ZI structure such that

$$\mathcal{X}_k = \{X - \mu_k \sim \text{subG}(\sigma^2) : \mathbb{P}(X = 0) = 0\} \quad\text{and}\quad \mathcal{Y}_k = \{Y \sim \text{Bernoulli}(p_k) : p_k \in (0, 1)\}, \quad\text{where } X \perp\!\!\!\perp Y.$$

To prove the first part of the theorem, let us choose a subclass $\mathcal{X}_k^* \subset \mathcal{X}_k$ such that $\mathcal{X}_k^* = \{X \sim \mathcal{N}(\mu_k, \sigma^2) : \mu_k \in \mathbb{R}\}$. Denote $P_k := \mathcal{N}(\mu_k, \sigma^2) \times \text{Ber}(p_k) \in \mathcal{M}_k^* := \mathcal{X}_k^* \times \mathcal{Y}_k$. The remaining task is to calculate

$$d_{\inf}(P_k, r_1, \mathcal{M}_k^*) = \inf_{\mu_k, p_k : \mu_k p_k > r_1}\text{KL}(P_1, P_k).$$

Here we first note that the independence between $\mathcal{X}_k^*$ and $\mathcal{Y}_k$ implies

$$\begin{aligned}
\text{KL}(P_1, P_k) &= \text{KL}\{\mathcal{N}(\mu_1, \sigma^2), \mathcal{N}(\mu_k, \sigma^2)\} + \text{KL}\{\text{Ber}(p_1), \text{Ber}(p_k)\} \\
&= \frac{(\mu_1 - \mu_k)^2}{2\sigma^2} + p_k\log(p_k/p_1) + (1 - p_k)\log\left(\frac{1 - p_k}{1 - p_1}\right).
\end{aligned}$$

Then consider the restriction $\mu_1 p_1 > \mu_k p_k$, there are two cases:

• If $\mu_1 \le r_1/p_k$. Then we can let $p_k = p_1$ and then $\mu_k < r_1/p_1 = r_1/p_k$ suffices to satisfy the constraint. In this case, $\text{KL}\{\text{Ber}(p_1), \text{Ber}(p_k)\} = 0$, and thus

$$\inf_{\mu_k, p_k : \mu_k p_k > r_1}\text{KL}(P_1, P_k) = \inf_{\mu_k < r_1/p_k = \mu_1}\frac{(\mu_1 - \mu_k)^2}{2\sigma^2} = \frac{(\mu_k - r_1/p_k)^2}{2\sigma^2}.$$

• If $\mu_1 > r_1/p_k$, we let $\mu_k = \mu_1$ and then $p_k < r_1/\mu_1 = r_1/\mu_k$ satisfies the constraint. Similarly, in this case

$$\begin{aligned}
\inf_{\mu_k, p_k : \mu_k p_k > r_1}\text{KL}(P_1, P_k) &= \inf_{p_k < r_1/\mu_1 = r_1/\mu_k = p_1}\left(p_k\log(p_k/p_1) + (1 - p_k)\log\left(\frac{1 - p_k}{1 - p_1}\right)\right) \\
&= p_k\log\left(\frac{p_k}{r_1/\mu_k}\right) + (1 - p_k)\log\left(\frac{1 - p_k}{1 - r_1/\mu_k}\right).
\end{aligned}$$

By combining the two cases, we complete the proof for the first part in Lemma 6.1.

For proving the second argument in the lemma, we note that $P_k := \mathcal{N}(\mu_k, \sigma^2) \times \text{Ber}(p_k) \in \mathcal{M}_k$. which implies

$$d_{\inf}(P_k, r_1, \mathcal{M}_k) \leq \inf_{\mu_k, p_k : \mu_k p_k > r_1} \text{KL}(P_1, P_k).$$

Thus, it suffices to bound the infimum derived in the first part. For the Gaussian part, it is straightforward to see that

$$\inf_{\mu_k < r_1/p_k} \frac{(\mu_k - r_1/p_k)^2}{2\sigma^2} = \frac{(\mu_k - r_1/p_k)^2}{4\sigma^2} = \frac{\Delta_k^2}{4p_k^2 \sigma^2}.$$

For the Bernoulli part, applying the Pinsker's inequality, we get

$$\inf_{p_k < r_1/\mu_k : \mu_k = \mu_1} \text{KL}\left\{ \text{Ber}(p_1), \text{Ber}(p_k) \right\} = \inf_{p_k < r_1/\mu_k : \mu_k = \mu_1} p_k \log\left( \frac{p_k}{r_1/\mu_k} \right) + (1 - p_k) \log\left( \frac{1 - p_k}{1 - r_1/\mu_k} \right)$$

$$\leq \inf_{p_k < r_1/\mu_k : \mu_k = \mu_1} \frac{(p_k - r_1/\mu_k)^2}{(r_1/\mu_k) \wedge (1 - r_1/\mu_k)}$$

$$\leq \inf_{p_k < r_1/\mu_k : \mu_k = \mu_1} \frac{(r_k - r_1)^2}{\mu_k^2 (r_1/\mu_k)} + \inf_{p_k < r_1/\mu_k : \mu_k = \mu_1} \frac{(r_k - r_1)^2}{\mu_k^2 (1 - r_1/\mu_k)}.$$

Next, we bound the above two terms by

$$\inf_{p_k < r_1/\mu_k : \mu_k = \mu_1} \frac{(r_k - r_1)^2}{\mu_k^2 (r_1/\mu_k)} \leq \inf_{p_k < r_1/\mu_k : \mu_k = \mu_1} \frac{\Delta_k^2}{\mu_1^2 (r_1/\mu_1)}$$

$$= \inf_{p_k < r_1/\mu_k : \mu_k = \mu_1} \frac{\Delta_k^2}{r_1 \mu_k}$$

$$\leq \inf_{p_k < r_1/\mu_k : \mu_k = \mu_1} \frac{\Delta_k^2 p_k}{r_1 \mu_k (r_1/\mu_k)} = \frac{\Delta_k^2 p_k}{r_1^2}$$

and

$$\inf_{p_k < r_1/\mu_k : \mu_k = \mu_1} \frac{(r_k - r_1)^2}{\mu_k^2 (1 - r_1/\mu_k)} \leq \inf_{p_k < r_1/\mu_k : \mu_k = \mu_1} \frac{\Delta_k^2}{\mu_k (\mu_k - r_1)}$$

$$= \inf_{p_k < r_1/\mu_k : \mu_k = \mu_1} \frac{\Delta_k^2}{\mu_k (\mu_1 - r_1)}$$

$$\leq \inf_{p_k < r_1/\mu_k : \mu_k = \mu_1} \frac{\Delta_k^2 p_k^2}{\mu_k (\mu_1 - r_1)(r_1/\mu_k)(r_1/\mu_1)}$$

$$= \frac{\Delta_k^2 p_k^2}{r_1^2 (1 - p_1)}.$$

Finally, combining these bounds for both the Gaussian and Bernoulli components, we observe that the dominant terms depend on $p_k, \Delta_k \in (0, 1)$. The asymptotic regret bound is confirmed as

$$\sum_{k=2}^{K} \frac{\Delta_k}{d_{\inf}(P_k, r_1, \mathcal{M}_k)} \gtrsim \sum_{k=2}^{K} \left( \frac{\Delta_k}{\frac{\Delta_k^2}{4p_k^2}} + \frac{\Delta_k}{\frac{\Delta_k^2 p_k}{2r_1^2}} + \frac{\Delta_k}{\frac{\Delta_k^2 p_k^2}{2r_1^2(1-p_1)}} \right)$$

$$= \sum_{k=2}^{K} \left( \frac{p_k^2}{\Delta_k} + \frac{1}{p_k \Delta_k} + \frac{1}{p_k^2 \Delta_k} \right)$$

$$\gtrsim \sum_{k=2}^{K} \left( \frac{p_k^2}{\Delta_k} + \frac{1}{p_k \Delta_k} \right),$$

which concludes the proof for the second part of the lemma.

$\square$

# G. Proof of the regrets for UCB-type algorithms

The proofs for our UCB-type algorithms also follow the standard approach used in UCB algorithms, which involves controlling two probabilities. The first probability relates to the underestimation of the optimal arm, characterized by $\mathbb{P}(U_1^\mu(t) \times U_1^p(t) < r_1)$, and this can be easily managed using the concentration results presented in Section 2. The second probability concerns the overestimation of suboptimal arms, characterized by $\mathbb{P}(U_k^\mu(t) \times U_k^p(t) > r_1) = \mathbb{P}(U_k^\mu(t) \times U_k^p(t) > r_k + \Delta_k)$. Since $\Delta_k > 0$, the sharp properties of our concentration results in Section 2 also controls this probability, ensuring an exponential decay rate over rounds.

## G.1. Proof of Theorem 4.1

*Proof.* For any $\delta > 0$, denote the upper confidence bound for $p_k$ until round $t$ as $U_k^p(t, \delta) := \widehat{p}_k(t) + \sqrt{\frac{\log(2/\delta)}{2c_k(t)}}$, with $\widehat{p}_k(t)$ be the point estimate at round $t$. Based on the estimated $\widehat{p}_k(t)$, define the upper confidence bound for $\mu_k$ as

$$U_k^\mu(t, \delta) := \widehat{\mu}_k(t) + 2\mathrm{e}D(\theta)C\left(\sqrt{\frac{\log(4/\delta)}{c_k(t)\widehat{p}_k(t)/2}} + E(\theta)\frac{\log^{(1/\theta)\vee 1}(4/\delta)}{c_k(t)\widehat{p}_k(t)/2}\right)$$

with $\widehat{\mu}_k(t)$ be the point estimate again. For simplicity, we also denote $\widehat{p}_k(t)$ as $\widehat{p}_{k,m}$ when $c_k(t) = m$, and similarly define $\widehat{\mu}_{k,m}$. Similarly, we denote $U_k^p(t, \delta)$ as $U_{k,m}^p(\delta)$ when $c_k(t) = m$, and similarly define $U_{k,m}^\mu(\delta)$.

Now we can define good events as follows

$$\mathcal{G}^0 := \{r_1 < \min_{t \in \{m_1, m_1+1..., T\}} U_{1,m_1}^\mu(\delta) \times U_{1,m_1}^p(\delta)\}$$

and

$$\mathcal{G}_k^r = \{U_{k,m_k}^\mu(\delta) \times U_{k,m_k}^p(\delta) < r_1\}.$$

Furthermore, define $\mathcal{G}_k^p = \{\widehat{p}_{k,m_k} > p_k - \epsilon_k\}$ for $k \in [K]$ and $\mathcal{G}_k = \mathcal{G}^0 \cap \mathcal{G}_k^r \cap \mathcal{G}_1^p \cap \mathcal{G}_k^p$ for $k \neq 1$, where $\epsilon_k$ will be determined later.

**Step 1:** For bounding $\mathbb{P}(\mathcal{G}_k^c)$, we use the inequality that

$$\mathbb{P}(A \cup B) = \mathbb{P}(A) + \mathbb{P}(B) - \mathbb{P}(A \cap B) = \mathbb{P}(A \cap B^c) + \mathbb{P}(B).$$

Then we can decompose

$$\mathbb{P}(\mathcal{G}_k^c) \leq \mathbb{P}(\mathcal{G}^{0c} \cup \mathcal{G}_1^{pc}) + \mathbb{P}(\mathcal{G}_k^{rc} \cup \mathcal{G}_k^{pc})$$
$$= \mathbb{P}((\mathcal{G}^0)^c \cap \mathcal{G}_1^p) + \mathbb{P}((\mathcal{G}_1^p)^c) + \mathbb{P}((\mathcal{G}_k^r)^c \cap \mathcal{G}_k^p) + \mathbb{P}((\mathcal{G}_k^p)^c).$$

with $\mathbb{P}((\mathcal{G}_1^p)^c)$ and $\mathbb{P}((\mathcal{G}_k^p)^c)$ bounding easily. Indeed, denote $d(p_1, p_2)$ as the KL-divergence between two Bernoulli distributions of probability $p_1$ and $p_2$, then

$$\mathbb{P}((\mathcal{G}_1^p)^c) = \mathbb{P}(\widehat{p}_{1,m_1} \leq p_1 - \epsilon_1) \leq \exp(-m_1 d(p_1, p_1 - \epsilon_1))$$

and similarly $\mathbb{P}((\mathcal{G}_k^p)^c) \leq \exp(-m_k d(p_k, p_k - \epsilon_k))$. For another two terms, we first consider to decompose the sample space as

$$\Omega = \{\widehat{p}_{1,m_1} \geq p_1 + \epsilon_1'\} \cup \{\widehat{p}_{1,m_1} < p_1 + \epsilon_1'\}$$

with $\epsilon_1' > 0$ will be determined later. Then

$$\mathbb{P}((\mathcal{G}^0)^c \cap \mathcal{G}_1^p)$$
$$\leq \mathbb{P}(\Omega \cap (\mathcal{G}^0)^c \cap \mathcal{G}_1^p)$$
$$\leq \mathbb{P}(\widehat{p}_{1,m_1} \geq p_1 + \epsilon_1') + \mathbb{P}(\widehat{p}_{1,m_1} \leq p_1 + \epsilon_1', r_1 \geq U_{1,m_1}^\mu(\delta) \times U_{1,m_1}^p(\delta), \widehat{p}_{1,m_1} > p_1 - \epsilon_1)$$
$$\leq \exp(-m_1 d(p_1 + \epsilon_1', p_1)) + \mathbb{P}(r_1 \geq U_{1,m_1}^\mu(\delta) \times U_{1,m_1}^p(\delta), p_1 - \epsilon_1 < \widehat{p}_{1,m_1} \leq p_1 + \epsilon_1').$$

Note that for any real numbers $a, b$ and any random variable $X, Y$ with $b, Y \geq 0$,

$$\mathbb{P}(ab \geq XY) \leq \mathbb{P}(\{a \geq X\} \text{ or } \{b \geq Y\}) \leq \mathbb{P}(a \geq X) + \mathbb{P}(b \geq Y).$$

By the above inequality, we can next bound the second term in the above bound,

$$\mathbb{P}\big(r_1 \geq U^\mu_{1,m_1}(\delta) \times U^p_{1,m_1}(\delta), p_1 - \epsilon_1 < \widehat{p}_{1,m_1} \leq p_1 + \epsilon'_1\big)$$

$$=\mathbb{P}\left\{\mu_1 p_1 \geq \left[\widehat{\mu}_{1,m_1} + 2\mathrm{e}D(\theta)C\left(\sqrt{\frac{\log(4/\delta)}{m_1\widehat{p}_{1,m_1}/2}} + E(\theta)\frac{\log^{(1/\theta)\vee 1}(4/\delta)}{m_1\widehat{p}_{1,m_1}/2}\right)\right]\right.$$
$$\left.\times\left[\widehat{p}_{1,m_1} + \sqrt{\frac{\log(2/\delta)}{2m_1}}\right], \, p_1 - \epsilon_1 < \widehat{p}_{1,m_1} \leq p_1 + \epsilon'_1\right\}$$

$$\leq\mathbb{P}\left\{\mu_1 p_1 \geq \left[\widehat{\mu}_{1,m_1} + 2\mathrm{e}D(\theta)C\left(\sqrt{\frac{\log(4/\delta)}{m_1(p_1+\epsilon'_1)/2}} + E(\theta)\frac{\log^{(1/\theta)\vee 1}(4/\delta)}{m_1(p_1+\epsilon'_1)/2}\right)\right]\right.$$
$$\left.\times\left[\widehat{p}_{1,m_1} + \sqrt{\frac{\log(2/\delta)}{2m_1}}\right], \, p_1 - \epsilon_1 < \widehat{p}_{1,m_1} \leq p_1 + \epsilon'_1\right\}$$

$$\leq\mathbb{P}\left\{\mu_1 p_1 \geq \left[\widehat{\mu}_{1,m_1} + 2\mathrm{e}D(\theta)C\left(\sqrt{\frac{\log(4/\delta)}{m_1(p_1+\epsilon'_1)/2}} + E(\theta)\frac{\log^{(1/\theta)\vee 1}(4/\delta)}{m_1(p_1+\epsilon'_1)/2}\right)\right]\right.$$
$$\left.\times\left[\widehat{p}_{1,m_1} + \sqrt{\frac{\log(2/\delta)}{2m_1}}\right]\right\}$$

$$\leq\mathbb{P}\left\{\mu_1 \geq \widehat{\mu}_{1,m_1} + 2\mathrm{e}D(\theta)C\left(\sqrt{\frac{\log(4/\delta)}{m_1(p_1+\epsilon'_1)/2}} + E(\theta)\frac{\log^{(1/\theta)\vee 1}(4/\delta)}{m_1(p_1+\epsilon'_1)/2}\right)\right\}$$
$$+ \mathbb{P}\left\{p_1 \geq \widehat{p}_{1,m_1} + \sqrt{\frac{\log(2/\delta)}{2m_1}}\right\}.$$

Since we have

$$\mathbb{P}\left\{p_1 \geq \widehat{p}_{1,m_1} + \sqrt{\frac{\log(2/\delta)}{2m_1}}\right\} \leq \frac{\delta}{2}$$

and

$$\mathbb{P}\left\{\mu_1 \geq \widehat{\mu}_{1,m_1} + 2\mathrm{e}D(\theta)C\left(\sqrt{\frac{\log(4/\delta)}{m_1(p_1+\epsilon'_1)/2}} + E(\theta)\frac{\log^{(1/\theta)\vee 1}(4/\delta)}{m_1(p_1+\epsilon'_1)/2}\right)\right\}$$

$$=\mathbb{P}\left\{\mu_1 \geq \widehat{\mu}_{1,m_1} + 2\mathrm{e}D(\theta)C\left(\sqrt{\frac{\log\left(\frac{4}{4(\delta/4)^{\frac{p_1}{p_1+\epsilon'_1}}}\right)}{m_1 p_1/2}} + E(\theta)\frac{\log^{(1/\theta)\vee 1}\left(\frac{4}{4(\delta/4)^{\left(\frac{p_1}{p_1+\epsilon'_1}\right)^{\theta\wedge 1}}}\right)}{m_1 p_1/2}\right)\right\}$$

$$\overset{\text{by } \frac{p_1}{p_1+\epsilon'_1} \leq \left(\frac{p_1}{p_1+\epsilon'_1}\right)^{\theta\wedge 1}}{\leq} \mathbb{P}\left\{\mu_1 \geq \widehat{\mu}_{1,m_1} + 2\mathrm{e}D(\theta)C\left(\sqrt{\frac{\log\left(\frac{4}{4(\delta/4)^{\left(\frac{p_1}{p_1+\epsilon'_1}\right)^{\theta\wedge 1}}}\right)}{m_1 p_1/2}} + E(\theta)\frac{\log^{(1/\theta)\vee 1}\left(\frac{4}{4(\delta/4)^{\left(\frac{p_1}{p_1+\epsilon'_1}\right)^{\theta\wedge 1}}}\right)}{m_1 p_1/2}\right)\right\}$$

$$\overset{\text{by Lemma 2.2}}{\leq} 4(\delta/4)^{\left(\frac{p_1}{p_1+\epsilon'_1}\right)^{\theta\wedge 1}},$$

whenever

$$m_1 \geq \frac{4}{p_1^2} \log \left( \frac{2}{4(\delta/4)^{\left(\frac{p_1}{p_1+\epsilon_1'}\right)^{\theta \wedge 1}}} \right) = \frac{4}{p_1^2} \log \left( \frac{1}{2} \left( \frac{4}{\delta} \right)^{\left(\frac{p_1}{p_1+\epsilon_1'}\right)^{\theta \wedge 1}} \right), \tag{G.1}$$

we can obtain

$$\mathbb{P}\big(r_1 \geq U_{1,m_1}^{\mu}(\delta) \times U_{1,m_1}^{p}(\delta), p_1 - \epsilon_1 < \widehat{p}_{1,m_1} \leq p_1 + \epsilon_1'\big) \leq 4(\delta/4)^{\left(\frac{p_1}{p_1+\epsilon_1'}\right)^{\theta \wedge 1}} + \delta/2.$$

which concludes that

$$\mathbb{P}\big((\mathcal{G}^0)^c \cap \mathcal{G}_1^p\big) \leq \exp\big(-m_1 d(p_1 + \epsilon_1', p_1)\big) + 4(\delta/4)^{\left(\frac{p_1}{p_1+\epsilon_1'}\right)^{\theta \wedge 1}} + \delta/2.$$

It remains to bound $\mathbb{P}\big((\mathcal{G}_k^r)^c \cap \mathcal{G}_k^p\big)$. We first decompose $\Omega = \{\widehat{p}_{k,m_k} \geq p_k + \epsilon_k'\} \cup \{\widehat{p}_{k,m_k} < p_k + \epsilon_k'\}$ and similarly obtain that

$$\mathbb{P}\big((\mathcal{G}_k^r)^c \cap \mathcal{G}_k^p\big)$$
$$\leq \mathbb{P}\big(\widehat{p}_{k,m_k} \geq p_k + \epsilon_k'\big) + \mathbb{P}\big(\widehat{p}_{k,m_k} \leq p_k + \epsilon_k', r_1 \leq U_{k,m_k}^{\mu}(\delta) \times U_{k,m_k}^{p}(\delta), \widehat{p}_{k,m_k} > p_k - \epsilon_k\big)$$
$$\leq \exp\big(-m_k d(p_k + \epsilon_k', p_k)\big) + \mathbb{P}\big(r_1 \leq U_{k,m_k}^{\mu}(\delta) \times U_{k,m_k}^{p}(\delta), p_k - \epsilon_k < \widehat{p}_{k,m_k} \leq p_k + \epsilon_k'\big).$$

The second term in above can be furthermore bounded by

$$\mathbb{P}\big(r_1 \leq U_{k,m_k}^{\mu}(\delta) \times U_{k,m_k}^{p}(\delta), p_k - \epsilon_k < \widehat{p}_{k,m_k} \leq p_k + \epsilon_k'\big)$$

$$= \mathbb{P}\left\{ \mu_1 p_1 \leq \left[ \widehat{\mu}_{k,m_k} + 2\mathrm{e}D(\theta)C\left( \sqrt{\frac{\log(4/\delta)}{m_k \widehat{p}_{k,m_k}/2}} + E(\theta)\frac{\log^{(1/\theta)\vee 1}(4/\delta)}{m_k \widehat{p}_{k,m_k}/2} \right) \right] \right.$$
$$\left. \times \left[ \widehat{p}_{k,m_k} + \sqrt{\frac{\log(2/\delta)}{2m_k}} \right], p_k - \epsilon_k < \widehat{p}_{k,m_k} \leq p_k + \epsilon_k' \right\}$$

$$= \mathbb{P}\left\{ \mu_k p_k + \Delta_k \leq \left[ \widehat{\mu}_{k,m_k} + 2\mathrm{e}D(\theta)C\left( \sqrt{\frac{\log(4/\delta)}{m_k \widehat{p}_{k,m_k}/2}} + E(\theta)\frac{\log^{(1/\theta)\vee 1}(4/\delta)}{m_k \widehat{p}_{k,m_k}/2} \right) \right] \right.$$
$$\left. \times \left[ \widehat{p}_{k,m_k} + \sqrt{\frac{\log(2/\delta)}{2m_k}} \right] - \Delta_k, p_k < \widehat{p}_{k,m_k} \leq p_k + \epsilon_k' \right\}$$

$$\leq \mathbb{P}\left\{ \mu_k p_k + \Delta_k \leq \left[ \widehat{\mu}_{k,m_k} + 2\mathrm{e}D(\theta)C\left( \sqrt{\frac{\log(4/\delta)}{m_k(p_k - \epsilon_k)/2}} + E(\theta)\frac{\log^{(1/\theta)\vee 1}(4/\delta)}{m_k(p_k - \epsilon_k)/2} \right) \right] \right.$$
$$\left. \times \left[ \widehat{p}_{k,m_k} + \sqrt{\frac{\log(2/\delta)}{2m_k}} \right], p_k - \epsilon_k < \widehat{p}_{k,m_k} \leq p_k + \epsilon_k' \right\}$$

$$\leq \mathbb{P}\left\{ \mu_k p_k + \Delta_k \leq \right.$$

$$\left. \left[ \widehat{\mu}_{k,m_k} + 2\mathrm{e}D(\theta)C\left( \sqrt{\frac{\log(4/\delta)}{m_k(p_k - \epsilon_k)/2}} + E(\theta)\frac{\log^{(1/\theta)\vee 1}(4/\delta)}{m_k(p_k - \epsilon_k)/2} \right) \right] \left[ \widehat{p}_{k,m_k} + \sqrt{\frac{\log(2/\delta)}{2m_k}} \right] \right\}$$

$$= \mathbb{P}\left\{ \left( \mu_k + \frac{\Delta_k}{2p_k} \right)\left( p_k + \frac{p_k \Delta_k}{2r_k + \Delta_k} \right) \leq \right.$$

$$\left. \left[ \widehat{\mu}_{k,m_k} + 2\mathrm{e}D(\theta)C\left( \sqrt{\frac{\log(4/\delta)}{m_k(p_k - \epsilon_k)/2}} + E(\theta)\frac{\log^{(1/\theta)\vee 1}(4/\delta)}{m_k(p_k - \epsilon_k)/2} \right) \right] \left[ \widehat{p}_{k,m_k} + \sqrt{\frac{\log(2/\delta)}{2m_k}} \right] \right\}$$

$$\leq P_{k,\mu} + P_{k,p},$$

where the last step is by the fact that for any real numbers $a, b$ and any random variable $X, Y$ with $b, Y \geq 0$,

$$\mathbb{P}(ab \leq XY) \leq \mathbb{P}(\{a \leq X\} \text{ or } \{b \leq Y\}) \leq \mathbb{P}(a \leq X) + \mathbb{P}(b \leq Y).$$

Now, the two parts in the upper bound of $\mathbb{P}\left(r_1 \leq U_{k,m_k}^\mu(\delta) \times U_{k,m_k}^p(\delta), \, p_k - \epsilon_k < \widehat{p}_{k,m_k} \leq p_k + \epsilon_k'\right)$ can be furthermore bounded. Indeed, $P_{k,p}$ can be bounded as

$$
\begin{aligned}
P_{k,p} &= \mathbb{P}\left\{ p_k + \frac{p_k \Delta_k}{2r_k + \Delta_k} \leq \widehat{p}_{k,m_k} + \sqrt{\frac{\log(2/\delta)}{2m_k}} \right\} \\
&= \mathbb{P}\left\{ \frac{p_k \Delta_k}{2r_k + \Delta_k} - \sqrt{\frac{\log(2/\delta)}{2m_k}} \leq \widehat{p}_{k,m_k} - p_k \right\} \\
&\leq \exp\left\{ -2m_k \left( \frac{p_k \Delta_k}{2r_k + \Delta_k} - \sqrt{\frac{\log(2/\delta)}{2m_k}} \right)^2 \right\} \\
&= \frac{\delta}{2} \exp\left\{ -\frac{2m_k p_k \Delta_k}{2r_k + \Delta_k} \left( \frac{p_k \Delta_k}{2r_k + \Delta_k} - \sqrt{\frac{2\log(2/\delta)}{m_k}} \right) \right\} \\
&\leq \delta/2
\end{aligned}
$$

as long as

$$m_k \geq \frac{2(2r_k + \Delta_k)^2}{p_k^2 \Delta_k^2} \log\left(\frac{2}{\delta}\right). \tag{G.2}$$

Similarly,

$$
\begin{aligned}
&P_{k,\mu} \\
&= \mathbb{P}\left\{ \mu_k + \frac{\Delta_k}{2p_k} \leq \widehat{\mu}_{k,m_k} + 2eD(\theta)C\left( \sqrt{\frac{\log(4/\delta)}{m_k(p_k - \epsilon_k)/2}} + E(\theta)\frac{\log^{(1/\theta)\vee 1}(4/\delta)}{m_k(p_k - \epsilon_k)/2} \right) \right\} \\
&= \mathbb{P}\left\{ \frac{\Delta_k}{2p_k} - 2eD(\theta)C\left( \sqrt{\frac{\log(4/\delta)}{m_k(p_k - \epsilon_k)/2}} + E(\theta)\frac{\log^{(1/\theta)\vee 1}(4/\delta)}{m_k(p_k - \epsilon_k)/2} \right) \leq \widehat{\mu}_{k,m_k} - \mu_k \right\} \\
&\overset{\text{by Lemma 2.2}}{\leq} \mathbb{P}\left\{ \frac{\Delta_k}{2p_k} - 2eD(\theta)C\left( \sqrt{\frac{\log(4/\delta)}{m_k(p_k - \epsilon_k)/2}} + E(\theta)\frac{\log^{(1/\theta)\vee 1}(4/\delta)}{m_k(p_k - \epsilon_k)/2} \right) \leq \widehat{\mu}_{k,m_k} - \mu_k, \right. \\
&\qquad\qquad \left. \left\| \mu_k - \widehat{\mu}_{k,m_k} \right\|_{\Psi_\theta, \frac{p_k E(\theta)}{2\sqrt{m_k}}} \leq \frac{4eD(\theta)C}{p_k \sqrt{m_k}} \right\} + \delta/2 \\
&\overset{\text{by (F.1)}}{\leq} \frac{2}{1 + \Psi_{\theta, \frac{3^{1/2-1/\theta}p_k E(\theta)}{2\sqrt{m_k}}}(\sqrt{3}s)} + \delta/2
\end{aligned}
$$

where

$$s = \left\| \mu_k - \widehat{\mu}_{k,m_k} \right\|_{\Psi_\theta, \frac{p_k E(\theta)}{2\sqrt{m_k}}}^{-1} \left[ \frac{\Delta_k}{2p_k} - 2eD(\theta)C\left( \sqrt{\frac{\log(4/\delta)}{m_k(p_k - \epsilon_k)/2}} + E(\theta)\frac{\log^{(1/\theta)\vee 1}(4/\delta)}{m_k(p_k - \epsilon_k)/2} \right) \right].$$

By the increasing property of $\Psi_{\theta,L}(\cdot)$, we obtain that

$$\Psi_{\theta,\frac{3^{1/2-1/\theta}p_k E(\theta)}{2\sqrt{m_k}}}\left(\sqrt{3}s\right)$$

$$\geq \Psi_{\theta,\frac{3^{1/2-1/\theta}p_k E(\theta)}{2\sqrt{m_k}}}\left(\frac{p_k\sqrt{3m_k}}{4eD(\theta)C}\left[\frac{\Delta_k}{2p_k}-2eD(\theta)C\left(\sqrt{\frac{\log(4/\delta)}{m_k(p_k-\epsilon_k)/2}}+E(\theta)\frac{\log^{(1/\theta)\vee 1}(4/\delta)}{m_k(p_k-\epsilon_k)/2}\right)\right]\right)$$

$$\overset{\text{by (G.3)}}{\geq}\Psi_{\theta,\frac{3^{1/2-1/\theta}p_k E(\theta)}{2\sqrt{m_k}}}\left(\frac{\sqrt{3p_k m_k}}{4eD(\theta)C}\left[2eD(\theta)C\left(\sqrt{\frac{\log(4/\delta)}{m_k(p_k-\epsilon_k)/2}}+E(\theta)\frac{\log^{(1/\theta)\vee 1}(4/\delta)}{m_k(p_k-\epsilon_k)/2}\right)\right]\right)$$

$$=\Psi_{\theta,\frac{3^{1/2-1/\theta}p_k E(\theta)}{2\sqrt{m_k}}}\left(\sqrt{\frac{3p_k\log(4/\delta)}{2(p_k-\epsilon_k)}}+\frac{\sqrt{3p_k}E(\theta)\log^{(1/\theta)\vee 1}(4/\delta)}{\sqrt{m_k}(p_k-\epsilon_k)}\right)$$

$$=\Psi_{\theta,\frac{3^{1/2-1/\theta}p_k E(\theta)}{2\sqrt{m_k}}}\left(\sqrt{\log\left(\frac{4}{\delta}\right)^{\frac{3p_k}{2(p_k-\epsilon_k)}}}+\frac{3^{1/2-1/\theta}p_k E(\theta)}{2\sqrt{m_k}}\log^{(1/\theta)\vee 1}\left(\frac{4}{\delta}\right)^{\frac{2^{\theta\wedge 1}}{3^{(1/\theta)\wedge 1}p_k^{(\theta\wedge 1)/2}(p_k-\epsilon_k)^{\theta\wedge 1}}}\right)$$

$$\overset{\text{by }\frac{3p_k}{2(p_k-\epsilon_k)}\leq\frac{2^{\theta\wedge 1}}{3^{(1/\theta)\wedge 1}p_k^{(\theta\wedge 1)/2}(p_k-\epsilon_k)^{\theta\wedge 1}}}{\geq}\Psi_{\theta,\frac{3^{1/2-1/\theta}p_k E(\theta)}{2\sqrt{m_k}}}\left(\sqrt{\log\left(\frac{4}{\delta}\right)^{\frac{3p_k}{2(p_k-\epsilon_k)}}}+\frac{3^{1/2-1/\theta}p_k E(\theta)}{2\sqrt{m_k}}\log^{(1/\theta)\vee 1}\left(\frac{4}{\delta}\right)^{\frac{3p_k}{2(p_k-\epsilon_k)}}\right)$$

$$=\left(\frac{4}{\delta}\right)^{\frac{3p_k}{2(p_k-\epsilon_k)}}-1$$

whenever

$$2\left(1+\frac{1}{\sqrt{p_k}}\right)eD(\theta)C\left(\sqrt{\frac{\log(4/\delta)}{m_k(p_k-\epsilon_k)/2}}+E(\theta)\frac{\log^{(1/\theta)\vee 1}(4/\delta)}{m_k(p_k-\epsilon_k)/2}\right)\leq\frac{\Delta_k}{2p_k}, \tag{G.3}$$

and a sufficient condition for this is

$$2\left(1+\frac{1}{\sqrt{p_k}}\right)eD(\theta)C\sqrt{\frac{\log(4/\delta)}{m_k(p_k-\epsilon_k)/2}}\leq\frac{\Delta_k}{4p_k}$$

and

$$2\left(1+\frac{1}{\sqrt{p_k}}\right)eD(\theta)CE(\theta)\frac{\log^{(1/\theta)\vee 1}(4/\delta)}{m_k(p_k-\epsilon_k)/2}\leq\frac{\Delta_k}{4p_k}.$$

Thus, we can take

$$m_k\geq\frac{128e^2 D^2(\theta)C^2 p_k(1+\sqrt{p_k})^2}{(p_k-\epsilon_k)\Delta_k^2}\log\left(\frac{4}{\delta}\right)+\frac{16eD(\theta)CE(\theta)\sqrt{p_k}(1+\sqrt{p_k})}{(p_k-\epsilon_k)\Delta_k}\log^{(1/\theta)\vee 1}\left(\frac{4}{\delta}\right). \tag{G.4}$$

Therefore, we can furthermore upper-bound $P_{k,\mu}$ as

$$P_{k,\mu}\leq\frac{2}{1+\Psi_{\theta,\frac{3^{1/2-1/\theta}p_k E(\theta)}{2\sqrt{m_k}}}\left(\sqrt{3}s\right)}+\delta/2$$

$$\leq\frac{2}{1+\left(\frac{4}{\delta}\right)^{\frac{3p_k}{2(p_k-\epsilon_k)}}-1}+\delta/2$$

$$=2\left(\frac{\delta}{4}\right)^{\frac{3p_k}{2(p_k-\epsilon_k)}}+\frac{\delta}{2}.$$

Thus, we obtain that

$$\mathbb{P}\left((\mathcal{G}_k^r)^c\cap\mathcal{G}_k^p\right)\leq\exp\left(-m_k d(p_k+\epsilon_k',p_k)\right)+2\left(\frac{\delta}{4}\right)^{\frac{3p_k}{2(p_k-\epsilon_k)}}+\delta.$$

under condition (G.2) and (G.4). To summarize these results, we have

$$\mathbb{P}(\mathcal{G}_k^c) \le \mathbb{P}\big((\mathcal{G}^0)^c \cap \mathcal{G}_1^p\big) + \mathbb{P}\big((\mathcal{G}_1^p)^c\big) + \mathbb{P}\big((\mathcal{G}_k^r)^c \cap \mathcal{G}_k^p\big) + \mathbb{P}\big((\mathcal{G}_k^p)^c\big)$$

$$\le \exp\big(-m_1 d(p_1 + \epsilon_1', p_1)\big) + 4(\delta/4)^{\left(\frac{p_1}{p_1+\epsilon_1'}\right)^{\theta \wedge 1}} + \delta/2 + \exp\big(-m_1 d(p_1, p_1 - \epsilon_1)\big)$$

$$+ \exp\big(-m_k d(p_k + \epsilon_k', p_k)\big) + 2\,(\delta/4)^{\frac{3p_k}{2(p_k - \epsilon_k)}} + \delta + \exp\big(-m_k d(p_k, p_k - \epsilon_k)\big),$$

whenever $m_1$ satisfies (G.1) and $m_k$ satisfies (G.2) and (G.4).

**Step 2:** Now, we deal with $\mathbb{E}[c_k(T) \cap \mathbb{1}(\mathcal{G}_k)]$. If $c_k(T) > \max\{m_1, m_k\}$, then arm $k$ was pulled more than $m_k$ times over the first $T$ rounds, and so there must exist a round $t \in [m_1, \dots, T]$ such that $A_t = k$. However, on the good event $\mathcal{G}_k$, we have

$$U_k^\mu(t, \delta) U_k^p(t, \delta) = U_{k,m_k}^\mu(\delta) U_{k,m_k}^p(\delta)$$

$$\overset{\text{on the event } \mathcal{G}_k^r}{<} r_1$$

$$\overset{\text{on the event } \mathcal{G}^0}{<} \min_{t \in [m_1, \dots, T]} U_1^\mu(t, \delta) U_1^p(t, \delta) \le U_1^\mu(t, \delta) U_1^p(t, \delta).$$

This means the agent will choose arm 1 instead of arm $k$ at time point $t$, which leads to a contradiction. Thus, we must have

$$c_k(T) \le \max\{m_1, m_k\}.$$

**Step 3:** Combining the inequality in **Step 1** and **Step 2**, we obtain

$$\mathbb{E}c_k(T) \le \mathbb{E}[c_k(T) \cap \mathbb{1}(\mathcal{E}_k)] + \mathbb{E}[c_k(T) \cap \mathbb{1}\{(\mathcal{E}_k)^c\}]$$

$$\le \mathbb{E}[c_k(T) \cap \mathbb{1}(\mathcal{E}_k)] + T\mathbb{P}(\mathcal{E}_k^c)$$

$$\le \max\{m_1, m_k\}$$

$$+ T\Big[\exp\big(-m_1 d(p_1 + \epsilon_1', p_1)\big) + \exp\big(-m_1 d(p_1, p_1 - \epsilon_1)\big) + 4(\delta/4)^{\left(\frac{p_1}{p_1+\epsilon_1'}\right)^{\theta \wedge 1}} + 2\,(\delta/4)^{\frac{3p_k}{2(p_k - \epsilon_k)}}$$

$$+ 3\delta/2 + \exp\big(-m_k d(p_k, p_k - \epsilon_k)\big) + \exp\big(-m_k d(p_k + \epsilon_k', p_k)\big)\Big]$$

$$\le \max\{m_1, m_k\} + T\Big[\exp\{-2m_1 \epsilon_1'^2\} + \exp\{-2m_1 \epsilon_1^2\}$$

$$+ \exp\{-2m_k \epsilon_k'^2\} + \exp\{-2m_k \epsilon_k^2\} + 4(\delta/4)^{\left(\frac{p_1}{p_1+\epsilon_1'}\right)^{\theta \wedge 1}} + 2\,(\delta/4)^{\frac{3p_k}{2(p_k - \epsilon_k)}} + 2\delta\Big]$$

whenever $m_1$ satisfies (G.1) and $m_k$ satisfies (G.2) and (G.4). Now, taking

$$m_1 = \frac{4}{p_1^2} \log\left(\frac{1}{2}\left(\frac{4}{\delta}\right)^{\left(\frac{p_1}{p_1+\epsilon_1'}\right)^{\theta \wedge 1}}\right)$$

and

$$m_k$$
$$= \left(\frac{2(2r_k + \Delta_k)^2}{p_k^2} + \frac{128 \mathrm{e}^2 D^2(\theta) C^2 p_k (1 + \sqrt{p_k})^2}{(p_k - \epsilon_k)}\right) \frac{\log(4/\delta)}{\Delta_k^2}$$

$$+ \frac{16 \mathrm{e} D(\theta) C E(\theta) \sqrt{p_k}(1 + \sqrt{p_k})}{(p_k - \epsilon_k)} \frac{\log^{(1/\theta) \vee 1}(4/\delta)}{\Delta_k}$$

$$\ge \frac{2(2r_k + \Delta_k)^2}{p_k^2 \Delta_k^2} \log\left(\frac{2}{\delta}\right)$$

$$+ \frac{128 \mathrm{e}^2 D^2(\theta) C^2 p_k (1 + \sqrt{p_k})^2}{(p_k - \epsilon_k)\Delta_k^2} \log\left(\frac{4}{\delta}\right) + \frac{16 \mathrm{e} D(\theta) C E(\theta) \sqrt{p_k}(1 + \sqrt{p_k})}{(p_k - \epsilon_k)\Delta_k} \log^{(1/\theta) \vee 1}\left(\frac{4}{\delta}\right).$$

with $\epsilon_k = \epsilon'_k = p_k/2$ for $k \in [K]$ satisfies (G.1), (G.2), and (G.4) by $r_k \in (0,1]$. Under these choice, we obtain

$$\exp\{-2m_1\epsilon'^2_1\} = \exp\{-2m_1\epsilon^2_1\}$$
$$= \exp\left\{-2\log\left(\frac{1}{2}\left(\frac{4}{\delta}\right)^{(1/2)^{\theta \wedge 1}}\right)\right\}$$
$$= \left(\frac{1}{2}\right)^{-2}\left[\left(\frac{4}{\delta}\right)^{(1/2)^{\theta \wedge 1}}\right]^{-2} = 4\left(\delta^2/16\right)^{(1/2)^{\theta \wedge 1}},$$

and

$$\exp\{-2m_k\epsilon'^2_k\} = \exp\{-2m_k\epsilon^2_k\}$$
$$= \exp\left\{-\frac{p_k^2}{2}\left[\left(\frac{2(2r_k+\Delta_k)^2}{p_k^2} + \frac{128e^2 D^2(\theta)C^2 p_k(1+\sqrt{p_k})^2}{p_k/2}\right)\frac{\log(4/\delta)}{\Delta_k^2}\right.\right.$$
$$\left.\left. + \frac{16eD(\theta)CE(\theta)\sqrt{p_k}(1+\sqrt{p_k})}{p_k/2}\frac{\log^{(1/\theta)\vee 1}(4/\delta)}{\Delta_k}\right]\right\}$$
$$= \exp\left\{-\left[(2r_k+\Delta_k)^2 + 128p_k^2e^2 D^2(\theta)C^2(1+\sqrt{p_k})\right]\frac{\log(4/\delta)}{\Delta_k^2}\right.$$
$$\left. - 16eD(\theta)CE(\theta)p_k^{3/2}(1+\sqrt{p_k})\frac{\log^{(1/\theta)\vee 1}(4/\delta)}{\Delta_k}\right\}$$
$$\leq \exp\left\{-(2r_k+\Delta_k)^2\frac{\log(4/\delta)}{\Delta_k^2}\right\}$$
$$\leq \exp\left\{-\Delta_k^2\frac{\log(4/\delta)}{\Delta_k^2}\right\} = \delta/4.$$

Aggregating these results, $\mathbb{E}c_k(T)$ can be furthermore upper bounded by

$$\mathbb{E}c_k(T) \leq \max\{m_1, m_k\} + T\left[8\left(\delta^2/16\right)^{(1/2)^{\theta \wedge 1}} + 2\delta/4 + 4(\delta/4)^{(1/2)^{\theta \wedge 1}} + 2(\delta/4)^3 + 2\delta\right]$$
$$\leq m_1 + m_k + T\left[8\left(\delta^2/16\right)^{1/2} + \delta/2 + 4(\delta/4)^{1/2} + 2(\delta/4)^3 + 2\delta\right]$$
$$= m_1 + m_k + T\left(2\sqrt{\delta} + 3\delta + \delta^3/32\right)$$
$$= \frac{4}{p_1^2}\log\left((4/\delta)^{(1/2)^{\theta \wedge 1}}/2\right) + \left(\frac{2(2r_k+\Delta_k)^2}{p_k^2} + \frac{128e^2 D^2(\theta)C^2 p_k(1+\sqrt{p_k})^2}{p_k/2}\right)\frac{\log(4/\delta)}{\Delta_k^2}$$
$$+ \frac{16eD(\theta)CE(\theta)\sqrt{p_k}(1+\sqrt{p_k})}{p_k/2}\frac{\log^{(1/\theta)\vee 1}(4/\delta)}{\Delta_k} + T\left(3\sqrt{\delta} + 4\delta + \delta^3/32\right)$$
$$\leq \frac{4}{p_1^2}\left(\frac{1}{2}\right)^{\theta \wedge 1}\log(4/\delta) + \left[(2r_k+\Delta_k)^2 + 128eD^2(\theta)C^2 p_k^2(1+\sqrt{p_k})^2\right]\frac{2\log(4/\delta)}{p_k^2\Delta_k^2}$$
$$+ 32eD(\theta)CE(\theta)\sqrt{p_k}(1+\sqrt{p_k})\frac{\log^{(1/\theta)\vee 1}(4/\delta)}{p_k\Delta_k} + T\left(3\sqrt{\delta} + 4\delta + \delta^3/32\right)$$
$$\overset{\text{by } r_k \leq 1 \text{ and } p_k \leq 1}{\leq} \frac{4}{p_1^2}\log(4/\delta) + 2\left(9 + 512eD^2(\theta)C^2\right)\frac{\log(4/\delta)}{p_k^2\Delta_k^2}$$
$$+ 64eD(\theta)CE(\theta)\frac{\log^{(1/\theta)\vee 1}(4/\delta)}{p_k\Delta_k} + T\left(3\sqrt{\delta} + 4\delta + \delta^3/32\right).$$

Now, choose $\delta = 4/T^2$,

$$\mathbb{E}c_k(T) \leq \frac{8\log T}{p_1^2} + 4\Big(9 + 512\mathrm{e}D^2(\theta)C^2\Big)\frac{\log T}{p_k^2\Delta_k^2}$$

$$+ 2^{6+(1/\theta)\vee 1}\mathrm{e}D(\theta)CE(\theta)\frac{\log^{(1/\theta)\vee 1} T}{p_k\Delta_k} + T\left(\frac{6}{T} + \frac{16}{T^2} + \frac{2}{T^6}\right).$$

Finally, we obtain the cumulative regret is bounded by

$$\mathcal{R}(T)$$

$$= \sum_{k=2}^{K}\Delta_k\mathbb{E}c_k(T)$$

$$\leq \frac{8}{p_1^2}\log T\sum_{k=2}^{K}\Delta_k + 4\Big(9 + 512\mathrm{e}D^2(\theta)C^2\Big)\sum_{k=2}^{K}\frac{\log T}{p_k^2\Delta_k}$$

$$+ 2^{6+(1/\theta)\vee 1}\mathrm{e}D(\theta)CE(\theta)\sum_{k=2}^{K}\frac{\log^{(1/\theta)\vee 1} T}{p_k} + \left(6 + \frac{16}{T} + \frac{2}{T^2}\right)\sum_{k=2}^{K}\Delta_k.$$

which gives the regret in the theorem. $\qquad\square$

## G.2. Proof of Theorem B.2

Before proving the heavy tailed bandit results. We first state some basic properties of $g(p, \epsilon)$ in the concentration of Lemma B.1. Define $h(p, \epsilon) := pg(p, \epsilon) = (1+\epsilon)2^{\frac{\epsilon}{1+\epsilon}}p^{\frac{1}{1+\epsilon}} + 2\sqrt{p} + \frac{4}{3}$, then $g$ is monotonically decreasing and $h$ is monotonically increasing with respect to $p$. Specially, they satisfy

$$h(p, \epsilon) \leq (1+\epsilon)\times 2\times 1 + 2\times 1 + \frac{4}{3} = \frac{2}{3}(8 + 2\epsilon)$$

for any $p \in (0,1)$, and

$$g(p_1, \epsilon) = \frac{(1+\epsilon)2^{\frac{\epsilon}{1+\epsilon}}}{p_1^{\frac{\epsilon}{1+\epsilon}}} + \frac{4}{3p_1} + \frac{2}{\sqrt{p_1}} \geq (1+\epsilon)2^{\frac{\epsilon}{1+\epsilon}}p_2^{\frac{1}{1+\epsilon}} + \frac{4}{3} + 2\sqrt{p_2} = h(p_2, \epsilon)$$

for any $p_1, p_2 \in (0,1)$.

*Proof.* The proof is similar to the proof of Theorem 4.1. The essential change is we use the concentration of trimming observable sample mean in Lemma B.1 instead of sub-Weibull concentrations. We will borrow some techniques in (Bubeck et al., 2013). For fixed $\epsilon$, we will write $g(p) = g(p, \epsilon)$ and $h(p) = h(p, \epsilon)$.

**Step 1:**

Similar as the technique in Bubeck et al. (2013), suppose $A_t = k$, we define the following bad events

$$\mathcal{B}^0(t) := \{r_1 \geq U_{1,c_1(t),t}^{\mu} \times U_{1,c_1(t)}^p\}, \qquad \mathcal{B}_k^{\mu}(t) := \left\{\widehat{\mu}_{k,c_k(t),t} \geq \mu_k + g(\widehat{p}_{k,t})M^{\frac{1}{1+\epsilon}}\left(\frac{\log t^2}{c_k(t)}\right)^{\frac{\epsilon}{1+\epsilon}}\right\}$$

and

$$\mathcal{B}_k^{\mathrm{count}}(t) := \left\{c_k(t) \leq \left[2(p_k + \varepsilon_k)g(p_k + \varepsilon_k)\right]^{\frac{1+\epsilon}{\epsilon}}\frac{M^{\frac{1}{\epsilon}}}{(\Delta_k - \varepsilon_k\mu_k)^{\frac{1+\epsilon}{\epsilon}}}\log T\right\}$$

$$= \left\{c_k(t) \leq \left[2h(p_k + \varepsilon_k)\right]^{\frac{1+\epsilon}{\epsilon}}\frac{M^{\frac{1}{\epsilon}}}{(\Delta_k - \varepsilon_k\mu_k)^{\frac{1+\epsilon}{\epsilon}}}\log T\right\}$$

with $\varepsilon_k \in (0, \Delta_k/\mu_k)$ determined later. Similarly, define

$$\mathcal{B}_k^p(t) = \{\widehat{p}_{k,t} > p_k + \varepsilon_k\}$$

On the event $\mathcal{B}^{0c} \cap \mathcal{B}_k^{\mu c} \cap \mathcal{B}_k^{\text{count}, c} \cap \mathcal{B}_k^{pc}$, we have

$$U_{1,c_1(t),t}^{\mu} \times U_{1,c_1(t),t}^{p}$$
$$> r_1$$
$$= r_k + \Delta_k$$
$$= \mu_k p_k + \Delta_k$$
$$\geq \mu_k p_k + 2h(p_k + \varepsilon_k) \left( \frac{\log t^2}{c_k(t)} \right)^{\epsilon/(1+\epsilon)} + \mu_k \varepsilon_k$$
$$\overset{h \text{ is increasing}}{\geq} \mu_k p_k + 2h(\widehat{p}_{k,t}) M^{1/(1+\epsilon)} \left( \frac{\log t^2}{c_k(t)} \right)^{\epsilon/(1+\epsilon)} + \mu_k \varepsilon_k$$
$$\geq \mu_k (\widehat{p}_{k,t} - \varepsilon_k) + 2h(\widehat{p}_{k,t}) M^{1/(1+\epsilon)} \left( \frac{\log t^2}{c_k(t)} \right)^{\epsilon/(1+\epsilon)} + \mu_k \varepsilon_k$$
$$= \mu_k \widehat{p}_{k,t} + 2h(\widehat{p}_{k,t}) M^{1/(1+\epsilon)} \left( \frac{\log t^2}{c_k(t)} \right)^{\epsilon/(1+\epsilon)}$$
$$\geq \left[ \widehat{\mu}_{k,c_k(t),t} - g(\widehat{p}_{k,t}) M^{\frac{1}{1+\epsilon}} \left( \frac{\log t^2}{c_k(t)} \right)^{\frac{\epsilon}{1+\epsilon}} \right] \widehat{p}_{k,t} + 2h(\widehat{p}_{k,t}) M^{1/(1+\epsilon)} \left( \frac{\log t^2}{c_k(t)} \right)^{\epsilon/(1+\epsilon)}$$
$$= \left[ \widehat{\mu}_{k,c_k(t),t} + \widehat{p}_{k,t} M^{1/(1+\epsilon)} \left( \frac{\log t^2}{c_k(t)} \right)^{\epsilon/(1+\epsilon)} \right] \widehat{p}_{k,t}$$
$$= U_{k,c_k(t),t}^{\mu} \times U_{k,c_k(t),t}^{p}.$$

This implies $A_t = 1$, which leads to a contradiction.

**Step 2:**

Consider the probability:

$$\mathbb{P}(\mathcal{B}^0 \cup \mathcal{B}_k^{\mu} \cup \mathcal{B}_k^{p}) = \mathbb{P}((\mathcal{B}^0 \cup \mathcal{B}_k^{\mu}) \cap (\mathcal{B}_k^{p})^c) + \mathbb{P}(\mathcal{B}_k^{p})$$
$$\leq \mathbb{P}(\mathcal{B}^0 \cap (\mathcal{B}_k^{p})^c) + \mathbb{P}(\mathcal{B}_k^{\mu} \cap (\mathcal{B}_k^{p})^c) + \mathbb{P}(\mathcal{B}_k^{p})$$

with $\mathbb{P}(\mathcal{B}_k^{p})$ and $\mathbb{P}(\mathcal{B}_k^{\mu} \cap (\mathcal{B}_k^{p})^c)$ can be bounded easily. Indeed, if we consider them, then

$$\mathbb{P}(\mathcal{B}_k^{p}(t)) = \mathbb{P}(\widehat{p}_{k,t} > p_k + \varepsilon_k) \leq \exp\left(-t\varepsilon_k^2/2\right).$$

Next,

$$\mathbb{P}(\mathcal{B}_k^{\mu}(t) \cap (\mathcal{B}_k^{p}(t))^c)$$
$$= \mathbb{P}\left( \left\{ \widehat{\mu}_{k,c_k(t),t} \geq \mu_k + g(\widehat{p}_{k,t}) M^{\frac{1}{1+\epsilon}} \left( \frac{\log t^2}{c_k(t)} \right)^{\frac{\epsilon}{1+\epsilon}} \right\} \cap \{\widehat{p}_{k,t} \leq p_k + \varepsilon_k\} \right)$$
$$\leq \mathbb{P}(p_k - \varepsilon_k > \widehat{p}_{k,t})$$
$$\quad + \mathbb{P}\left( \left\{ \widehat{\mu}_{k,c_k(t),t} \geq \mu_k + g(\widehat{p}_{k,t}) M^{\frac{1}{1+\epsilon}} \left( \frac{\log t^2}{c_k(t)} \right)^{\frac{\epsilon}{1+\epsilon}} \right\} \cap \{p_k - \varepsilon_k \leq \widehat{p}_{k,t} \leq p_k + \varepsilon_k\} \right)$$
$$\overset{\text{by } g \text{ is decreasing}}{\leq} \mathbb{P}(p_k - \varepsilon_k > \widehat{p}_{k,t}) + \mathbb{P}\left( \widehat{\mu}_{k,c_k(t),t} \geq \mu_k + g(p_k + \varepsilon_k) M^{\frac{1}{1+\epsilon}} \left( \frac{\log t^2}{c_k(t)} \right)^{\frac{\epsilon}{1+\epsilon}} \right),$$

with

$$\mathbb{P}(p_k - \varepsilon_k > \widehat{p}_{k,t}) \leq \exp\left(-t\varepsilon_k^2/2\right).$$

and

$$\mathbb{P}\left(\widehat{\mu}_{k,c_k(t),t} \geq \mu_k + g(p_k + \varepsilon_k) M^{\frac{1}{1+\epsilon}}\left(\frac{\log t^2}{c_k(t)}\right)^{\frac{\epsilon}{1+\epsilon}}\right)$$

$$\leq 2\exp\left(-\frac{n\left(g(p_k + \varepsilon_k) M^{\frac{1}{1+\epsilon}}\left(\frac{\log t^2}{c_k(t)}\right)^{\frac{\epsilon}{1+\epsilon}}\right)^{\frac{1+\epsilon}{\epsilon}}}{M^{\frac{1}{\epsilon}} g^{\frac{1+\epsilon}{\epsilon}}(p_k)}\right)$$

$$= 2\exp\left(-\frac{g^{\frac{1+\epsilon}{\epsilon}}(p_k + \varepsilon_k)}{g^{\frac{1+\epsilon}{\epsilon}}(p_k)}\frac{\log t^2}{c_k(t)}\right)$$

$$\overset{\text{we need } p_k+\varepsilon_k \leq 1}{\leq} 2\exp\left(-\frac{h^{\frac{1+\epsilon}{\epsilon}}(p_k)}{g^{\frac{1+\epsilon}{\epsilon}}(p_k)}\frac{\log t^2}{c_k(t)}\right)$$

$$= 2\exp\left(-p_k^{\frac{1+\epsilon}{\epsilon}}\frac{\log t^2}{c_k(t)}\right)$$

$$\overset{\text{Proof in Proposition 1 of (Bubeck et al., 2013)}}{\leq} 2\sum_{c_k(t)=1}^{t}\frac{1}{c_k(t)^4} \leq \frac{2}{t^3}.$$

Thus, we obtain

$$\mathbb{P}\left(\mathcal{B}_k^{\mu}(t) \cap (\mathcal{B}_k^p(t))^c\right) \leq \exp\left(-t\varepsilon_k^2/2\right) + 2/t^3.$$

Similarly, we can show that

$$\mathbb{P}(\mathcal{B}^0 \cap (\mathcal{B}_k^p)^c) = \mathbb{P}\{r_1 \geq U_{1,c_1(t-1),t}^{\mu} \times U_{1,c_1(t-1)}^p, \widehat{p}_{k,t} < p_k + \varepsilon_k\}$$

$$\leq \exp\left(-t\varepsilon_1^2/2\right) + 2/t^3.$$

for any $\varepsilon_1 \in (0, p_1)$.

**Step 3:** Denote

$$V_k := \left[2p_k g(p_k)\right]^{\frac{1+\epsilon}{\epsilon}}\frac{M^{\frac{1}{\epsilon}}}{\Delta_k^{\frac{1+\epsilon}{\epsilon}}}\log T$$

Take $\varepsilon_1 = p_1/2$ and $\varepsilon_k = \frac{\Delta_k}{2\mu_k}$ satisfy $\varepsilon_1 \in (0, p_1)$ and $\varepsilon_k \in (0, \Delta_k/\mu_k)$ for $k = 2, \ldots, K$. From Step 2, we know that

$$\mathbb{P}(\mathcal{B}^0(t) \cup \mathcal{B}_k^{\mu}(t) \cup \mathcal{B}_k^p)$$
$$\leq \exp\left(-t\varepsilon_k^2/2\right) + \left[\exp\left(-t\varepsilon_k^2/2\right) + 2/t^2\right] + \left[\exp\left(-t\varepsilon_1^2/2\right) + 2/t^2\right]$$
$$= 2\exp\left(-\frac{t\Delta_k^2}{8\mu_k^2}\right) + \exp\left(-\frac{tp_1^2}{8}\right) + \frac{4}{t^3},$$

thus

$$
\begin{aligned}
\mathbb{E}c_k(T) &= \mathbb{E}\sum_{t=1}^{T} \mathbb{1}(A_t = k) \\
&\leq \sum_{t=1}^{T} \mathbb{E}\big(\mathbb{1}\{A_t = k\} \cap \mathcal{B}_k^{\text{count}}(t)\big) + \sum_{t=1}^{T} \mathbb{E}\big(\mathbb{1}\{A_t = k\} \cap \mathcal{B}_k^{\text{count},\,c}(t)\big) \\
&\leq V_k + \sum_{t=1}^{T} \mathbb{E}\big(\mathbb{1}\{A_t = k\} \cap \mathcal{B}_k^{\text{count},\,c}(t)\big) \\
&= V_k + \sum_{t > V_k}^{T} \mathbb{E}\big(\mathbb{1}\{A_t = k\} \cap \mathcal{B}_k^{\text{count},\,c}(t)\big) \\
&\leq V_k + \sum_{t > V_k}^{T} \mathbb{P}\big(\mathcal{B}_k^{\text{count},\,c}(t)\big) \\
&\leq V_k + \sum_{t > V_k}^{T} \mathbb{P}(\mathcal{B}^0(t) \cup \mathcal{B}_k^{\mu}(t) \cup \mathcal{B}_k^{p}) \\
&\leq V_k + \sum_{t > V_k}^{T} \left[2\exp\left(-\frac{t\Delta_k^2}{8\mu_k^2}\right) + \exp\left(-\frac{tp_1^2}{8}\right) + \frac{4}{t^3}\right] \\
&\leq V_k + 2\int_0^{+\infty} \exp\left(-\frac{t\Delta_k^2}{8\mu_k^2}\right)\,\mathrm{d}t + \int_0^{+\infty} \exp\left(-\frac{tp_1^2}{8}\right)\,\mathrm{d}t + 4\sum_{t=1}^{+\infty}\frac{1}{t^3} \\
&\leq V_k + 2\times\frac{8\mu_k^2}{p_k^2\Delta_k^2} + \frac{8}{p_1^2} + 4\times 1.5 \leq V_k + \frac{16}{p_k^2\Delta_k^2} + \frac{8}{p_1^2} + 6.
\end{aligned}
$$

Finally, by plugging the above into the decomposition $\mathcal{R}(T) = \sum_{k=2}^{K}\Delta_k\mathbb{E}c_k(T)$, we obtain the result in the theorem. $\qquad\square$

**Proof of problem-independent regret in heavy-tailed UCB:**

*Proof.* Still plug the bound for $\mathbb{E}c_k(T)$,

$$
\mathcal{R}(T) = \sum_{k=2}^{K}\Delta_k\mathbb{E}c_k(T) \leq \sum_{k=2}^{K}\Delta_k V_k + \sum_{k=2}^{K}\Delta_k\left(\frac{16}{p_k^2\Delta_k^2} + \frac{8}{p_1^2} + 6\right).
$$

For the second part, still apply Cauchy's inequality, we obtain

$$
\begin{aligned}
\sum_{k=2}^{K}\Delta_k\left(\frac{16}{p_k^2\Delta_k^2} + \frac{8}{p_1^2} + 6\right) &= \left(\sum_{\Delta_k < \Delta} + \sum_{\Delta_k \geq \Delta}\right)\Delta_k\left(\frac{16}{p_k^2\Delta_k^2} + \frac{8}{p_1^2} + 6\right) \\
&\leq T\Delta + \sum_{\Delta_k \geq \Delta}\left(\frac{8}{p_1^2} + 6\right)\Delta_k + \frac{1}{\Delta}\sum_{k=2}^{K}\frac{16}{p_k^2} \\
&\overset{\text{by Cauchy's inequality with } \Delta = \sqrt{\frac{16}{T}\sum_{k=2}^{K}\frac{1}{p_k^2}}}{\leq} 2K\left(\frac{4}{p_1^2} + 3\right) + 8\sqrt{T\sum_{k=2}^{K}p_k^{-2}}.
\end{aligned}
$$

For the first part, we use Hölder's inequality instead as follows:

$$
\sum_{k=2}^{K} \Delta_k V_k = \sum_{k=2}^{K} \Delta_k V_k^{\frac{1}{1+\epsilon}} V_k^{\frac{\epsilon}{1+\epsilon}}
$$

$$
\leq \sum_{k=2}^{K} \Delta_k V_k^{\frac{1}{1+\epsilon}} \left( \left(2 p_k g(p_k, \epsilon)\right)^{\frac{1+\epsilon}{\epsilon}} \frac{M^{\frac{1}{\epsilon}}}{\Delta_k^{\frac{1+\epsilon}{\epsilon}}} \log T \right)^{\frac{\epsilon}{1+\epsilon}}
$$

$$
\underset{\text{by Hölder's inequality}}{\leq} K^{\frac{\epsilon}{1+\epsilon}} \left( \sum_{k=2}^{K} V_k \right)^{\frac{1}{1+\epsilon}} \times \max_{k \in [K]} 2 p_k g(p_k, \epsilon) M^{\frac{1}{1+\epsilon}} \log^{\frac{\epsilon}{1+\epsilon}} T
$$

$$
\leq 2 \left( (1+\epsilon) 2^{\frac{\epsilon}{1+\epsilon}} + 10/3 \right) (MT)^{\frac{1}{1+\epsilon}} (K \log T)^{\frac{\epsilon}{1+\epsilon}} .
$$

Thus, we complete the proof of the problem-independent bound. □

# H. Proof of the regrets for TS-type algorithms

## H.1. Proof of Theorem 4.2

The primary proof idea behind our TS-type algorithm can be divided into two parts: first, controlling the overestimation of suboptimal arms is achieved through truncation in the clipped distribution, a technique we have already used in the proof of UCB-type algorithms with some modifications. Second, restricting the underestimation of the optimal arm can be accomplished through the anti-concentrations for the posterior distributions detailed in Appendix F.

*Proof.* Denote $\widetilde{\mu}_k(t)$ and $\widetilde{p}_k(t)$ is the posterior sample for the non-zero part in $k$-th arm at round $t$, and others use the same notations in the proof of Theorem 4.1. Define

$$
\Xi_\mu = \mu_1 - \min_{s \in [T]} \left\{ \widehat{\mu}_{1,s} + \sqrt{\frac{4\gamma \left[1 + \log^{-1}(1 + 1/\sqrt{sT})\right] \sigma^2}{\widehat{p}_{1,s}^2 s} \log^+ \left( \frac{4 \left[1 + \log^{-1}(1 + 1/\sqrt{sT})\right] \sigma^2 T}{\widehat{p}_{1,s}^2 s K} \right)} \right\}
$$

and

$$
\Xi_p = p_1 - \min_{s \in [T]} \left\{ \widehat{p}_{1,s} + \sqrt{\frac{\gamma}{4s} \log^+ \left( \frac{T}{4sK} \right)} \right\} .
$$

Let $\Xi = \Xi_\mu \times \Xi_p$, then we can decompose the regret as

$$
\mathcal{R}(T) = \sum_{k=2}^{K} \Delta_k \mathbb{E} c_k(T)
$$

$$
\leq \mathbb{E}[2T\Xi] + \mathbb{E} \left[ \sum_{k:\Delta_k \geq 2\Xi} \Delta_k c_k(T) \right] \tag{H.1}
$$

$$
\leq \mathbb{E}[2T\Xi] + 2e\sqrt{2KT} + \mathbb{E} \left[ \sum_{k:\Delta_k \geq (2\Xi) \vee (2e\sqrt{2p_k K/T})} \Delta_k c_k(T) \right] .
$$

For any $x \geq 0$, we have

$$
\mathbb{P}\left(\Xi \geq x\right) = \mathbb{P}\left(\Xi_\mu \times \Xi_p \geq x\right)
$$

$$
\leq \mathbb{P}\left(\Xi_\mu \geq x/2\right) + \mathbb{P}\left(\Xi_p \geq x/2\right) .
$$

By Lemma 2.2, we know that the Orlicz norm for $\widehat{\mu}_{1,s} - \mu_1$ satisfies

$$
\left\| \widehat{\mu}_{1,s} - \mu_1 \right\|_{\psi_2} \leq \frac{2\sigma}{p_1 \sqrt{s}} \tag{H.2}
$$

with probability $1 - \delta/2$ whenever $s \geq 4p_1^{-2} \log(2/\delta)$. Thus, by the same decomposition technique in proof of Theorem 4.1, and let $\delta_x$ be the Dirac delta function, we have

$$\mathbb{P}\left(\Xi_\mu \geq x/2\right)$$

$$= \mathbb{P}\left(\exists\, s \in [T] : \mu_1 - \right.$$

$$\left. \min_{1 \leq s \leq T} \left\{ \widehat{\mu}_{1,s} + \sqrt{ \frac{4\gamma \left[ 1 + \log^{-1}(1 + 1/\sqrt{sT}) \right] \sigma^2}{\widehat{p}_{1,s}^2 s} \log^+ \left( \frac{4 \left[ 1 + \log^{-1}(1 + 1/\sqrt{sT}) \right] \sigma^2 T}{\widehat{p}_{1,s}^2 s K} \right) } \right\} - \frac{x}{2} \geq 0 \right)$$

$$= \mathbb{P}\left(\exists\, s \in [T] : \mu_1 - \min_{1 \leq s \leq T} \left\{ \widehat{\mu}_{1,s} + \sqrt{ \frac{4\sigma^2 \gamma}{p_1^2 s} \log^+ \left( \frac{4\sigma^2 T}{p_1^2 s K} \right) } \right\} - \frac{x}{2} \geq 0 \right)$$

$$+ \mathbb{P}\left(\exists\, s \in [T] : \widehat{p}_{1,s} - p_1 \geq \frac{p_1}{\log(1 + 1/\sqrt{sT})} \right) \delta_x.$$

By Lemma 9.3 in (Lattimore & Szepesvári, 2020), we have

$$\mathbb{P}\left(\exists\, s \in [T] : \mu_1 - \min_{1 \leq s \leq T} \left\{ \widehat{\mu}_{1,s} + \sqrt{ \frac{4\sigma^2 \gamma}{p_1^2 s} \log^+ \left( \frac{4\sigma^2 T}{p_1^2 s K} \right) } \right\} - \frac{x}{2} \geq 0 \right) \leq \frac{15K}{T(x/2)^2} = \frac{60K}{x^2 T},$$

and by Hoeffding inequality, we have

$$\mathbb{P}\left(\exists\, s \in [T] : \widehat{p}_{1,s} - p_1 \geq \frac{p_1}{\log(1 + 1/\sqrt{sT})} \right) \leq \sum_{s=1}^{T} \exp\left\{ -\frac{2sp_1^2}{\log^2(1 + 1/\sqrt{sT})} \right\}$$

$$\leq \sum_{s=1}^{T} \exp\left\{ -\frac{2sp_1^2}{s^{-1}T^{-1}} \right\} \, \mathrm{d}s$$

$$\leq \frac{1}{p_1 \sqrt{T}} \int_0^\infty \exp\left\{ -2u^2 \right\} \, \mathrm{d}u = \frac{1}{p_1} \sqrt{\frac{\pi}{2T}}.$$

Similarly, one can obtain

$$\mathbb{P}\left(\Xi_p \geq x/2\right) = \mathbb{P}\left(\exists\, s \in [T] : p_1 - \min_{s \in [T]} \left\{ \widehat{p}_{1,s} + \sqrt{ \frac{\gamma}{4s} \log^+ \left( \frac{T}{4sK} \right) } \right\} - \frac{x}{2} \geq 0 \right) \leq \frac{60K}{x^2 T}$$

by $\widehat{p}_{1,s}$ is sub-Gaussian with variance proxy at most $\frac{1}{4s}$. Thus, for the first term in (H.1), we have

$$\mathbb{E}[2T\Xi] \leq 2T \int_0^{+\infty} \mathbb{P}\left(\Xi \geq x\right) \, \mathrm{d}x$$

$$\leq 2T \int_0^{+\infty} 1 \wedge \frac{120K}{x^2 T} \, \mathrm{d}x + \frac{2T}{p_1} \sqrt{\frac{\pi}{2T}} = 8\sqrt{30KT} + \frac{\sqrt{2\pi T}}{p_1}. \tag{H.3}$$

Now, consider the collection of the 'good' sets defined as

$$\mathcal{K} := \left\{ k \in [K] : \Delta_k \geq (2\Xi) \vee (2\mathrm{e}\sqrt{2p_k K/T}) \right\},$$

then by Theorem 36.2 in Lattimore & Szepesvári (2020) with $\epsilon = \Delta_k/2$ in $\mathcal{K}$, we have

$$\Delta_k \mathbb{E}c_k(T) \leq \Delta_k + \Delta_k \mathbb{E}\left[ \sum_{t=K+1}^{T} \mathbb{1}\left( A_t = k, E_k^c(t) \right) \right] + \Delta_k \mathbb{E}\left[ \sum_{s=1}^{T-1} \left( \frac{1}{G_{1,s}(\Delta_k/2)} - 1 \right) \right] \tag{H.4}$$

where

$$E_k^c(t) = \left\{ \widetilde{\mu}_k(t) \times \widetilde{p}_k(t) > r_1 - \frac{\Delta_k}{2} \right\}$$

and $G_{k,s}(\epsilon) = 1 - F_{k,s}(\mu_1 - \epsilon)$ with $F_{k,s}$ is the **non-clipped** full posterior of the $k$-th arm[1]. Note that $\widetilde{\mu}_k(t)$ comes from $\mathrm{cl}\mathcal{N}\left(\widehat{\mu}_k, \frac{2}{\rho_k c_k \widehat{p}_k}; \tau_k(t)\right)$

$$E_k^c(t) \subseteq \left\{ \tau_k(t) \times \zeta_k(t) > r_1 - \frac{\Delta_k}{2} \right\} = \left\{ \tau_k(t) \times \zeta_k(t) > r_k + \frac{\Delta_k}{2} \right\}.$$

Denote

$$\kappa_k(\mu) = \sum_{s=1}^{T} \mathbb{1}\left\{ \tau_k(s) > \mu_k + \frac{\Delta_k}{4p_k} \right\}$$

$$= \sum_{s=1}^{T} \mathbb{1}\left\{ \widehat{\mu}_{k,s} + \sqrt{\frac{4\gamma\left[1 + \log^{-1}(1 + 1/\sqrt{sT})\right]\sigma^2}{\widehat{p}_{k,s}^2 s} \log^+\left(\frac{4\left[1 + \log^{-1}(1 + 1/\sqrt{sT})\right]\sigma^2 T}{\widehat{p}_{k,s}^2 s K}\right)} > \mu_k + \frac{\Delta_k}{4p_k} \right\}$$

$$= \sum_{s=1}^{T} \mathbb{1}\left\{ \mathcal{G}_{k,s}(\mu) \right\},$$

and

$$\kappa_k(p) = \sum_{s=1}^{T} \mathbb{1}\left\{ \zeta_k(s) > p_k + \frac{p_k \Delta_k}{4r_k + \Delta_k} \right\}$$

$$= \sum_{s=1}^{T} \mathbb{1}\left\{ \widehat{p}_{k,s} + \sqrt{\frac{\gamma}{4s} \log^+\left(\frac{T}{4sK}\right)} > p_k + \frac{p_k \Delta_k}{4r_k + \Delta_k} \right\}$$

$$= \sum_{s=1}^{T} \mathbb{1}\left\{ \mathcal{G}_{k,s}(p) \right\},$$

then

$$\Delta_k \mathbb{E}\left[ \sum_{t=K+1}^{T} \mathbb{1}\left(A_t = k, E_k^c(t)\right) \right] \leq \Delta_k \mathbb{E}\left[ \sum_{t=K+1}^{T} \mathbb{1}\left(A_t = k\right)\mathbb{1}\left(\tau_k(t) \times \zeta_k(t) > r_k + \frac{\Delta_k}{2}\right) \right]$$

$$\leq \Delta_k \mathbb{E}\left[ \sum_{s=1}^{T} \mathbb{1}\left(\tau_k(s) \times \zeta_k(s) > r_k + \frac{\Delta_k}{2}\right) \right]$$

$$\leq \Delta_k \mathbb{E}\left[ \sum_{s=1}^{T} \mathbb{1}\left\{ \mathcal{G}_{k,s}(\mu) \cup \mathcal{G}_{k,s}(p) \right\} \right]$$

$$\leq \Delta_k \left[ \sum_{s=1}^{T} \mathbb{P}\left(\mathcal{G}_{k,s}(\mu)\right) + \sum_{s=1}^{T} \mathbb{P}\left(\mathcal{G}_{k,s}(p)\right) \right]$$

---

[1]There is a trick for converting the clipped distribution to the non-truncated full posterior distribution. We omit here, and the details can be seen in Jin et al. (2022).

Similar as we are dealing with the first part in (H.1), we have

$$
\begin{aligned}
&\sum_{s=1}^{T} \mathbb{P}\big(\mathcal{G}_{k,s}(\mu)\big) \\
&\leq \sum_{s=1}^{T} \mathbb{P}\left( \widehat{\mu}_{k,s} + \sqrt{\frac{4\sigma^2\gamma}{p_k^2 s} \log^+\left(\frac{4\sigma^2 T}{p_k^2 sK}\right)} > \mu_k + \frac{\Delta_k}{4p_k} \right) + \sum_{s=1}^{T} \mathbb{P}\left( \widehat{p}_{k,s} - p_k \geq \frac{p_k}{\log(1 + 1/\sqrt{sT})} \right) \\
&\leq \sum_{s=1}^{T} \mathbb{P}\left( \widehat{\mu}_{k,s} + \sqrt{\frac{4\sigma^2\gamma}{p_k^2 s} \log^+\left(\frac{4\sigma^2 T}{p_k^2 sK}\right)} > \mu_k + \frac{\Delta_k}{4p_k} \right) + \frac{1}{p_k}\sqrt{\frac{\pi}{2T}} \\
&\overset{\text{by Lemma 2 in Jin et al. (2021)}}{\leq} \frac{\Delta_k}{2p_k} + \frac{24 p_k}{\Delta_k} + \frac{8\gamma p_k}{\Delta_k}\left[\log^+\left(\frac{T\Delta_k^2}{8p_k^2 K}\right) + \sqrt{2\gamma\pi \log^+\left(\frac{T\Delta_k^2}{8p_k^2 K}\right)}\right] + \frac{1}{p_k}\sqrt{\frac{\pi}{2T}},
\end{aligned}
$$

and

$$
\begin{aligned}
\sum_{s=1}^{T} \mathbb{P}\big(\mathcal{G}_{k,s}(p)\big) &\leq \sum_{s=1}^{T} \mathbb{P}\left( \widehat{p}_{k,s} + \sqrt{\frac{\gamma}{4s}\log^+\left(\frac{T}{4sK}\right)} > p_k + \frac{p_k \Delta_k}{4r_k + \Delta_k} \right) \\
&\leq \sum_{s=1}^{T} \mathbb{P}\left( \widehat{p}_{k,s} - p_k + \sqrt{\frac{\gamma}{4s}\log^+\left(\frac{T}{4sK}\right)} > p_k \right) \\
&\overset{\text{by Lemma 2 in Jin et al. (2021)}}{\leq} p_k + \frac{12}{p_k} + \frac{4\gamma}{p_k}\left[\log^+\left(\frac{Tp_k^2}{4K}\right) + \sqrt{2\gamma\pi \log^+\left(\frac{Tp_k^2}{4K}\right)}\right].
\end{aligned}
$$

Note that $x^{-1}\log^+(ax^2)$ is decreasing for $x \geq e/\sqrt{a}$, we have

$$
\frac{1}{\Delta_k}\log^+\left(\frac{T\Delta_k^2}{8p_k^2 K}\right) \leq \frac{1}{e}\sqrt{\frac{T}{2p_k K}}
$$

for any $\Delta_k \geq 2e\sqrt{2p_k K/T}$ and

$$
\frac{1}{p_k}\log^+\left(\frac{Tp_k^2}{4K}\right) \leq \frac{1}{p_k}\log^+ e = \frac{1}{p_k}
$$

for any $T \geq 4eK$. Thus, we have the bounds on $k \in \mathcal{K}$ such that

$$
\begin{aligned}
\sum_{s=1}^{T} \mathbb{P}\big(\mathcal{G}_{k,s}(\mu)\big) &\leq \frac{\Delta_k}{2p_k} + \frac{24p_k}{\Delta_k} + \frac{8\gamma p_k}{\Delta_k}\left[\log^+\left(\frac{T\Delta_k^2}{8p_k^2 K}\right) + \sqrt{2\gamma\pi \log^+\left(\frac{T\Delta_k^2}{8p_k^2 K}\right)}\right] + \frac{1}{p_k}\sqrt{\frac{\pi}{2T}} \\
&\leq \frac{\Delta_k}{2p_k} + \frac{24p_k}{2e\sqrt{2p_k K/T}} + 8\gamma p_k\left[\frac{1}{e}\sqrt{\frac{T}{2p_k K}} + \sqrt{\frac{2\gamma\pi}{2e\sqrt{2p_k K/T}}\frac{1}{e}\sqrt{\frac{T}{2p_k K}}}\right] + \frac{1}{p_k}\sqrt{\frac{\pi}{2T}} \\
&\lesssim \frac{\Delta_k}{p_k} + \sqrt{\frac{p_k T}{K}} + \frac{1}{p_k\sqrt{T}}
\end{aligned}
$$

and

$$
\begin{aligned}
\sum_{s=1}^{T} \mathbb{P}\big(\mathcal{G}_{k,s}(p)\big) &\leq p_k + \frac{12}{p_k} + \frac{4\gamma}{p_k}\left[\log^+\left(\frac{Tp_k^2}{4K}\right) + \sqrt{2\gamma\pi \log^+\left(\frac{Tp_k^2}{4K}\right)}\right] \\
&\leq p_k + \frac{12}{p_k} + 4\gamma\left[\frac{1}{p_k} + \sqrt{\frac{2\gamma\pi}{p_k^2}}\right] \\
&\lesssim p_k + \frac{1}{p_k}.
\end{aligned}
$$

Therefore, we have

$$\Delta_k \mathbb{E}\left[\sum_{t=K+1}^{T} \mathbb{1}\left(A_t = k, E_k^c(t)\right)\right] \leq \Delta_k \sum_{s=1}^{T}\left[\mathbb{P}\left(\mathcal{G}_{k,s}(\mu)\right) + \mathbb{P}\left(\mathcal{G}_{k,s}(p)\right)\right]$$

$$\lesssim \sqrt{\frac{p_k T}{K}} + \frac{\Delta_k}{p_k \sqrt{T}} + \frac{\Delta_k}{p_k}.$$

It remains to deal with the last term in (H.4). For doing this, we will first prove the following result: for any $\epsilon > 0$, there exists a universal constant $c > 0$ such that

$$\mathbb{E}\left[\sum_{s=1}^{T-1}\left(\frac{1}{G_{1,s}(\epsilon)} - 1\right)\right] \leq \frac{c}{p_1^2 \epsilon^2}. \tag{H.5}$$

Indeed, let $Z_{k,s}$ be the random variable denoting the number of consecutive independent trails until a sample of the distribution $\mathcal{P}_{k,s} := \mathcal{N}\left(\widehat{\mu}_{k,s}, \frac{2\sigma^2}{\rho_k s \widehat{p}_{k,s}^2}\right) \times \text{Beta}(\alpha_{k,s}, \beta_{k,s})$ becomes greater than $r_1 - \epsilon$, then $\mathbb{E}\left[\frac{1}{G_{1,s}(\epsilon)} - 1\right] = \mathbb{E}\Upsilon_{1,s}$. Consider an integer $q \geq 1$ and $z \asymp \sqrt{\rho'}$ with some $\rho' \in (\rho, 1)$ determined later. Let $M_{k,q}$ be the maximum of $q$ independent samples from $\mathcal{P}_{k,s}$ and $\mathcal{F}_{k,s}$ be the filtration consisting the history of plays of Algorithm C.1 up to the $s$-th pull of arm 1. Then

$$\begin{aligned}
&\mathbb{P}(\Upsilon_{1,s} \leq q) \\
&\geq \mathbb{P}\left(M_{1,q} > r_1 - \epsilon\right) \\
&\geq \mathbb{E}\left[\mathbb{E}\left[M_{1,q} > \frac{\alpha_{1,s}\widehat{\mu}_{1,s}}{\alpha_{1,s} + \beta_{1,s}} + \frac{z}{\widehat{p}_{1,s}\sqrt{\rho s}}, \frac{\alpha_{1,s}\widehat{\mu}_{1,s}}{\alpha_{1,s} + \beta_{1,s}} + \frac{z}{\widehat{p}_{1,s}\sqrt{\rho s}} \geq r_1 - \epsilon \mid \mathcal{F}_{1,s}\right]\right] \\
&= \mathbb{E}\left[\mathbb{1}\left\{\frac{\alpha_{1,s}\widehat{\mu}_{1,s}}{\alpha_{1,s} + \beta_{1,s}} + \frac{z}{\widehat{p}_{1,s}\sqrt{\rho s}} \geq r_1 - \epsilon\right\}\mathbb{P}\left(M_{1,q} > \frac{\alpha_{1,s}\widehat{\mu}_{1,s}}{\alpha_{1,s} + \beta_{1,s}} + \frac{z}{\widehat{p}_{1,s}\sqrt{\rho s}} \mid \mathcal{F}_{1,s}\right)\right].
\end{aligned} \tag{H.6}$$

Then by Lemma F.4,

$$\begin{aligned}
&\mathbb{P}\left(M_{1,q} > \frac{\alpha_{1,s}\widehat{\mu}_{1,s}}{\alpha_{1,s} + \beta_{1,s}} + \frac{z}{\sqrt{\rho s}\widehat{p}_{1,s}} \mid \mathcal{F}_{1,s}\right) \\
&= \mathbb{P}\left(M_{1,q} > \frac{\alpha_1 + B_{1,s}}{\alpha_1 + \beta_1 + s}\widehat{\mu}_{1,s} + \frac{z}{\widehat{p}_{1,s}\sqrt{\rho s}} \mid \mathcal{F}_{1,s}\right) \\
&\overset{\text{by the choice of } z}{\geq} 1 - \left[1 - c(\alpha_1 + B_{1,s}, \beta_1 + s - B_{1,s})\frac{q^{-\rho'}}{\sqrt{2\pi}}\frac{\sqrt{8\rho'\log q}}{8\rho'\log q + 1} + \text{e}^{-\frac{2s}{p_1^2}}\right]^q \\
&\overset{\text{by fact (H.8)}}{\geq} 1 - \left[1 + \text{e}^{-\frac{2s}{p_1^2}} - \frac{\sqrt{1-p_1}}{2^{3+2\alpha_1+2\beta_1}\pi\text{e}} \times \text{e}^{-s(2-p_1)\log 2}\frac{q^{-\rho'}}{\sqrt{2\pi}}\frac{\sqrt{8\rho'\log q}}{8\rho'\log q + 1}\right]^q \\
&\overset{\text{by fact (H.10)}}{\geq} 1 - \left[1 - \frac{\sqrt{1-p_1}}{2^{4+2\alpha_1+2\beta_1}\pi\text{e}} \times \text{e}^{-s(2-p_1)\log 2}\frac{q^{-\rho'}}{\sqrt{2\pi}}\frac{\sqrt{8\rho'\log q}}{8\rho'\log q + 1}\right]^q \\
&\overset{\text{by } (1-x)^q \leq \text{e}^{-qx}}{\geq} 1 - \exp\left[-\frac{\sqrt{1-p_1}}{2^{4+2\alpha_1+2\beta_1}\pi\text{e}} \times \text{e}^{-s(2-p_1)\log 2}\frac{q^{1-\rho'}}{\sqrt{128\pi\log q}}\right] \\
&\geq 1 - \exp\left[-\frac{q^{1-\rho'}}{\sqrt{128\pi\log q}}\right]
\end{aligned} \tag{H.7}$$

for some $q \geq \text{e}^2$ and $\rho' > 1/2$ when we take

$$z = \frac{2\sigma(2\alpha_{1,s} + \beta_{1,s})\sqrt{2\rho'\log q}}{2(\alpha_{1,s} + \beta_{1,s})} \overset{\text{by fact (H.9)}}{\geq} \frac{\sigma(2\alpha_{1,s} + \beta_{1,s})\sqrt{2\rho'\log q} + \widehat{\mu}_{1,s}\beta_{1,s}}{2(\alpha_{1,s} + \beta_{1,s})}.$$

In the above, we use the the following three facts are

$$
c(\alpha_1 + B_{1,s}, \beta_1 + s - B_{1,s}) = \frac{\mathrm{B}^{-1}(\alpha_1 + B_{1,s}, \beta_1 + s - B_{1,s})}{(\beta_1 + s - B_{1,s})} \left[ \frac{\beta_1 + s - B_{1,s}}{2(\alpha_1 + \beta_1 + s)} \right]^{\beta_1 + s - B_{1,s}}
$$

$$
\times \left[ \frac{2\alpha_1 + \beta_1 + s + B_{1,s}}{2(\alpha_1 + \beta_1 + s)} \right]^{\alpha_1 + B_{1,s}} \left[ \frac{\beta_1 + s - B_{1,s} + 4}{2(\beta_1 + s - B_{1,s} + 1)} \right]
$$

$$
\overset{\text{by Stirling's formula}}{\geq} \frac{1}{2\pi \mathrm{e}(\beta_1 + (1-p_1)s)} \frac{(\alpha_1 + \beta_1 + s)^{\alpha_1 + \beta_1 + s - 1/2}}{(\alpha_1 + p_1 s)^{\alpha_1 + p_1 s - 1/2}(\beta_1 + (1-p_1)s)^{\beta_1 + (1-p_1)s - 1/2}}
$$

$$
\times \left[ \frac{\beta_1 + (1-p_1)s}{2(\alpha_1 + \beta_1 + s)} \right]^{\beta_1 + (1-p_1)s} \left[ \frac{2\alpha_1 + \beta_1 + (1+p_1)s}{2(\alpha_1 + \beta_1 + s)} \right]^{\alpha_1 + p_1 s} \times \frac{1}{2}
$$

$$
= \frac{4^{-s}}{4^{1+\alpha_1+\beta_1} \mathrm{e}\pi(\beta_1 + (1-p_1)s)} \frac{(\alpha_1 + \beta_1 + s)^{-1/2}}{(\beta_1 + (1-p_1)s)^{-1/2}} \times \left( \frac{2\alpha_1 + \beta_1 + (1+p_1)s}{\alpha_1 + p_1 s} \right)^{\alpha_1 + p_1 s}
$$

$$
\geq \frac{\sqrt{1-p_1}}{2^{3+2\alpha_1+2\beta_1} \mathrm{e}\pi} \times 4^{-s} \times 2^{\alpha_1 + p_1 s} = \frac{\sqrt{1-p_1}}{2^{3+2\alpha_1+2\beta_1} \mathrm{e}\pi} \times \mathrm{e}^{-s(2-p_1)\log 2}.
$$

(H.8)

for any $s \geq 1$,

$$
\sigma(2\alpha_{1,s} + \beta_{1,s})\sqrt{2\rho' \log q} \geq \widehat{\mu}_{1,s}\beta_{1,s}
$$

$$
\Longleftrightarrow \sqrt{2\rho' \log q} \geq \frac{\widehat{\mu}_{1,s}(\beta_1 + s - B_{1,s})}{\sigma(2\alpha_1 + \beta_1 + s + B_{1,s})}
$$

$$
\Longleftarrow \sqrt{2\rho' \log q} \geq \frac{\mu_1 + 2\sigma\mu_1}{\sigma} \quad \text{with probability at least } 1 - \mathrm{e}^{-\frac{(2\sigma\mu_1)^2 s}{2\sigma^2}}
$$

$$
\Longleftarrow q \geq \exp\left[ \frac{(2\sigma + 1)^2 r_1^2}{2\rho' p_1^2 \sigma^2} \right] \quad \text{with probability at least } 1 - \mathrm{e}^{-\frac{2r_1^2 s}{p_1^2}}
$$

$$
\Longleftarrow q \geq \exp\left[ \frac{(2\sigma + 1)^2}{2\rho' p_1^2 \sigma^2} \right] \quad \text{with probability at least } 1 - \mathrm{e}^{-\frac{2s}{p_1^2}},
$$

(H.9)

and

$$
\mathrm{e}^{-\frac{2s}{p_1^2}} \leq \frac{\sqrt{1-p_1}}{2^{4+2\alpha_1+2\beta_1} \mathrm{e}\pi} \times \mathrm{e}^{-s(2-p_1)\log 2} \frac{q^{1-\rho'}}{\sqrt{128\pi \log q}}
$$

$$
\overset{(2-p_1)\log 2 \leq \frac{2}{p_1^2}}{\Longleftarrow} 1 \leq \frac{\sqrt{1-p_1}}{2^{4+2\alpha_1+2\beta_1} \mathrm{e}\pi} \times \frac{q^{1-\rho'}}{\sqrt{128\pi \log q}}
$$

$$
\overset{\text{let } d_\alpha := \max_{x \geq 1} x^{-\alpha} \log x}{\Longleftarrow} 1 \leq \frac{\sqrt{1-p_1}}{2^{4+2\alpha_1+2\beta_1} \pi} \frac{q^{1-\rho'}}{\sqrt{128\pi d_{1-\rho'} q^{1-\rho'}}}
$$

$$
\Longleftrightarrow q \geq \left[ \frac{2^{7+2\alpha_1+2\beta_1+1/2}\pi^{3/2}\mathrm{e}}{\sqrt{(1-p_1)d_{1-\rho'}}} \right]^{\frac{2}{1-\rho'}}
$$

(H.10)

with $d_\alpha < \infty$ for any $\alpha > 0$. Now, take

$$
q \geq \mathrm{e}^2 \vee \exp\left[ \frac{(2\sigma + 1)^2}{2\rho' p_1^2 \sigma^2} \right] \vee \left[ \frac{2^{7+2\alpha_1+2\beta_1+1/2}\pi^{3/2}\mathrm{e}}{\sqrt{(1-p_1)d_{1-\rho'}}} \right]^{\frac{2}{1-\rho'}} \vee \exp\left[ \frac{160}{(1-\rho')^2} \right]
$$

(H.11)

in (H.7), we obtain that

$$
\mathbb{P}\left( M_{1,q} > \frac{\alpha_{1,s}\widehat{\mu}_{1,s}}{\alpha_{1,s} + \beta_{1,s}} + \frac{z}{\widehat{p}_{1,s}\sqrt{\rho s}} \,\Big|\, \mathcal{F}_{1,s} \right) \geq 1 - \frac{1}{q^2}.
$$

On the other hand,

$$
\mathbb{P}\left(\frac{\alpha_{1,s}\widehat{\mu}_{1,s}}{\alpha_{1,s}+\beta_{1,s}}+\frac{z}{\widehat{p}_{1,s}\sqrt{\rho s}}\geq r_1-\epsilon\right)
$$

$$
\geq\mathbb{P}\left(\frac{\alpha_{1,s}\widehat{\mu}_{1,s}}{\alpha_{1,s}+\beta_{1,s}}+\frac{2\sigma(2\alpha_{1,s}+\beta_{1,s})\sqrt{2\rho'\log q}}{2(\alpha_{1,s}+\beta_{1,s})\widehat{p}_{1,s}\sqrt{\rho s}}\geq\mu_1 p_1\right)
$$

$$
=\mathbb{P}\left(\widehat{\mu}_{1,s}-\mu_1+\frac{\sigma(2\alpha_{1,s}+\beta_{1,s})}{\alpha_{1,s}\widehat{p}_{1,s}}\sqrt{\frac{2\rho'\log q}{\rho s}}\geq\mu_1\left[\frac{\alpha_{1,s}+\beta_{1,s}}{\alpha_{1,s}}p_1-1\right]\right)
$$

$$
=\mathbb{P}\left(\underbrace{\widehat{\mu}_{1,s}-\mu_1+\frac{s\sigma[2\alpha_1+\beta_1+B_{1,s}]}{(\alpha_1+B_{1,s})B_{1,s}}\sqrt{\frac{2\rho'\log q}{\rho s}}\geq\mu_1\frac{-(1-p_1)\alpha_1+p_1\beta_1-(B_{1,s}-p_1 s)}{\alpha_1+B_{1,s}}}_{\text{event }\mathcal{U}_{1,s}}\right)
$$

$$
=\mathbb{P}\left(\mathcal{U}_{1,s}\mid B_{1,s}\geq p_1 s\right)\mathbb{P}(B_{1,s}\geq p_1 s)
$$

$$
+\sum_{v=1}^{+\infty}\mathbb{P}\left(\mathcal{U}_{1,s}\mid -\frac{s}{v}<B_{1,s}-p_1 s<-\frac{s}{v+1}\right)\mathbb{P}\left(-\frac{s}{v}<B_{1,s}-p_1 s<-\frac{s}{v+1}\right)
$$

$$
\geq\mathbb{P}\left(\widehat{\mu}_{1,s}-\mu_1+\frac{\sigma}{s^{-1}B_{1,s}}\sqrt{\frac{2\rho'\log q}{\rho s}}\geq 0\mid B_{1,s}\geq p_1 s\right)\mathbb{P}(B_{1,s}\geq p_1 s)
$$

$$
+\sum_{v=1}^{+\infty}\mathbb{P}\left(\widehat{\mu}_{1,s}-\mu_{1,s}+A_1(s,v)\sqrt{\frac{2\rho'\log q}{\rho s}}\geq\mu_1 A_2(s,v)\mid -\frac{s}{v}<B_{1,s}-p_1 s<-\frac{s}{v+1}\right)
$$

$$
\times\mathbb{P}\left(-\frac{s}{v}<B_{1,s}-p_1 s<-\frac{s}{v+1}\right)
$$

$$
\overset{\text{by (H.12) and (H.13)}}{\geq}\left(1-q^{-\rho'/\rho}\right)\mathbb{P}(B_{1,s}\geq p_1 s)+\sum_{v=1}^{+\infty}\left(1-e^{2p_1^{-4}}q^{-2p_1^{-1}\rho'/\rho}\right)\mathbb{P}\left(-\frac{s}{v}<B_{1,s}-p_1 s<-\frac{s}{v+1}\right)
$$

$$
\geq 1-q^{-\rho'/\rho}-e^{2p_1^{-4}}q^{-2p_1^{-1}\rho'/\rho},
$$

where we define

$$
A_1(s,v):=\frac{\sigma[s^{-1}(2\alpha_1+\beta_1)+s^{-1}B_{1,s}]}{s^{-1}B_{1,s}[s^{-1}\alpha_1+s^{-1}B_{1,s}]},\qquad A_2(s,v):=\frac{s^{-1}[-(1-p_1)\alpha_1+p_1\beta_1]-(s^{-1}B_{1,s}-p_1)}{s^{-1}\alpha_1+s^{-1}B_{1,s}}.
$$

We also use the facts that

$$
\mathbb{P}\left(\widehat{\mu}_{1,s}-\mu_1+\frac{\sigma}{s^{-1}B_{1,s}}\sqrt{\frac{2\rho'\log q}{\rho s}}\geq 0\mid B_{1,s}\geq p_1 s\right)
$$

$$
=1-\mathbb{P}\left(\mu_1-\widehat{\mu}_{1,s}\geq\frac{\sigma}{s^{-1}B_{1,s}}\sqrt{\frac{2\rho'\log q}{\rho s}}\mid B_{1,s}\geq p_1 s\right) \tag{H.12}
$$

$$
\overset{\text{by the fact that }B_{1,s}(\widehat{\mu}_{1,s}-\mu_1)\sim\text{subG}(B_{1,s}\sigma^2)\text{ for any }B_{1,s}\geq 1}{\geq}1-\exp\left[-\frac{\rho'\log q}{\rho s^{-1}B_{1,s}}\right]
$$

$$
\overset{\text{by }s^{-1}B_{1,s}\leq 1}{\geq}1-q^{-\rho'/\rho},
$$

and similarly

$$\mathbb{P}\left(\widehat{\mu}_{1,s} - \mu_1 + A_1(s,v)\sqrt{\frac{2\rho'\log q}{\rho s}} \geq \mu_1 A_2(s,v) \mid -\frac{s}{v} < B_{1,s} - p_1 s < -\frac{s}{v+1}\right)$$

$$\overset{\text{by } (1-p_1)\alpha_1 \leq p_1\beta_1}{\geq} \mathbb{P}\left(\widehat{\mu}_{1,s} - \mu_1 + A_1(s,v)\sqrt{\frac{2\rho'\log q}{\rho s}} \geq \frac{\mu_1(p_1 - s^{-1}B_{1,s})}{s^{-1}\alpha_1 + s^{-1}B_{1,s}} \mid -\frac{s}{v} < B_{1,s} - p_1 s < -\frac{s}{v+1}\right)$$

$$\overset{\text{by similar trick in } (\text{H}.12)}{\geq} 1 - \sup_{s^{-1}B_{1,s} \in [p_1 - v^{-1}, p_1 - (v+1)^{-1}]} \exp\left\{-\frac{B_{1,s}}{2\sigma^2}\left[\frac{\mu_1(p_1 - s^{-1}B_{1,s})}{s^{-1}\alpha_1 + s^{-1}B_{1,s}} - A_1(s,v)\sqrt{\frac{2\rho'\log q}{\rho s}}\right]^2\right\}$$

$$\overset{\text{by the fact } (\text{H}.14)}{\geq} 1 - \sup_{s^{-1}B_{1,s} \in [p_1 - v^{-1}, p_1 - (v+1)^{-1}]} \exp\left[\frac{v^{-2}\mu_1^2}{[p_1 + (v+1)^{-1}]^2} - \frac{B_{1,s}}{2\sigma^2}A_1^2(s,v)\frac{2\rho'\log q}{\rho s}\right]$$

$$\overset{\text{by the fact } (\text{H}.15)}{\geq} 1 - \exp\left[\frac{v^{-2}\mu_1^2}{[p_1 + (v+1)^{-1}]^2} - \frac{1}{p_1 - \frac{1}{v+1}}\frac{\rho'\log q}{\rho s}\right]$$

$$\overset{\text{by the fact } (\text{H}.16)}{\geq} 1 - \mathrm{e}^{2p_1^{-4}}q^{-2p_1^{-1}\rho'/\rho}$$

$$\tag{H.13}$$

where the fact we used is

$$\left[\frac{\mu_1(p_1 - s^{-1}B_{1,s})}{s^{-1}\alpha_1 + s^{-1}B_{1,s}}\right]^2 - 2A_1(s,v)\frac{\mu_1(p_1 - s^{-1}B_{1,s})}{s^{-1}\alpha_1 + s^{-1}B_{1,s}}\sqrt{\frac{2\rho'\log q}{\rho s}}$$

$$= \frac{\mu_1(p_1 - s^{-1}B_{1,s})}{[s^{-1}\alpha_1 + s^{-1}B_{1,s}]^2}\left[\mu_1(p_1 - s^{-1}B_{1,s}) - \frac{2\sigma[s^{-1}(2\alpha_1 + \beta_1) + s^{-1}B_{1,s}]}{s^{-1}B_{1,s}}\sqrt{\frac{2\rho'\log q}{\rho s}}\right] \tag{H.14}$$

$$\leq \frac{\mu_1\frac{1}{v}}{\left[p_1 + \frac{1}{v+1}\right]^2}\frac{\mu_1}{v} = \frac{v^{-2}\mu_1^2}{[p_1 + (v+1)^{-1}]^2},$$

$$\frac{B_{1,s}}{2\sigma^2}A_1^2(s,v)\frac{2\rho'\log q}{\rho s} = \frac{[s^{-1}(2\alpha_1 + \beta_1) + s^{-1}B_{1,s}]^2}{s^{-1}B_{1,s}[s^{-1}\alpha_1 + s^{-1}B_{1,s}]^2}\frac{\rho'\log q}{\rho s}$$

$$\geq \frac{1}{s^{-1}B_{1,s}}\frac{\rho'\log q}{\rho s} \tag{H.15}$$

$$\overset{\text{by } s^{-1}B_{1,s} \leq p_1 - \frac{1}{v+1}}{\geq} \frac{1}{p_1 - \frac{1}{v+1}}\frac{\rho'\log q}{\rho s},$$

and

$$\frac{v^{-2}\mu_1^2}{[p_1 + (v+1)^{-1}]^2} - \frac{1}{p_1 - \frac{1}{v+1}}\frac{\rho'\log q}{\rho s} \leq \frac{2\mu_1^2}{p_1^2} - \frac{2}{p_1}\frac{\rho'\log q}{\rho s} \tag{H.16}$$

uniformly on $v, s \geq 1$ and $s^{-1}B_{1,s} \in [p_1 - v^{-1}, p_1 - (v+1)^{-1}]$. Therefore, by plugging these inequalities into (H.6), we conclude that

$$\mathbb{P}(\Upsilon_{1,s} < q) \leq 1 - q^{-2} - q^{-\rho'/\rho} - \mathrm{e}^{2p_1^{-4}}q^{-2p_1^{-1}\rho'/\rho}$$

for any $q$ satisfied (H.11), and then

$$\mathbb{E}\Upsilon_{1,s} = \sum_{q=0}^{+\infty}\mathbb{P}(\Upsilon_{1,s} \geq q)$$

$$\leq \mathrm{e}^2 + \exp\left[\frac{(2\sigma+1)^2}{2\rho'p_1^2\sigma^2}\right] + \left[\frac{2^{7+2\alpha_1+2\beta_1+1/2}\pi^{3/2}\mathrm{e}}{\sqrt{(1-p_1)d_{1-\rho'}}}\right]^{\frac{2}{1-\rho'}} + \exp\left[\frac{160}{(1-\rho')^2}\right]$$

$$+ \sum_{q=1}^{+\infty}\left[q^{-2} + q^{-\rho'/\rho} + \mathrm{e}^{2p_1^{-4}}q^{-2p_1^{-1}\rho'/\rho}\right]$$

$$\leq \mathrm{e}^2 + \exp\left[\frac{(2\sigma+1)^2}{2\rho'p_1^2\sigma^2}\right] + \left[\frac{2^{7+2\alpha_1+2\beta_1+1/2}\pi^{3/2}\mathrm{e}}{\sqrt{(1-p_1)d_{1-\rho'}}}\right]^{\frac{2}{1-\rho'}} + \exp\left[\frac{160}{(1-\rho')^2}\right]$$

$$+ 1 + \frac{1}{1-\rho'/\rho} + \frac{\mathrm{e}^{2p_1^{-4}}}{1-2p_1^{-1}\rho'/\rho}.$$

for any $s \in \mathbb{N}$. Let $2p_1^{-1}\rho'/\rho = 1/2$, we immediately conclude that

$$\mathbb{E}\left[\frac{1}{G_{1,s}(\epsilon)} - 1\right] = \mathbb{E}\Upsilon_{1,s} \leq c \tag{H.17}$$

with $c = c(p_1, \rho, \sigma^2)$ is fully determined by $p_1, \rho, \sigma^2$ and free of $s$. Now, let $\mathcal{E}_{1,s} = \{\widehat{\mu}_{1,s} \times \widehat{p}_{1,s} > r_1 - \epsilon/2\}$, then

$$\mathbb{P}\big(\Upsilon_{1,s} > r_1 - \epsilon \mid \mathcal{E}_{1,s}\big)$$
$$\geq \mathbb{P}\big(\Upsilon_{1,s} > \widehat{\mu}_{1,s} \times \widehat{p}_{1,s} - \epsilon/2 \mid \mathcal{E}_{1,s}\big)$$
$$= 1 - \mathbb{P}\Big(\widehat{\mu}_{1,s} \times \widehat{p}_{1,s} > \mathcal{N}\big(\widehat{\mu}_{1,s}, 2\sigma^2/(\rho_k s \widehat{p}_{1,s}^2)\big) \times \mathrm{Beta}(\alpha_{1,s}, \beta_{1,s}) + \epsilon/2 \mid \mathcal{E}_{1,s}\Big)$$
$$\geq 1 - \Big[\mathbb{P}\Big(\widehat{\mu}_{1,s} > \mathcal{N}\big(\widehat{\mu}_{1,s}, 2\sigma^2/(\rho s \widehat{p}_{1,s}^2)\big) + \epsilon/(2p_1) \mid \mathcal{E}_{1,s}\Big)$$
$$\qquad + \mathbb{P}\Big(\widehat{p}_{1,s} > \mathrm{Beta}(\alpha_{1,s}, \beta_{1,s}) + p_1\epsilon/(4r_1 + \epsilon)\Big)\Big]. \tag{H.18}$$

Note that we have

$$\mathbb{P}\Big(\widehat{\mu}_{1,s} > \mathcal{N}\big(\widehat{\mu}_{1,s}, 2\sigma^2/(\rho s \widehat{p}_{1,s}^2)\big) + \epsilon/(2p_1) \mid \mathcal{E}_{1,s}\Big)$$

$$= \mathbb{P}\left(B_{1,s}\widehat{\mu}_{1,s} - \mathcal{N}\left(B_{1,s}\widehat{\mu}_{1,s}, \frac{2s\sigma^2}{\rho}\right) > \frac{\epsilon B_{1,s}}{2p_1} \mid \mathcal{E}_{1,s}\right)$$

$$\overset{\text{by the inequality does not rely anything on } \mathcal{E}_{1,s}}{\leq} \frac{1}{2}\mathbb{E}\exp\left[-\frac{\epsilon^2 B_{1,s}^2}{4p_1^2} \times \frac{\rho}{4s\sigma^2}\right]$$

$$= \frac{1}{2}\left[\mathbb{E}\left[\exp\left(-\frac{s\rho\epsilon^2}{16p_1\sigma^2}\big(s^{-1}B_{1,s}\big)^2\right) \mid s^{-1}B_{1,s} > p_1/2\right]\mathbb{P}\big(s^{-1}B_{1,s} > p_1/2\big)\right.$$

$$\left. + \mathbb{E}\left[\exp\left(-\frac{s\rho\epsilon^2}{16p_1\sigma^2}\big(s^{-1}B_{1,s}\big)^2\right) \mid s^{-1}B_{1,s} \leq p_1/2\right]\mathbb{P}\big(s^{-1}B_{1,s} \leq p_1/2\big)\right]$$

$$\leq \frac{1}{2}\left[\exp\left(-\frac{s\rho\epsilon^2}{64p_1\sigma^2}\right) + \mathbb{P}\big(0 < s^{-1}B_{1,s} \leq p_1/2\big)\right]$$

$$\overset{\text{by Theorem 2 in (Ahle, 2017)}}{\leq} \frac{1}{2}\left[\exp\left(-\frac{s\rho\epsilon^2}{64p_1\sigma^2}\right) + \exp\big(-sd(p_1/2, p_1)\big)\right]$$

$$= \frac{1}{2}\left[\exp\left(-\frac{s\rho\epsilon^2}{64p_1\sigma^2}\right) + \exp\left(-s\Big(\frac{p_1}{2}\log\frac{1-p_1}{2-p_1} + \log\frac{2-p_1}{2-2p_1}\Big)\right)\right]$$

$$\overset{\text{by } \frac{x}{1+x} \leq \log x}{\leq} \frac{1}{2}\left[\exp\left(-\frac{s\rho\epsilon^2}{64p_1\sigma^2}\right) + \exp\big(-sp_1(1-p_1/2)\big)\right],$$

and

$$\mathbb{P}\Big(\widehat{p}_{1,s} > \text{Beta}(\alpha_{1,s}, \beta_{1,s}) + p_1\epsilon/(4r_1 + \epsilon)\Big)$$

$$= \mathbb{P}\left(\text{Beta}(\alpha_{1,s}, \beta_{1,s}) - \mathbb{E}\,\text{Beta}(\alpha_{1,s}, \beta_{1,s}) < \frac{(\alpha_1 + \beta_1)B_{1,s} - \alpha_1 s}{(\alpha_1 + \beta_1 + s)s} - \frac{p_1\epsilon}{4r_1 + \epsilon}\right)$$

$$\overset{\text{when } s \geq \frac{2(4+\epsilon)}{p_1\epsilon}}{\leq} \mathbb{P}\left(\text{Beta}(\alpha_{1,s}, \beta_{1,s}) - \mathbb{E}\,\text{Beta}(\alpha_{1,s}, \beta_{1,s}) < -\frac{p_1\epsilon}{2(4+\epsilon)}\right)$$

$$\leq 2\exp\left[-\frac{p_1^2\epsilon^2}{18(4+\epsilon)\,[(4+\epsilon)+4p_1\epsilon]}s\right],$$

where we use the result in Theorem 1 of Skorski (2023): if $\alpha_{1,s} \geq \beta_{1,s}$, we have

$$\mathbb{P}\left(\text{Beta}(\alpha_{1,s}, \beta_{1,s}) - \mathbb{E}\,\text{Beta}(\alpha_{1,s}, \beta_{1,s}) < -\frac{p_1\epsilon}{2(4+\epsilon)}\right)$$

$$\leq \exp\left[-\left(\frac{p_1\epsilon}{2(4+\epsilon)}\right)^2 \frac{(\alpha_{1,s} + \beta_{1,s})^2(\alpha_{1,s} + \beta_{1,s} + 1)}{2\alpha_{1,s}\beta_{1,s}}\right]$$

$$= \exp\left[-\left(\frac{p_1\epsilon}{2(4+\epsilon)}\right)^2 \frac{(\alpha_1 + \beta_1 + s)^2(1 + \alpha_1 + \beta_1 + s)}{2(\alpha_1 + B_{1,s})(\beta_1 + s - B_{1,s})}\right]$$

$$\leq \exp\left[-\left(\frac{p_1\epsilon}{2(4+\epsilon)}\right)^2 \frac{s^3}{2(\alpha_1 + s/2)(\beta_1 + s/2)}\right]$$

$$\overset{\text{by } \alpha_1, \beta_1 \leq 1}{\leq} \exp\left[-\frac{p_1^2\epsilon^2}{18(4+\epsilon)^2}s\right];$$

and if $\alpha_{1,s} < \beta_{1,s}$, we have

$$\mathbb{P}\left( \mathrm{Beta}(\alpha_{1,s}, \beta_{1,s}) - \mathbb{E}\,\mathrm{Beta}(\alpha_{1,s}, \beta_{1,s}) < -\frac{p_1\epsilon}{2(4+\epsilon)} \right)$$

$$\leq \exp\left[ -\frac{p_1^2\epsilon^2}{8(4+\epsilon)^2} \left[ \frac{\alpha_{1,s}\beta_{1,s}}{(\alpha_{1,s}+\beta_{1,s})^2(\alpha_{1,s}+\beta_{1,s}+1)} + \frac{2(\beta_{1,s}-\alpha_{1,s})}{3(\alpha_{1,s}+\beta_{1,s})(\alpha_{1,s}+\beta_{1,s}+2)}\frac{p_1\epsilon}{2(4+\epsilon)} \right]^{-1} \right]$$

$$= \exp\left[ -\frac{p_1^2\epsilon^2}{8(4+\epsilon)^2} \left[ \frac{(\alpha_1+B_{1,s})(\beta_1+s-B_{1,s})}{(\alpha_1+\beta_1+s)^2(1+\alpha_1+\beta_1+s)} + \frac{2(\beta_1-\alpha_1+s-B_{1,s})}{3(\alpha_1+\beta_1+s)(2+\alpha_1+\beta_1+s)}\frac{p_1\epsilon}{2(4+\epsilon)} \right]^{-1} \right]$$

$$\leq \exp\left[ -\frac{p_1^2\epsilon^2}{8(4+\epsilon)^2} \left[ \frac{(\alpha_1+s/2)(\beta_1+s/2)}{s^3} + \frac{p_1\epsilon(\beta_1-\alpha_1)}{3(4+\epsilon)s^2} \right]^{-1} \right]$$

$$\leq \exp\left[ -\frac{p_1^2\epsilon^2}{8(4+\epsilon)^2} \left[ \frac{9s^2}{4s^3} + \frac{p_1\epsilon}{3(4+\epsilon)s^2} \right]^{-1} \right]$$

$$\leq \exp\left[ -\frac{p_1^2\epsilon^2}{18(4+\epsilon)\left[(4+\epsilon)+4p_1\epsilon\right]}s \right].$$

By plugging these inequalities into (H.18), we obtain

$$\mathbb{P}\big(\Upsilon_{1,s} > r_1 - \epsilon \mid \mathcal{E}_{1,s}\big)$$
$$\geq 1 - \frac{1}{2}\left[ \exp\left( -\frac{s\rho\epsilon^2}{64p_1\sigma^2} \right) + \exp\big( -sp_1(1-p_1/2) \big) \right] - 2\exp\left[ -\frac{p_1^2\epsilon^2}{18(4+\epsilon)\left[(4+\epsilon)+4p_1\epsilon\right]}s \right].$$

On the other hand, the probability of $\mathcal{E}_{1,s}$ can be directly bounded through the concentrations for Gaussian and Binomial distribution as

$$\mathbb{P}(\mathcal{E}_{1,s}) = \mathbb{P}\big(\widehat{\mu}_{1,s} \times \widehat{p}_{1,s} > r_1 - \epsilon/2\big)$$
$$\geq 1 - \left[ \mathbb{P}\big(\mu_1 > \widehat{\mu}_{1,s} + \epsilon/(2p_1)\big) + \mathbb{P}\big(p_1 > \widehat{p}_{1,s} + p_1\epsilon/(4r_1+\epsilon)\big) \right]$$
$$\geq 1 - \exp\left[ -\frac{s\epsilon^2}{8p_1^2\sigma^2} \right] - \exp\left[ -\frac{2sp_1^2\epsilon^2}{(4+\epsilon)^2} \right]$$

by the facts that

$$\mathbb{P}\big(\mu_1 > \widehat{\mu}_{1,s} + \epsilon/(2p_1)\big) = \mathbb{P}\big(\mu_1 - \widehat{\mu}_{1,s} > \epsilon/(2p_1)\big) \leq \exp\left( -\frac{s\epsilon^2}{8p_1^2\sigma^2} \right)$$

and

$$\mathbb{P}\big(p_1 > \widehat{p}_{1,s} + p_1\epsilon/(4r_1+\epsilon)\big) = \mathbb{P}\big(p_1 - \widehat{p}_{1,s} > p_1\epsilon/(4r_1+\epsilon)\big) \leq \exp\left( -\frac{2sp_1^2\epsilon^2}{(4r_1+\epsilon)^2} \right).$$

Note that $\rho \in (1/2, 1)$, we have $\exp\left(-\frac{s\rho\epsilon^2}{64p_1\sigma^2}\right)$, $\exp(-sp_1(1-p_1/2))$, $\exp\left(-\frac{sp_1^2\epsilon^2}{18(4+\epsilon)[(4+\epsilon)+4p_1\epsilon]}\right)$, $\exp\left(-\frac{s\epsilon^2}{8p_1^2\sigma^2}\right)$, and $\exp\left(-\frac{2sp_1^2\epsilon^2}{(4r_1+\epsilon)^2}\right)$ are all less than $\exp\left(-\frac{sp_1^2\epsilon^2}{(4+\epsilon)^2(1\vee\sigma^2)}\right)$. Then, by combining the inequalities for $\mathbb{P}\big(\Upsilon_{1,s} > r_1 - \epsilon \mid \mathcal{E}_{1,s}\big)$ and $\mathbb{P}(\mathcal{E}_{1,s})$, we have

$$\mathbb{P}\big(\Upsilon_{1,s} > r_1 - \epsilon \mid \mathcal{E}_{1,s}\big)\mathbb{P}(\mathcal{E}_{1,s})$$
$$\geq \left[ 1 - \exp\left( -\frac{sp_1^2\epsilon^2}{(4+\epsilon)^2(1\vee\sigma^2)} \right) - 2\exp\left( -\frac{sp_1^2\epsilon^2}{(4+\epsilon)^2(1\vee\sigma^2)} \right) \right]\left[ 1 - 2\exp\left( -\frac{sp_1^2\epsilon^2}{(4+\epsilon)^2(1\vee\sigma^2)} \right) \right]$$
$$\geq \left[ 1 - 3\exp\left( -\frac{sp_1^2\epsilon^2}{(4+\epsilon)^2(1\vee\sigma^2)} \right) \right]^2.$$

Thus, if we take $L \geq \frac{(4+\epsilon)^2(1\vee\sigma^2)}{p_1^2\epsilon^2}\log(3(1-1/\sqrt{2})^{-1})$, it will yield

$$\mathbb{E}\left[\sum_{s=L}^{T}\left(\frac{1}{G_{1,s}(\epsilon)}-1\right)\right] \leq \mathbb{E}\left[\sum_{s=L}^{T}\left(\frac{1}{\mathbb{P}(\Upsilon_{1,s}>r_1-\epsilon\mid\mathcal{E}_{1,s})\mathbb{P}(\mathcal{E}_{1,s})}-1\right)\right]$$

$$\leq \sum_{s=L}^{T}\left[\left[1-3\exp\left(-\frac{sp_1^2\epsilon^2}{(4+\epsilon)^2(1\vee\sigma^2)}\right)\right]^{-2}-1\right]$$

$$\overset{\text{by }(1-3x)^{-2}-1\leq 12x\text{ for any }x<1/3-1/(3\sqrt{2})\text{ and the condition for }L}{\leq} 12\sum_{s=L}^{T}\exp\left(-\frac{sp_1^2\epsilon^2}{(4+\epsilon)^2(1\vee\sigma^2)}\right)$$

$$\leq 12\int_{L}^{+\infty}\exp\left(-\frac{sp_1^2\epsilon^2}{(4+\epsilon)^2(1\vee\sigma^2)}\right)\,\mathrm{d}s + 12\exp\left(-\frac{Lp_1^2\epsilon^2}{(4+\epsilon)^2(1\vee\sigma^2)}\right)$$

$$= 12\exp\left(-\frac{Lp_1^2\epsilon^2}{(4+\epsilon)^2(1\vee\sigma^2)}\right)\left[1+\frac{(4+\epsilon)^2(1\vee\sigma^2)}{p_1^2\epsilon^2}\right]$$

$$\overset{\text{by the condition for }L}{\leq} 4\left(1-1/\sqrt{2}\right)\left[1+\frac{(4+\epsilon)^2(1\vee\sigma^2)}{p_1^2\epsilon^2}\right].$$

The above inequality together inequality (H.17) implies (H.5) immediately. Therefore, we obtain the last term in (H.4) is bounded by

$$\Delta_k\mathbb{E}\left[\sum_{s=1}^{T-1}\left(\frac{1}{G_{1,s}(\Delta_k/2)}-1\right)\right] \lesssim \Delta_k\frac{4}{p_1^2\Delta_k^2}$$

$$\overset{\text{by }\Delta_k\geq(2\Xi)\vee(2\mathrm{e}\sqrt{2p_kK/T})}{\lesssim} \frac{1}{p_1\sqrt{p_k}}\sqrt{\frac{T}{K}},$$

and therefore,

$$\Delta_k\mathbb{E}c_k(T) \leq \Delta_k + \Delta_k\mathbb{E}\left[\sum_{t=K+1}^{T}\mathbb{1}\left(A_t=k,E_k^c(t)\right)\right] + \Delta_k\mathbb{E}\left[\sum_{s=1}^{T-1}\left(\frac{1}{G_{1,s}(\Delta_k/2)}-1\right)\right]$$

$$\lesssim \Delta_k + \sqrt{\frac{p_kT}{K}} + \frac{\Delta_k}{p_k\sqrt{T}} + \frac{\Delta_k}{p_k} + \frac{1}{p_1\sqrt{p_k}}\sqrt{\frac{T}{K}}$$

$$\asymp \frac{\Delta_k}{p_k} + \frac{1}{p_1\sqrt{p_k}}\sqrt{\frac{T}{K}}.$$

By substituting the above inequality and (H.3) back into (H.1), we obtain the result stated in the theorem. $\square$

## I. Proof of Regret Bounds for Contextual Bandit Algorithms

Let $\mathbf{V}_t := \sum_{\ell=1:Y_\ell=1}^{t}\mathbf{Z}_\ell\mathbf{Z}_\ell^\top$ and $\mathbf{U}_t := \sum_{\ell=1}^{t}\mathbf{W}_\ell\mathbf{W}_\ell^\top$ be the sample variance for the non-zero part and binary part at round $t$, respectively. The proof for contextual bandits is divided into three main steps after decomposing the regret into two periods.

The first step is to find the minimal round $\tau$ such that the minimal eigenvalue of $\mathbf{V}_t$ and $\mathbf{U}_t$ satisfies $\min\{\lambda_{\min}(\mathbf{V}_t),\lambda_{\min}(\mathbf{U}_t)\} \geq 1$ for any $t \geq \tau$ with a high probability, which bounds the regret in the first $\tau$ periods. The second step uses the self-normalized martingale result to bound the regret from $\tau$ to $T$ rounds. The final step combines the regret over both periods.

*Proof.* **Step 0:** We start with decomposing the regret. Assume $A_t^*$ is the optimal action at round $t$, and let $\mathbf{Z}_t^* := \psi_X(\mathbf{x}_t,A_t^*)$

and $p_{A_t} := h\big(\psi_Y(\mathbf{x}_t, A_t)^\top \boldsymbol{\theta}\big)$, then

$$
\begin{aligned}
\mathcal{R}(T) &= \sum_{t=1}^{T} \Big[ p_{A_t^*} g(\boldsymbol{\beta}^{*\top} \mathbf{Z}_t^*) - p_{A_t} g(\boldsymbol{\beta}^{*\top} \mathbf{Z}_t) \Big] \\
&= \sum_{t=1}^{\tau} \Big[ p_{A_t^*} g(\boldsymbol{\beta}^{*\top} \mathbf{Z}_t) - p_{A_t} g(\boldsymbol{\beta}^{*\top} \mathbf{Z}_t^*) \Big] + \sum_{t=\tau+1}^{T} \Big[ p_{A_t^*} g(\boldsymbol{\beta}^{*\top} \mathbf{Z}_t^*) - p_{A_t} g(\widehat{\boldsymbol{\beta}}_t^\top \mathbf{Z}_t) \Big] \qquad \text{(I.1)} \\
&\leq \big[ 2L_g + g(0) \big] \tau + \sum_{t=\tau+1}^{T} \Big[ p_{A_t^*} g(\boldsymbol{\beta}^{*\top} \mathbf{Z}_t^*) - \widehat{p}_{A_t} g(\widehat{\boldsymbol{\beta}}_t^\top \mathbf{Z}_t) \Big]
\end{aligned}
$$

where the last inequality is by

$$
\begin{aligned}
&\sum_{t=1}^{\tau} \Big[ p_{A_t^*} g(\boldsymbol{\beta}^{*\top} \mathbf{Z}_t^*) - p_{A_t} g(\boldsymbol{\beta}^{*\top} \mathbf{Z}_t) \Big] \\
&= \sum_{t=1}^{\tau} \Big\{ \Big[ p_{A_t^*} g(\boldsymbol{\beta}^{*\top} \mathbf{Z}_t^*) - p_{A_t^*} g(\boldsymbol{\beta}^{*\top} \mathbf{Z}_t) \Big] + \Big[ p_{A_t^*} g(\boldsymbol{\beta}^{*\top} \mathbf{Z}_t) - p_{A_t} g(\boldsymbol{\beta}^{*\top} \mathbf{Z}_t) \Big] \Big\} \\
&\overset{\text{by Assumption C.1 (ii)}}{\leq} \sum_{t=1}^{\tau} \Big\{ L_g \big| \boldsymbol{\beta}^{*\top} \mathbf{Z}_t^* - \boldsymbol{\beta}^{*\top} \mathbf{Z}_t \big| + \Big[ L_g \big| \boldsymbol{\beta}^{*\top} \mathbf{Z}_t - 0 \big| + g(0) \Big] (p_{A_t^*} - p_{A_t}) \Big\} \\
&\leq \sum_{t=1}^{\tau} \Big\{ L_g \sqrt{\|\boldsymbol{\beta}^*\|_2^2 \|\mathbf{Z}_t^* - \mathbf{Z}_t\|_2^2} + \Big[ L_g \sqrt{\|\boldsymbol{\beta}^*\|_2^2 \|\mathbf{Z}_t\|_2^2} + g(0) \Big] (p_{A_t^*} - \widehat{p}_{A_t^*}) \Big\} \\
&\overset{\text{by Assumption C.1 (i)}}{\leq} \big[ 2L_g + g(0) \big] \tau
\end{aligned}
$$

and $\tau \in \mathbb{N}$ will be determined later.

For each arm $k \in [K]$, let $\mathbf{V}_k(m) := \sum_{t=1:Y_{t,k}=1}^{m} \mathbf{Z}_{t,k} \mathbf{Z}_{t,k}^\top$ is the sample variance for the non-zero part and $B_k(m) := \{t \in [m] : Y_{t,k} = 1\}$ of the arm $k$ with pulling it $m$ times.

**Step 1:** Let $\boldsymbol{\Sigma}_k := \mathbb{E}[\mathbf{Z}_{t,k} \mathbf{Z}_{t,k}^\top]$ and $\boldsymbol{\Omega}_k := \mathbb{E}[\mathbf{W}_{t,k} \mathbf{W}_{t,k}^\top]$ be the variance matrices for the i.i.d sample $\psi_X(\mathbf{x}_t, A_k)$ and $\psi_Y(\mathbf{x}_t, A_k)$, respectively. Note $\mathbf{V}_k(m)$ has the same distribution with

$$
\sum_{i=1}^{B_k(m)} \mathbf{Z}_{i,k} \mathbf{Z}_{i,k}^\top,
$$

due to $X_t$ is independent with $Y_t$ for any fixed $t \in [m]$. Then by applying Proposition 1 in Li et al. (2017), for any $\delta \in (0,1)$ there exist positive, universal constants $C_1$ and $C_2$ such that

$$
\lambda_{\min}\big(\mathbf{V}_k(m)\big) \geq 1
$$

with probability at least $1 - \delta$, as long as

$$
B_k(m) \geq \left( \frac{C_1 \sqrt{d} + C_2 \sqrt{\log(1/\delta)}}{\lambda_{\min}(\boldsymbol{\Sigma}_k)} \right)^2 + \frac{2}{\lambda_{\min}(\boldsymbol{\Sigma}_k)}.
$$

Note that $B_k(m)$ is the summation of independent Bernoulli$(p_{k,t})$ variables with $p_{k,t} \geq p_*$ we have $B_k(m) \geq mp_*/2$ with probability at least $1 - \delta$ whenever $m \geq 4p_*^{-2}$ by Hoeffding's inequality.

Similarly, we also apply Proposition 1 in Li et al. (2017) for $\mathbf{U}_k(m) = \sum_{t=1}^{m} \mathbf{W}_{t,k} \mathbf{W}_{t,k}^\top$, and know that there exist some universal $C_3, C_4$ constants such that $\lambda_{\min}\big(\mathbf{U}_k(m)\big) \geq 1$ whenever

$$
m \geq \left( \frac{C_3 \sqrt{q} + C_4 \sqrt{\log(1/\delta)}}{\lambda_{\min}(\boldsymbol{\Omega}_k)} \right)^2 + \frac{2}{\lambda_{\min}(\boldsymbol{\Omega}_k)}.
$$

Now, by noticing Assumption C.2 (ii), we let

$$\tau_k = \max \left\{ \frac{2}{p_*^2} \left[ \left( \frac{C_1 \sqrt{d} + C_2 \sqrt{\log(1/\delta)}}{\lambda_{\min}(\mathbf{\Sigma}_k)} \right)^2 + \frac{2}{\lambda_{\min}(\mathbf{\Sigma}_k)} \right], \frac{4 \log(1/\delta)}{p_*^2}, \left( \frac{C_3 \sqrt{q} + C_4 \sqrt{\log(1/\delta)}}{\lambda_{\min}(\mathbf{\Omega}_k)} \right)^2 + \frac{2}{\lambda_{\min}(\mathbf{\Omega}_k)} \right\}.$$

then we have

$$\min \left\{ \lambda_{\min}(\mathbf{V}_k(\tau_k)), \lambda_{\min}(\mathbf{U}_k(\tau_k)) \right\} \geq 1$$

with probability at least $1 - 3\delta$.

**Step 2.1** Define

$$\mathbf{V}_n = \sum_{k \in [K]} \mathbf{V}_k(m_k) + \lambda_V \mathbf{I}_d$$

with $n$ is the round such that for arm $k$ pulled $m_k$ times. Then from **Step 1** and the fact that $\tau = \max_{k \in [K]} \tau_k$ by Assumption C.2 (ii), we know that for any $n \geq \tau + 1$, the minimal eigenvalue of $\mathbf{V}_n$ and $\mathbf{U}_n$ satisfies

$$\lambda_{\min}(\mathbf{V}_n) \geq 1 + \lambda_V, \qquad \text{and} \qquad \lambda_{\min}(\mathbf{U}_n) \geq 1 + \lambda_U$$

with probability at least $1 - 3\delta$. Next, since $\mathbf{V}_{t+1} = \mathbf{V}_t + Y_t \cdot \mathbf{Z}_t \mathbf{Z}_t^\top$,

$$\det \mathbf{V}_{t+1} = \det \mathbf{V}_t \times \det \left( \mathbf{I}_d + Y_t \cdot \mathbf{V}_t^{-1/2} \mathbf{Z}_t \mathbf{Z}_t^\top \mathbf{V}_t^{-1/2} \right)$$

which furthermore implies

$$\begin{aligned}
\log \det \mathbf{V}_T &= \log \left( \det \mathbf{V}_\tau \prod_{t=\tau+1:Y_t=1}^{T} (1 + \|\mathbf{Z}_t\|_{\mathbf{V}_t^{-1}}^2) \right) \\
&= \log \det \mathbf{V}_\tau + \sum_{t=\tau+1:Y_t=1}^{T} \log \left( 1 + \|\mathbf{Z}_t\|_{\mathbf{V}_t^{-1}}^2 \right) \\
&\geq \log \det \mathbf{V}_\tau + \frac{1}{2} \sum_{t=\tau+1:Y_t=1}^{T} \left( 1 \wedge \|\mathbf{Z}_t\|_{\mathbf{V}_t^{-1}}^2 \right) \\
&= \log \det \mathbf{V}_\tau + \frac{1}{2} \sum_{t=\tau+1:Y_t=1}^{T} \|\mathbf{Z}_t\|_{\mathbf{V}_t^{-1}}^2,
\end{aligned}$$

where the last step is due to $\|\mathbf{Z}_t\|_2 \leq 1$ and $\lambda_{\min}(\mathbf{V}_t) > 1$ for $t \geq \tau + 1$. Therefore, we conclude that

$$\sum_{t=\tau+1:Y_t=1}^{T} \|\mathbf{Z}_t\|_{\mathbf{V}_t^{-1}} \overset{\text{by Cauchy}}{\leq} \sqrt{T \sum_{t=\tau+1:Y_t=1}^{T} \|\mathbf{Z}_t\|_{\mathbf{V}_t^{-1}}^2} \leq \sqrt{2T \log \frac{\det \mathbf{V}_T}{\det \mathbf{V}_\tau}}. \tag{I.2}$$

Note that by the inequality of arithmetic and geometric means,

$$\det \mathbf{V}_T \leq \left( \frac{\text{tr}(\mathbf{V}_T)}{d} \right)^d \leq \left( \frac{\text{tr}(\mathbf{V}_\tau) + T - \tau}{d} \right)^d,$$

and the fact that

$$\text{tr}(\mathbf{V}_\tau) \leq \lambda_V + \tau \quad \text{and} \quad \det \mathbf{V}_\tau \geq (1 + \lambda_V)^d.$$

Our inequality (I.2) can be furthermore bounded by

$$\sum_{t=\tau+1:Y_t=1}^{T} \|\mathbf{Z}_t\|_{\mathbf{V}_t^{-1}} \leq \sqrt{2T \log \frac{\det \mathbf{V}_T}{\det \mathbf{V}_\tau}} \leq \sqrt{2Td \log \left( \frac{\lambda_V + T}{d(1 + \lambda_V)} \right)}. \tag{I.3}$$

Similarly, we have

$$\sum_{t=\tau+1}^{T} \|\mathbf{W}_t\|_{\mathbf{U}_t^{-1}} \leq \sqrt{2Tq \log\left(\frac{\lambda_U + T}{q(1 + \lambda_U)}\right)}, \tag{I.4}$$

as $Y_t$ would always be observed, which is a specific case above.

**Step 2.2** Let $\Psi_{X,t}(\boldsymbol{\beta}) := \sum_{\ell=1:Y_\ell=1}^{t} \left[g(\mathbf{Z}_\ell^\top \boldsymbol{\beta}) - g(\mathbf{Z}_\ell^\top \boldsymbol{\beta}^*)\right] \mathbf{Z}_\ell$. By Assumption C.1 (iii), we have

$$\|\Psi_{X,t}(\boldsymbol{\beta})\|_{\mathbf{V}_t^{-1}}^2 \geq \kappa_g^2 \|\boldsymbol{\beta} - \boldsymbol{\beta}^*\|_2^2 \tag{I.5}$$

for any $\boldsymbol{\beta} \in \Gamma \subseteq \{\boldsymbol{\beta} : \|\boldsymbol{\beta} - \boldsymbol{\beta}^*\|_2 \leq 1\}$. Next, note that $\widehat{\boldsymbol{\beta}}_t$ directly comes solving $\sum_{\ell=1:Y_\ell=1}^{t} \left[X_\ell - g(\mathbf{Z}_\ell^\top \boldsymbol{\beta})\right] \mathbf{Z}_\ell = \mathbf{0}$, then

$$\begin{aligned}
\Psi_{X,t}(\widehat{\boldsymbol{\beta}}_t) &= \sum_{\ell \in [t]:Y_\ell=1} \left[g(\mathbf{Z}_\ell^\top \widehat{\boldsymbol{\beta}}_t) - g(\mathbf{Z}_\ell^\top \boldsymbol{\beta}^*)\right] \mathbf{Z}_\ell \\
&= \sum_{\ell \in [t]:Y_\ell=1} \left[g(\mathbf{Z}_\ell^\top \widehat{\boldsymbol{\beta}}_t) - g(\mathbf{Z}_\ell^\top \boldsymbol{\beta}^*)\right] \mathbf{Z}_\ell + \sum_{\ell \in [t]:Y_\ell=1} \left[X_\ell - g(\mathbf{Z}_\ell^\top \widehat{\boldsymbol{\beta}}_t)\right] \mathbf{Z}_\ell \\
&= \sum_{\ell \in [t]:Y_\ell=1} \left[X_\ell - g(\mathbf{Z}_\ell^\top \boldsymbol{\beta}^*)\right] \mathbf{Z}_\ell = \sum_{\ell \in [t]:Y_\ell=1} \varepsilon_\ell \mathbf{Z}_\ell.
\end{aligned}$$

Note that $\varepsilon_\ell \mid \mathcal{F}_{\ell-1} \sim \text{subG}(\sigma^2)$ directly implies $\varepsilon_\ell \mid \sigma\langle Y_s = 1 : s \in [\ell]\rangle \cap \mathcal{F}_{\ell-1} \sim \text{subG}(\sigma^2)$. Let

$$\frac{1}{2\sigma^2} \|\Psi_{X,t}(\widehat{\boldsymbol{\beta}}_t)\|_{\mathbf{V}_t^{-1}}^2 = \max_{\mathbf{v} \in \mathbb{R}^d} \left(\mathbf{v}^\top \sum_{\ell \in [t]:Y_\ell=1} \sigma^{-1} \varepsilon_\ell \mathbf{Z}_\ell - \frac{1}{2} \|\mathbf{v}\|_{\mathbf{V}_t}^2\right) := \max_{\mathbf{v} \in \mathbb{R}^d} M_{X,t}(\mathbf{v}),$$

then $\overline{M}_{X,t}(\mathbf{v}) := \int M_{X,t}(\mathbf{v}) \, d\mu(\mathbf{v})$ is also an $\mathbb{F}$-adapted non-negative super-martingale with initial value $= 1$ for any probability measure $\mu(\mathbf{v})$ on $\mathbb{R}^d$. Thus, by Theorem 3.9 in Lattimore & Szepesvári (2020), we have

$$\mathbb{P}\left(\sup_{t \in \mathbb{N}} \overline{M}_{X,t}(\mathbf{v}) \geq 1/\delta\right) \leq \frac{\mathbb{E}\overline{M}_{X,0}(\mathbf{v})}{1/\delta} = \delta$$

for the probability measure $\mu = \mathcal{N}(0, \lambda_U^{-1} \mathbf{I}_d)$. Plugging in the explicit formula for $\overline{M}_{X,t}(\mathbf{v})$, we obtain that

$$\mathbb{P}\left(\exists t \in \mathbb{N} : \frac{1}{2\sigma^2} \|\Psi_{X,t}(\widehat{\boldsymbol{\beta}}_t)\|_{\mathbf{V}_t^{-1}}^2 \geq 2\log(1/\delta) + \log\left(\frac{\det \mathbf{V}_t}{\lambda_V^d}\right)\right) \leq \delta \tag{I.6}$$

for any $\lambda_U > 0$ and $\delta \in (0, 1)$. Combining inequality (I.5) and (I.6), we conclude that

$$\begin{aligned}
\|\widehat{\boldsymbol{\beta}}_t - \boldsymbol{\beta}^*\|_2 &\leq \kappa_g^{-1} \sqrt{2} \sigma \sqrt{2\log(1/\delta) + \log\left(\frac{\det \mathbf{V}_t}{\lambda_V^d}\right)} \\
&\leq \kappa_g^{-1} \sigma \sqrt{4\log(1/\delta) + d\log(1 + \lambda_V^{-1} t/d)} = \rho_{X,t}
\end{aligned}$$

holds with probability at least $1 - \delta$ for any $t \geq \tau + 1$, where the last inequality is due to

$$\frac{\det \mathbf{V}_t}{\lambda_V^d} \leq \left(\text{tr}\left(\frac{\mathbf{V}_t}{\lambda_V d}\right)\right)^d \leq \left(1 + \frac{\{\ell \in [t] : Y_\ell = 1\}}{\lambda_V d}\right)^d \leq \left(1 + \frac{t}{\lambda_V d}\right)^d.$$

By using the similar argument, we can also show that

$$\|\widehat{\boldsymbol{\theta}}_t - \boldsymbol{\theta}^*\|_2 \leq \kappa_h^{-1} \sqrt{4\log(1/\delta) + q\log(1 + \lambda_U^{-1} t/q)} = \rho_{Y,t}$$

holds with probability at least $1 - \delta$ for any $t \geq \tau + 1$.

**Step 2.3** The design of Algorithm C.2 for choosing $A_t \in [K]$ ensures

$$\widehat{\boldsymbol{\beta}}_t^\top \mathbf{Z}_t^* + \rho_{X,t} \|\mathbf{Z}_t^*\|_{\mathbf{V}_t^{-1}} \leq \widehat{\boldsymbol{\beta}}_t^\top \mathbf{Z}_t + \rho_{X,t} \|\mathbf{Z}_t\|_{\mathbf{V}_t^{-1}},$$

i.e.,

$$\widehat{\boldsymbol{\beta}}_t^\top (\mathbf{Z}_t^* - \mathbf{Z}_t) \le \rho_{X,t} \big( \|\mathbf{Z}_t\|_{\mathbf{V}_t^{-1}} - \|\mathbf{Z}_t^*\|_{\mathbf{V}_t^{-1}} \big)$$

and

$$\widehat{\boldsymbol{\theta}}_t^\top \mathbf{W}_t^* + \rho_{Y,t} \|\mathbf{W}_t^*\|_{\mathbf{U}_t^{-1}} \le \widehat{\boldsymbol{\theta}}_t^\top \mathbf{W}_t + \rho_{Y,t} \|\mathbf{W}_t\|_{\mathbf{U}_t^{-1}},$$

i.e.,

$$\widehat{\boldsymbol{\theta}}_t^\top (\mathbf{W}_t^* - \mathbf{W}_t) \le \rho_{Y,t} \big( \|\mathbf{W}_t\|_{\mathbf{U}_t^{-1}} - \|\mathbf{W}_t^*\|_{\mathbf{U}_t^{-1}} \big).$$

Apply the last two inequalities in **Step 2.2**, we obtain

$$
\begin{aligned}
(\mathbf{Z}_t^* - \mathbf{Z}_t)^\top \boldsymbol{\beta}^* &= (\mathbf{Z}_t^* - \mathbf{Z}_t)^\top \widehat{\boldsymbol{\beta}}_t - (\mathbf{Z}_t^* - \mathbf{Z}_t)^\top (\widehat{\boldsymbol{\beta}}_t - \boldsymbol{\beta}^*) \\
&\le \rho_{X,t} \big( \|\mathbf{Z}_t\|_{\mathbf{V}_t^{-1}} - \|\mathbf{Z}_t^*\|_{\mathbf{V}_t^{-1}} \big) + \|\mathbf{Z}_t^* - \mathbf{Z}_t\|_{\mathbf{V}_t^{-1}} \|\widehat{\boldsymbol{\beta}}_t - \boldsymbol{\beta}^*\|_2 \\
&\overset{\text{by the choose of } \rho_{X,t} \text{ and } t > \tau}{\le} \rho_{X,t} \big( \|\mathbf{Z}_t\|_{\mathbf{V}_t^{-1}} - \|\mathbf{Z}_t^*\|_{\mathbf{V}_t^{-1}} + \|\mathbf{Z}_t^* - \mathbf{Z}_t\|_{\mathbf{V}_t^{-1}} \big) \\
&\le 2\rho_{X,t} \|\mathbf{Z}_t\|_{\mathbf{V}_t^{-1}}.
\end{aligned}
$$

with probability at least $1 - 4\delta$ for any $t \ge \tau + 1$. Similarly, by applying the same technique for $(\mathbf{W}_t^* - \mathbf{W}_t)^\top \boldsymbol{\theta}^*$, we can obtain that

$$(\mathbf{Z}_t^* - \mathbf{Z}_t)^\top \boldsymbol{\beta}^* \le 2\rho_{X,t} \|\mathbf{Z}_t\|_{\mathbf{V}_t^{-1}} \qquad \text{and} \qquad (\mathbf{W}_t^* - \mathbf{W}_t)^\top \boldsymbol{\theta}^* \le 2\rho_{Y,t} \|\mathbf{W}_t\|_{\mathbf{U}_t^{-1}}, \tag{I.7}$$

both hold with probability at least $1 - 5\delta$ for any $t \ge \tau + 1$.

**Step 3:** Now, with the inequality (I.7), we could deal with the remaining part in (I.1) in **Step 0**. By following the same technique for proving $\sum_{t=1}^{\tau} [p_{A_t^*} g(\boldsymbol{\beta}^{*\top} \mathbf{Z}_t) - \widehat{p}_{A_t} g(\widehat{\boldsymbol{\beta}}_t^\top \mathbf{Z}_t^*)] \le [2L_g + g(0)]\tau$, we have

$$
\begin{aligned}
&\sum_{t=\tau+1}^{T} \Big[ p_{A_t^*} g(\boldsymbol{\beta}^\top \mathbf{Z}_t^*) - p_{A_t} g(\boldsymbol{\beta}^\top \mathbf{Z}_t) \Big] \\
&\overset{\text{by Assumption C.1 (ii)}}{\le} \sum_{t=\tau+1}^{T} \Big\{ L_g \big( \boldsymbol{\beta}^{*\top} \mathbf{Z}_t^* - \boldsymbol{\beta}^{*\top} \mathbf{Z}_t \big) + \Big[ L_g |\boldsymbol{\beta}^{*\top} \mathbf{Z}_t - 0| + g(0) \Big] L_h \big( \boldsymbol{\theta}^{*\top} \mathbf{W}_t^* - \boldsymbol{\theta}^{*\top} \mathbf{W}_t \big) \Big\} \\
&\overset{\text{by (I.7)}}{\le} 2L_g \sum_{t=\tau+1}^{T} \rho_{X,t} \|\mathbf{Z}_t\|_{\mathbf{V}_t^{-1}} + 2(L_g + g(0)) L_h \sum_{t=\tau+1}^{T} \rho_{Y,t} \|\mathbf{W}_t\|_{\mathbf{U}_t^{-1}} \\
&\le 2L_g \rho_{X,T} \sum_{t=\tau+1}^{T} \|\mathbf{Z}_t\|_{\mathbf{V}_t^{-1}} + 2(L_g + g(0)) L_h \rho_{Y,T} \sum_{t=\tau+1}^{T} \|\mathbf{W}_t\|_{\mathbf{U}_t^{-1}} \\
&\overset{\text{by (I.3) and (I.4)}}{\le} 2\kappa_g^{-1} \sigma L_g \sqrt{4\log(1/\delta) + d\log(1 + \lambda_V^{-1} T/d)} \sqrt{2Td \log\left( \frac{\lambda_V + T}{d(1 + \lambda_V)} \right)} \\
&\qquad + 2\kappa_h^{-1}(L_g + g(0)) L_h \sqrt{4\log(1/\delta) + q\log(1 + \lambda_U^{-1} t/q)} \sqrt{2Tq \log\left( \frac{\lambda_U + T}{q(1 + \lambda_U)} \right)},
\end{aligned}
$$

which completes the regret bound for our UCB algorithm. The regret bound for the TS algorithm follows similarly, leveraging the anti-concentration result in (C.2) along with the proof techniques in Theorem 1 of Agrawal & Goyal (2013). Specifically, it incorporates anti-concentration results for $(\widetilde{\boldsymbol{\beta}}_t - \widehat{\boldsymbol{\beta}}_t)(\mathbf{Z}_t^* - \mathbf{Z}_t)$ and $(\widetilde{\boldsymbol{\theta}}_t - \widehat{\boldsymbol{\theta}}_t)(\mathbf{W}_t^* - \mathbf{W}_t)$. $\qquad \square$

