# OpenReview forum: "Zero-Inflated Bandits"
_ICML.cc/2025/Conference — ICML 2025 poster_

### Official Review · Reviewer_AjGD · 2025-03-07

**Overall Recommendation:** 4

**Summary:**

This paper considers stochastic multi-armed and contextual bandits where the reward distributions are zero-inflated distributions. Formally, each reward observation is a draw of a product random variable $R_t = X_tY_t$ where $X_t$ is a distribution with mean $\mu$ and $Y_t$ is a Bernoulli distribution with parameter $p$. It is assumed that $\mu$ and $p$ are unknown and both may vary across arms.

The paper proposes UCB and TS algorithms for various versions of these problems, with accompanying theoretical and empirical analysis.

The main innovation in the design of UCB and TS algorithms is to not form a single confidence bound for $p\mu$ the expected value of $R_t$, or draw a single TS sample from it, but to form separate indices for $\mu$ and $p$ and combine these, since a quantities based on the entire zero-inflated distribution may scale unnecessarily in $\mu$. The more substantial challenge lies in the analysis, where handling a random number of observations of the non-zero component presents additional complications over the standard analysis used in parametric bandit proofs.

# POST REBUTTAL: Satisfied with proposed modifications and retain a positive score.

**Claims And Evidence:**

Yes

**Essential References Not Discussed:**

See above.

**Experimental Designs Or Analyses:**

I felt the experiments were appropriate, and checked the specification in Appendix C, but had some observations about the presentation of results:

-	There doesn’t appear to be an explanation of what the figures are showing in the Experiment section. Are the plotted lines means or medians, are the vertical lines plus/minus standard deviations, or quantiles, or max/mins? Why do some curves appear not to have error bars at all?
-	There are elements of the experimental setup that are unclear. E.g. for the MAB problems, what are the parameters of the Gaussian/Mixed Gaussian/Exponential components, I could only find details of p.

**Methods And Evaluation Criteria:**

Yes

**Other Comments Or Suggestions:**

Minor observations:
-	L038: a chapter or page reference to Lattimore and Szepesvari would be helpful here, as done on L068 and L430.
-	L154: clarify the context in which it is commonly assumed – I presume this means in design and analysis of bandit algorithms (but the rest of the paragraph was about defining sub-Weibulls so it’s not the clearest).
-	L177: Error with the reference to Corollary “??”
-	A lot of equations are squashed into in-line text, presumably to save vertical space in the 8 page template. I’d recommend using some of the post-acceptance additional allowance to remedy this so things are easier to read, especially on pages 4 and 5.
-	L238: I’m not sure the less than equal to notation used in Theorem 4.1 is defined?


A proof read for grammar would improve the readability of the paper, e.g.:
-	L048: distributions*
-	L076: follow*
-	L078: games*
-	L115: follow*
-	L146: prone to be heavily influenced by their estimation errors*
-	L234-5: We note*
-	L230: established used as an adjective here suggests they already exist and or not a new contribution, I think you mean that you establish these results in this paper?

**Other Strengths And Weaknesses:**

The paper provides a thorough treatment of the zero-inflated bandit problem, covering MAB and contextual problems, TS and UCB-based approaches, and problem-dependent and problem-independent results. The experiments are sensible to evaluate these algorithms and the work is well motivated and connected to the literature and potential avenues for future work. Aspects of the theoretical and methodological contribution may indeed be translations of existing tools to this new setting, but I think there is sufficient innovation here to interest an ICML audience.

**Questions For Authors:**

Mostly repeating questions from above for ease of reference:

-	Sections 1 and 2 refer to Figure 1 (a) and (b) but without a full explanation of what the figures show. For instance, it’s not clear how 1 (a) provides an example of bandit algorithms failing to utilize the distribution property, and how the confidence bounds in 1 (b) are constructed. Could this be rectified?
-	Can you give a more detailed sense of how Lemma 2.2 and A.1 resolve the issues of handling the distribution of X in the main text? It seems this is one of the main contributions, and while we get a good sense of the challenge, and the fact that these lemmas resolve it, there isn’t a good sense of how within the main text.
-	Similarly in Section 4.2, it’s hard to tell whether there is novelty in the proofs that are seconded to appendix, and if so how much. I think it would help readers to understand the theoretical contribution if this could be concisely expressed in the main text.
-	There doesn’t appear to be an explanation of what the figures are showing in the Experiment section. Are the plotted lines means or medians, are the vertical lines plus/minus standard deviations, or quantiles, or max/mins? Why do some curves appear not to have error bars at all?
-	There are elements of the experimental setup that are unclear. E.g. for the MAB problems, what are the parameters of the Gaussian/Mixed Gaussian/Exponential components, I could only find details of p.

**Relation To Broader Scientific Literature:**

I felt this was mostly done appropriately, it seems that

-	Liu et al. (2023, arxiv:2311.14349) Thompson sampling for zero-inflated count outcomes with an application to the Drink Less mobile health study.

is worth mentioning but it was the only other relevant paper on zero-inflation in bandits I could find.

**Theoretical Claims:**

I could not read the entirety of the supplementary material in the time available, but I checked the results leading to Lemma 2.2 in detail and did not find issues. I did however feel that the theoretical contribution could be better explained in the main text:

-	Can you give a more detailed sense of how Lemma 2.2 and A.1 resolve the issues of handling the distribution of X in the main text? It seems this is one of the main contributions, and while we get a good sense of the challenge, and the fact that these lemmas resolve it, there isn’t a good sense of how within the main text.
-	Similarly in Section 4.2, it’s hard to tell whether there is novelty in the proofs that are seconded to appendix, and if so how much. I think it would help readers to understand the theoretical contribution if this could be concisely expressed in the main text.

---

> ### Author Rebuttal · Authors · 2025-03-31
>
> We sincerely appreciate your thoughtful and encouraging review. Below, we provide point-by-point responses to your comments. We look forward to refining our manuscript to address these valuable suggestions.
>
> *Theoretical Claims:*
>
> We sincerely thank you for your thoughtful and encouraging comment. As you rightly pointed out, obtaining valid concentration bounds for the non-zero component $X$ is a core challenge in both algorithm design and regret analysis. Lemma 2.2 (sub-Weibull noise) and Lemma A.1 (heavy-tailed noise) are indeed key technical contributions that enable us to build confidence intervals. By decoupling the ZI mechanism from the tail structure of X, these lemmas preserve tightness without imposing overly conservative assumptions.
>
> We also appreciate your suggestion regarding Section 4.2. Due to space limitations, we deferred many details of the regret analysis to the appendix, including several nontrivial proof techniques that are tailored to the ZI structure and product-form rewards. In the revision, we will explicitly outline in the main text which parts of the analysis are standard and which are novel, to help readers more clearly appreciate the theoretical contribution.
>
> *Experimental Designs & Analyses:*
>
> All plots show mean cumulative regret over 25 independent runs, with shading indicating $\pm 1 / 10$ standard deviation. Sometimes the shaded region is nearly invisible when variability is small. We will update figure captions and the main text to clarify this, ensuring the error bars' scale is apparent.
>
> To improve reproducibility, we will explicitly detail our reward models:
>
> - Gaussian: Nonzero parts from a $N(\mu_k, 1)$, with $\mu_k \sim U(0, 100)$;
>
> - Mixed Gaussian rewards: The non-zero rewards from
> $$
> (1 - p_k) \times N \left( \frac{\mu_k}{2(1 - p_k)}, \sigma^2\right) + p_k \times N \left( \frac{\mu_k}{2p_k}, \sigma^2\right)
> $$
> which ensures that the overall mean of the non-zero component is $\mu_k$.
>
> - Exponential: The non-zero rewards from an exponential distribution with mean $\mu_k$, where $\mu_k \sim U(0, 100)$.
>
> We will include these details in Section 5 and Appendix C of our revised submission.
>
> *Relation To Broader Scientific Literature:*
>
> Thank you for bringing [1] to our attention. We will include this citation in our introduction, noting that it treats zero-inflated count outcomes via Poisson/negative binomial models. Our method accommodates more general real-valued ZI rewards (Gaussian, exponential, heavy-tailed, etc.). Despite this difference in scope, we recognize the close connection and will highlight its relevance.
>
> *Other Comments & Suggestions:*
>
> - L038: We will include the specific reference to Chapter 9 in [1], consistent with the other citations in the manuscript.
>
> - L154: Yes, the assumption refers to a common modeling assumption for reward distributions in the design and analysis of bandit algorithms. We will clarify this in the revised version.
>
> -L177: This was intended to refer to a corollary comparing constant terms in regret bounds between our method and the canonical UCB algorithm. We appreciate the reviewer catching this and will correct it in the final version.
>
> - R238: The ``less than or equal to" symbol ($\lesssim$) is used to indicate inequality up to constants independent of bandit specific parameters.
>
> - R230: We appreciate the clarification. Our intention was to say that these results are established within this paper. We will revise the sentence to avoid ambiguity.
>
> We also appreciate your suggestions regarding inline equations, grammar, and formatting. We will use the post-acceptance phase to improve readability, split long inline equations, and proofread thoroughly.
>
> *Question:*
>
> We appreciate your helpful question. The purpose of Figure 1(a) is to present a motivating real-world example (introduced in [L81]) that illustrates the prevalence of zero rewards in practice. This highlights the limitations of traditional bandit algorithms that do not account for such structural sparsity and motivates the need for methods that explicitly model the zero-inflated nature of the reward.
>
> Figure 1(b) compares illustrative confidence bounds under different algorithms. In addition to our proposed method and the Monte Carlo baseline (constructed from empirical samples), the other bounds correspond to those used by the UCB baselines described in Appendix C (Simulation Supplement). We acknowledge that these details were not sufficiently clarified in the main text, and we will revise the captions and discussion accordingly to make the interpretations of both Figure 1(a) and 1(b) more transparent in the updated manuscript.
>
> *References:*
>
> [1] Liu, X., Deliu, N., Chakraborty, T., Bell, L., & Chakraborty, B. (2023). Thompson sampling for zero-inflated count outcomes with an application to the Drink Less mobile health study. arXiv preprint arXiv:2311.14359.
>
> [2] Lattimore, T., & Szepesvári, C. (2020). Bandit algorithms. Cambridge University Press.

---

> > ### Comment · Reviewer_AjGD · 2025-04-04
> >
> > Thank you for your detailed response and congratulations of a well-written paper with positive reviews. I welcome the commitments to make improvements to the paper, and retain my accept score.

---

### Official Review · Reviewer_5ywx · 2025-03-14

**Overall Recommendation:** 3

**Summary:**

This paper “Zero-Inflated Bandits” focuses on the issue of sparse rewards in multi-armed bandit (MAB) and contextual bandit applications. The authors propose a zero-inflated bandit (ZIB) algorithm framework to enhance learning efficiency by leveraging the zero-inflated distribution structure.For zero-inflated multi-armed bandits, the paper presents a model that characterizes the reward distribution with parameters such as non - zero probability \(p_{k}\) and mean of the non - zero part \(\mu_{k}\). To address the shortcomings of naive approaches, the product method is introduced to construct more effective upper confidence bounds (UCB). The Thompson Sampling (TS) approach is also extended for this model. In the context of zero-inflated contextual bandits, the model is further extended, and UCB and TS algorithm templates are proposed. These algorithms construct confidence bounds for exploration and estimate parameters to optimize decisions.Theoretical analysis of the regret bounds for UCB and TS algorithms in both MAB and contextual bandits is conducted. Extensive experiments, including simulations in MAB and contextual bandits and a real - data application, demonstrate that the proposed UCB and TS algorithms consistently achieve lower sub - linear regrets, outperforming baseline methods that ignore the zero-inflated structure or directly quantify uncertainty.

**Claims And Evidence:**

The independence assumptions in the TS algorithm proofs are too strong. In reality, factors affecting rewards are often interrelated. I suggest the authors list all assumptions in a dedicated section, discuss their implications, justifications, and potential consequences of violation to enhance the research's transparency.

**Essential References Not Discussed:**

There are some citation omissions. The work by Peng Y, et al. titled "A practical semi-parametric contextual bandit" presented at IJCAI in 2019 should be included in the references. This paper likely contributes to the semi-parametric bandit literature and its omission undermines the comprehensiveness of the paper's literature review. Another is Liu X, et al. Thompson sampling for zero-inflated count outcomes with an application to the Drink Less mobile health study, which is very related to this work.

**Experimental Designs Or Analyses:**

(a) There is an absence of comparison with heavy tail and long tail bandit algorithms in experiments. Given the relevance, such comparisons are crucial to assess the proposed algorithms' performance comprehensively. (b) The absence of comparisons with semi-parametric bandit algorithms in the experiments is a significant oversight. Given the relevance of semi-parametric bandit research to the topic of this paper, such comparisons are essential for a comprehensive evaluation of the proposed algorithms. (c) if the author could conduct a real online AB testing for the method, it would be better because the real environment will break assumptions normally.

**Methods And Evaluation Criteria:**

The paper's omission of analyzing the time and space complexity of the proposed algorithms is a weakness. Understanding the computational resources required by these algorithms is crucial for practical applications, especially in large - scale and real - time scenarios. Without such analysis, it's hard to assess the algorithms' scalability and efficiency. I recommend the authors conduct and report this analysis.

**Other Comments Or Suggestions:**

None

**Other Strengths And Weaknesses:**

None

**Questions For Authors:**

How can one pre-determine whether the current environment is long-tailed or zero-inflated before utilization?

**Relation To Broader Scientific Literature:**

Focusing on zero-inflated bandits is valuable for real - world applications with sparse rewards, and the proposed algorithms can enhance learning efficiency.

**Theoretical Claims:**

The paper briefly mentions the link to heavy tail and long tail bandit research. Since zero-inflated distributions can be a special case of heavy tail distributions, a more in - depth discussion is needed. This should cover how the proposed algorithms relate to existing ones in these areas and how the distributions' properties interact.

---

> ### Author Rebuttal · Authors · 2025-03-31
>
> Thank you for your thorough and constructive feedback. Below, we respond to each of your points. We hope this addresses your concerns.
>
> *Claims & Evidence:*
>
> Our main independence assumption is that $X$ (nonzero rewards) and $Y$ (the indicator of a nonzero outcome) are independent in the decomposition $R = X \cdot Y$. We do not assume independence across arms or across time. This independence assumption is introduced mainly for notation simplification and analytical tractability, as discussed in lines L130–L140. We agree we should state this more explicitly in both the MAB and GLM contexts.
>
> *Methods & Evaluation Criteria:*
>
> We appreciate the suggestion to detail computational cost. We have found that our ZI-based approach retains the same big-O complexity as standard baselines for both MAB and GLM. The only difference is a small constant overhead from maintaining two estimators (one for the zero indicator, one for nonzero magnitude) instead of a single estimate of $R$. We will clarify this in our revision.
>
> *Theoretical Claims:*
>
> Thank you for your insightful suggestion. We agree that connections to heavy-tailed bandits [1,2,3] and asymmetric bandits [4,5,6] are relevant and worth highlighting. While our approach is specifically tailored to the ZI structure, it remains compatible with heavy-tailed settings (e.g., allowing only $(1+\epsilon)$-th moments as in [1,2,3]) and provides a structural alternative to modeling asymmetry, rather than relying on auxiliary tools like empirical quantiles or calibration, which increase computational complexity (e.g., [5] and [6]). We will incorporate these related works and clarify distinctions in our revision—thank you again for the valuable feedback.
>
> *Experimental Designs & Analyses:*
>
> Following your advice, we included Q-SAR [6] for MAB and SPUCB [7] for GLM as baselines in two uploaded anonymous figures ([Figure 1](https://anonymous.4open.science/r/ZIB_ICML-2535/MAB_extra_QSAR.pdf) and [Figure 2](https://anonymous.4open.science/r/ZIB_ICML-2535/GLM_CB_extra.pdf)). In zero-inflated regimes, Q-SAR's reliance on quantile updates can become unstable when zeros dominate; it tends to over/underestimate crucial quantiles. In contrast, our ZI-based methods explicitly model the sparse reward mechanism, yielding more stable learning and lower regret in both MAB and contextual settings.
>
> We also evaluated a standard A/B testing baseline (uniform allocation) in a zero-inflated Gaussian environment with periodically drifting means, as shown in the [anonymous figure](https://anonymous.4open.science/r/ZIB_ICML-2535/AB_testing_new.pdf). This controlled synthetic setting introduces nonstationarity by perturbing each arm’s mean with Gaussian noise (standard deviation 5) every $T/3$ rounds. We also explored alternative drift magnitudes and observed qualitatively similar trends. Even under moderate nonstationarity, our UCB and TS methods adapt better than fixed allocation. We will include these comparisons in the revised appendix.
>
> *Supplementary Material & Essential References Not Discussed:*
>
> We agree the current appendix is lengthy. To address this, we will: (1) Add a summary table at the appendix start, mapping each section to its key results; (2) Reduce repetition and highlight core takeaways, ensuring a clearer structure. We will also reference in the related work and clarify the ties to zero-inflated approaches.
>
> *Questions:*
>
> Determining whether an environment is zero-inflated or heavy-tailed can be guided by domain knowledge (e.g., high-frequency ``no feedback" in ads or loan offers) and data diagnostics (e.g., moment tests [9], sub-G plots [10]). Though a rigorous classification lies beyond this paper’s scope, we consider it an important direction for real-world deployments.
>
> References:
>
> [1] Bubeck, S., Cesa-Bianchi, N., & Lugosi, G. (2013). Bandits with heavy tail.
>
> [2] Zhang, J., & Cutkosky, A. (2022). Parameter-free regret in high probability with heavy tails.
>
> [3] Cheng, D., Zhou, X., & Ji, B. (2024). Taming Heavy-Tailed Losses in Adversarial Bandits and the Best-of-Both-Worlds Setting.
>
> [4] Zhang, M., \& Ong, C. S. (2021). Quantile bandits for best arms identification.
>
> [5] Shi, Z., Kuruoglu, E. E., & Wei, X. (2022). Thompson Sampling on Asymmetric Stable Bandits.
>
> [6] Zhang, M., & Ong, C. S. (2021). Quantile bandits for best arms identification.
>
> [7] Peng, Y., Xie, M., Liu, J., Meng, X., Li, N., Yang, C., … & Jin, R. (2019, August). A practical semi-parametric contextual bandit.
>
> [8] Liu, X., Deliu, N., Chakraborty, T., Bell, L., & Chakraborty, B. (2023). Thompson sampling for zero-inflated count outcomes with an application to the Drink Less mobile health study.
>
> [9] Trapani, L. (2016). Testing for (in)finite moments.
>
> [10] Zhang, H., Wei, H., & Cheng, G. (2023). Tight non-asymptotic inference via sub-Gaussian intrinsic moment norm.

---

### Official Review · Reviewer_ZEJa · 2025-03-14

**Overall Recommendation:** 3

**Summary:**

The submission studies multi-armed bandits whose reward function follows the zero-inflated (ZI) distribution. The motivation is to investigate the advantages of distribution modeling and exploiting the problem specific structure. UCB and TS are modified to solve the MAB and the contextual bandit problems under the zero-inflated regime. The corresponding regret bounds and experimental verifications are provided.


## Update after rebuttal

The authors' reply addressed my concerns. Given that, I have revised my recommendation to weak accept.

**Claims And Evidence:**

Theoretical claims are supported by proofs.

Experimental claims have less support. See Experimental Designs Or Analyses for more comments.

**Essential References Not Discussed:**

Literature with sparse rewards should be included as a part of related work. Related papers are https://arxiv.org/abs/1706.01383 and https://proceedings.neurips.cc/paper_files/paper/2023/hash/9408564a4229f4a933ac9bd09a29ee96-Abstract-Conference.html.

**Experimental Designs Or Analyses:**

There is only one real-world dataset used to verify the proposed method. However, in Introduction, there are other practical scenarios that can be used to verify the proposed methods [054R–081L]. In addition, the baselines in the real-world experiment are UCB and TS, which are developed for MAB. Instead, appropriate baselines should be the contextual bandit algorithms.

One motivation for proposing the ZI model is to deal with sparse reward signals. Baselines aimed at tackling sparse rewards should be included in the synthetic experiments. Related papers are https://arxiv.org/abs/1706.01383 and https://proceedings.neurips.cc/paper_files/paper/2023/hash/9408564a4229f4a933ac9bd09a29ee96-Abstract-Conference.html. Moreover, the proposed method does not always perform the best in figures 5 [1032] and 7 [1131]. The pros and cons of the proposed method are not discussed in the main text.

**Methods And Evaluation Criteria:**

The notion of regret is an appropriate evaluation criterion for the theoretical results.

The synthetic datasets and the real-world dataset are suitable for evaluating the ZI methods.

**Other Comments Or Suggestions:**

Please see the comments in the above parts.

**Other Strengths And Weaknesses:**

Weaknesses

The submission did not justify the rationale for the ZI model [118L]. Why is ZI a good model for sparse rewards while the others are not? What is the intuition behind $X$ and $Y$? Why does $X$ have to be defined by $\mu$ and $\epsilon$? Can’t $X$ just be a distribution? Why is $\epsilon$ restricted to sub-Weibull, sub-Gaussian, and heavy-tailed only?

There is no discussion of the applicability of ZI in realistic scenarios (applications are mentioned in Introduction, but it is difficult for a reader to connect the applications to the modeling part [110L–140L]).

What is the feedback signal? There are X, Y, R in the main text [110L–140L], but only R and Y in the algorithm.

There are unreferenced pointers ([177R] and [1254]).

**Questions For Authors:**

Please see the questions in the above parts.

**Relation To Broader Scientific Literature:**

The developed algorithms and the technical contributions provide bandit solutions for the ZI regime. It is, however, unclear if the technique of this submission can be generalized to other regimes.

**Theoretical Claims:**

The submission provides matching regret bounds for the ZI methods. Due to the multiplicative nature of the reward function (equation (1)), the submission developed technical tools to facilitate the proof. These technical contributions are Lemmas E.2 [1391], E.3 [1480], E.4 [1502].

However, if the motivation of this submission is to investigate the benefits of exploiting the ZI structure, then proving matching regret bounds is simply a sanity check on the proposed methods. Ideally, the proofs should show when and how the regret bounds are improved by considering the ZI structure. Unfortunately, these results are lacking.

---

> ### Author Rebuttal · Authors · 2025-03-31
>
> Thank you for your detailed review. We greatly value your time and suggestions, and we hope the following clarifications and enhancements address your concerns. We respectfully ask you to consider revising your evaluation score if our replies resolve your reservations.
>
> *Theoretical Claims:*
>
> Our principal contribution lies not only in matching known regret bounds (a necessary sanity check) but in showing how and why leveraging the ZI structure yields performance gains. As discussed in Section 2 ([R110–R149]) and supported by Lemma D.1 [1278] and Figure 10 [1347], ignoring ZI can inflate variance estimates and produce looser concentration bounds. In contrast, identifying and estimating the nonzero reward part more accurately (rather than lumping all observations into a single variance term) avoids under-exploration. This insight is partly recognized in various works (e.g., [1], [2]) but had not been systematically applied to ZI bandits. We will revise the main text to highlight the link between better variance estimation (using ZI) and improved regret.
>
> *Experimental Designs & Analyses:*
>
> - In Section 5 ([R345–R356]), we indeed used contextual UCB and TS baselines that incorporate covariates via a GLM-based model (described in Appendix C.2). We will clarify this in the main text.
>
> - Our chosen U.S. online auto loan dataset is both large and exhibits notable ZI properties (high reward sparsity, heterogeneous covariates). Additional datasets are desirable but space-limited, and we plan to expand this line of empirical validation in future work.
>
> - We compare our UCB (or TS) method to other UCB (or TS) algorithms in each experimental setting. Occasional underperformance against certain proxy-base UCB methods occurs mainly with exponential rewards when $p_k \sim U[0.1, 0.3].$ However, as shown in Lemma D.1, such proxies can become unreliable under high variance or ZI. Our approach is generally more robust and practical due to directly modeling ZI.
> We have also added two uploaded anonymous figures ([Figure 1](https://anonymous.4open.science/r/ZIB_ICML-2535/size_ratio_1.pdf) and [Figure 2](https://anonymous.4open.science/r/ZIB_ICML-2535/size_ratio_2.pdf)) to illustrate how large variance-to-mean ratios may briefly favor alternative baselines but ultimately highlight the value of stable ZI modeling.
>
> *Essential References Not Discussed:* We appreciate you pointing us to [3] and [4]. These works consider sparsity in different senses (e.g., many arms having zero mean or zero losses in partial monitoring). Our ZI setting, by contrast, deals with a stochastic zero draw, even for arms with nonzero mean, leading to different concentration/variance behaviors. We will clarify these distinctions in our introduction and related work sections.
>
> *Weaknesses:*
>
> - Rationale: Our approach targets scenarios where actions yield zero reward with high probability (not merely zero mean). This modeling accurately depicts domains like loan offers, recommender systems, or online ads, where a user typically rejects or ignores an action, creating a structural zero.
>
> - Decomposing Rewards: We define $R = X \times Y$, with $Y = 1(R \neq 0)$ and $X = 1(R \neq 0) \cdot R$. This decomposition (discussed in [L130–L140]) separates the high-probability zero event from the nonzero reward.
>
> - Intuition Behind $X, Y, \mu$, and $\epsilon$: We let $\mu = E[X]$ and $\epsilon = X - \mu$ to center analyses on deviation from the mean. We classify $\epsilon$ into sub-Gaussian, sub-Weibull, or heavy-tailed categories, each with different tail decay properties (following [1], [2]). Handling these classes separately is standard in bandit theory. More extreme no-moment or adversarial settings (e.g., [5]–[8]) fall outside this paper’s scope, though we will mention them as possible extensions.
>
> - Observed Variables: Only $R$ is directly observed during interaction. We define $X$ and $Y$ to analyze the structure of zero vs. nonzero rewards, but the algorithm indeed only sees $R$. We will clarify this point.
>
> - Unreferenced Pointers: We have fixed [1254] (now pointing to Section 5) and [177R] (referring to a corollary comparing constants with classical UCB). These errors will be corrected in the revised manuscript.
>
> References:
>
> [1] Lattimore, T., & Szepesvári, C. (2020). Bandit Algorithms.
>
> [2] Zhou, P., Wei, H., & Zhang, H. (2024). Selective Reviews of Bandit Problems in AI via a Statistical View.
>
> [3] Kwon, J., Perchet, V., & Vernade, C. (2017). Sparse Stochastic Bandits.
>
> [4] Tsuchiya, T., Ito, S., & Honda, J. (2023). Stability-Penalty-Adaptive FTRL: Sparsity, Game-Dependency, and Best-of-Both-Worlds.
>
> [5] Yun, H., & Park, B. U. (2023). Exponential concentration for geometric-median-of-means.
>
> [6] Bubeck, S., Cesa-Bianchi, N., & Lugosi, G. (2013). Bandits with heavy tail.
>
> [7] Zhang, J., & Cutkosky, A. (2022). Parameter-free regret in high probability with heavy tails.
>
> [8] Cheng, D., Zhou, X., & Ji, B. (2024). Taming Heavy-Tailed Losses

---

> > ### Comment · Reviewer_ZEJa · 2025-04-02
> >
> > Thank you for the reply. My concerns are addressed. I will revise my score.

---

> > > ### Author Response · Authors · 2025-04-04
> > >
> > > Thank you so much for your constructive and positive feedback, and appreciating our rebuttal!!

---

### Official Review · Reviewer_TjeF · 2025-03-14

**Overall Recommendation:** 4

**Summary:**

This paper considers a multi-armed bandit setting, where reward distributions are contaminated with a zero point mass with weight $p$. To accommodate this special reward structure, which returns rewards of 0 with probability $1-p$, and rewards distributed according to an arm-specific sub-Weibull distribution otherwise. The authors leverage a product trick, which uses a union bound over the uncertainty in parameter $p$ and the mean of the nonzero distribution to obtain product confidence intervals, which underlie the modified UCB and Thompson Sampling algorithms proposed in their work. The authors provide theoretical results that show their approach achieves known minimax lower bounds, and their empirical results demonstrate the benefits of their approach on both synthetic and real-world data.

**Claims And Evidence:**

All claims and proofs are well supported, although the regret rates could be discussed more cleanly with respect to existing work.

For example, as long as the mixed distribution satisfies some subgaussian (or sub-Weibull) property, then minimax regret rates will be attained by specifying the correct subgaussian (or sub-Weibull) factor. The main gain (at least in terms of order optimality of regret) is only in the heavy-tailed case, which could be emphasized more cleanly.

**Essential References Not Discussed:**

All relevant references seem to be present in this work.

**Experimental Designs Or Analyses:**

The experiments do correspond closely with the claims of the authors, and capture the multiple nonzero reward generating distributions that may be contaminated with a zero point mass.

It would be helpful to have higher simulation numbers - 50 and 25 simulations for figures 2 and 3 seems somewhat smaller than expected. Likewise, it would be helpful to get better intuition on why this approach works well for synthetic data, and works relatively worse on the real world data.

**Methods And Evaluation Criteria:**

The proposed methods and evaluation criteria (for both the synthetic example and real-world experiment) evaluate the proposed method.

One interesting case to see (at least from an empirical standpoint) would be bounded rewards (rather than just sub-gaussian reward distributions), which often occurs in practice. It may be the case that modeling the nonzero distribution may be most beneficial when the nonzero distribution component is skewed far away from zero.

It could also be interesting to test larger values of $p$ - this setting could also be adversarial to the benefits of this approach.

**Other Comments Or Suggestions:**

There seems to be a reference error in Line 176 of the manuscript (end of Section 2.1).

**Other Strengths And Weaknesses:**

We summarize the strengths and weaknesses of this paper below:

** Strengths **
* Most importantly, this work considers a common setup that occurs in practice, across many different fields. Zero-inflated distributions for arm rewards is a practically relevant setting to study.
* The authors offer a simple, computationally lightweight solution to this setup that requires little modifications to existing bandit algorithms.
* The method appears to perform empirically well, especially for heavy-tailed distributions.

** Weaknesses **
* For heavy-tailed distributions contaminated with a zero point mass, those distributions are no longer so heavy-tailed. It is unclear whether the modeling of zero inflation or a poorly specified scale factor is the cause of increased performance.
* The authors rely on a union bound, which is sufficient for order optimal regret. One wonders if there could be a better way to split the confidence (even if union bounding) than $\alpha/2$, or to avoid wasteful union bounds all-together.

**Questions For Authors:**

Beyond zero inflation, are there other reward models that are best captured with this hierarchical approach? The product method for constructing confidence intervals appears to be a general approach for more complicated reward generation (i.e., we don't necessarily need to fix one distribution to be Bernoulli). Are there other settings where this could be done?

**Relation To Broader Scientific Literature:**

The key results of this paper lies in its practical relevance - zero-inflated distributions are very common in settings such as digital advertisement. While this paper does not introduce novel tools, it provides a practical, simple solution for a setting that occurs in many practically relevant scenarios where bandits are applied.

**Theoretical Claims:**

Proofs were briefly skimmed, but not evaluated in detail.

---

> ### Author Rebuttal · Authors · 2025-03-31
>
> We sincerely thank you for your thoughtful and encouraging feedback. Below, we respond to your comments point by point, and we hope our replies provide clear and satisfactory answers to your questions.
>
> *Claims & Evidence:*
>
> While sub-Gaussian or sub-Weibull assumptions allow existing algorithms to attain minimax rates, ZI introduces unique challenges. In particular, a preponderance of zeros can inflate variance estimates and skew concentration. By explicitly modeling and leveraging the ZI structure, our approach preserves minimax guarantees across a wide array of distributions, including those with heavy-tailed nonzero components. We will revise Lemma 2.1 and related text ([L161–L164]) to emphasize these benefits more clearly.
>
> *Methods & Evaluation:*
>
> We tested our approach with bounded, skewed Beta rewards ([anonymous figure](https://anonymous.4open.science/r/ZIB_ICML-2535/MAB_for_Beta.pdf)) of the form $X_k = 2\mu_k Beta(p_k, p_k)$ so that $E[X_k] = \mu_k$. Our UCB method outperforms baselines in most settings except when zero-inflation is minimal, where an exact Hoeffding-based UCB can be slightly better (though it relies on unavailable knowledge). Even at high $p_k \sim U(0.75, 0.95)$ Low ZI ([anonymous figure](https://anonymous.4open.science/r/ZIB_ICML-2535/MAB_for_large_p.pdf)), our approach remains strong under Gaussian and Exponential components. We will add these experiments to the appendix.
>
> *Experimental Designs & Analyses:*
>
> We agree that increasing simulation replications to 100 will yield more reliable comparisons, and we have begun doing so for both MAB and GLM contextual bandits. The performance gap between synthetic and real data likely stems from unobserved confounders in the real-world dataset; still, under the same ZI assumptions, our UCB and TS methods outperform other baselines. We will clarify these observations in the revision.
>
> *Weakness 1:*
>
> Our discussion of heavy-tailed rewards refers to the conditional distribution of nonzero outcomes, since the zero mass makes the entire distribution no longer heavy-tailed in the strict sense. By modeling the zero mechanism separately, our approach avoids erroneously inflating or deflating uncertainty due to frequent zeros, enhancing robustness across diverse reward distributions.
>
> *Weakness 2:*
>
> As highlighted in [L421–R423], our current allocation of the failure probability between the concentration bounds for $X$ and $Y$ can be conservative. A more refined analysis, possibly peeling confidence sets for $X$ and $Y$ individually, could yield tighter bounds and smaller constants, aligning with advanced concentration methods (e.g., Section 9.3 of [1], Section 1.2 of [2]). We will mention these refinements in the revision.
>
> *Other Comments & Suggestions:*
>
> We have fixed the reference error at [176R]. Furthermore, as noted in our Broader Implications [R424–R436], our product-form confidence intervals naturally extend to hierarchical or multi-layer reward structures. By decomposing the variance of each sub-component and applying Freedman or Bernstein-type bounds, one can tackle even more complex reward mechanisms. This direction may be especially relevant for recommendation systems or multi-stage decision-making.
>
> For instance, consider a reward model of the form
> $$
>          (Y_1, \ldots, Y_m) \sim \operatorname{Multi} (1; p_1, \ldots, p_m), \qquad R = \sum_{j = 1}^m X_j Y_j,
> $$
> where the reward is determined by sampling one of the $X_j$’s with probability $p_j$. This model captures structured reward uncertainty arising from latent selection or allocation mechanisms. In such settings, the reward inherits a mixture structure that can be decomposed, and concentration bounds can be constructed component-wise. Specifically, Freedman's inequality or Bernstein-type bounds for martingales can be adapted to leverage variance information
> $$
> P(S_n - n r > t) \leq P (S_n - nr \geq t, V_n \leq v) + P (V_n > v),
> $$
> where $S_n$ is the cumulative reward and $V_n$ is its empirical variance. The first term allows for tighter bounds via Bernstein-type inequalities when variance is controlled, while the second term can be analyzed similarly to our ZI setting, noting that
> $$
>          V =\sum_{j = 1}^m p_j E [X_j]^2 - \bigg( \sum_{j = 1}^m p_j E [X_j] \bigg)^2.
> $$
> This decomposition lends itself naturally to algorithm design, in which estimates and confidence bounds are tracked separately for each layer of the reward model. We believe such hierarchical formulations represent a promising direction for future research beyond ZI, especially in domains like recommendation systems, multi-stage decision-making, or online pricing, where reward generation often involves latent stochastic mechanisms.
>
> *References:*
>
> [1] Lattimore, T., & Szepesvári, C. (2020). Bandit Algorithms. Cambridge University Press.
>
> [2] Ren, H., & Zhang, C. H. (2024). On Lai’s Upper Confidence Bound in Multi-Armed Bandits. arXiv preprint arXiv:2410.02279.

---

### Decision · Program_Chairs · 2025-05-01

**Decision:**

Accept (poster)

**Comment:**

This paper discusses a variant of bandit problems where zero-reward appears with high probability and the nonzero part follows some sub-Weibull distributions. UCB-based and Thompson-Sampling-based algorithms are proposed with some regret bounds. For this paper the reviewers agreed in the opinion that the formulation of the paper is well-motivated and the algorithms are supported by good theoretical and empirical results. On the other hand, many concerns have been raised by the reviewers. In particular, the concern on the substantial benefit of considering the zero-inflated setting is shared between the reviewers, which I also agree. Though these concerns seem to be largely solved by the rebuttal, I expect that the authors thoroughly address theses concerns in the final version.